# Modeling Covariate Transition for Efficient Estimation of Longitudinal Treatment Effects in Randomized Experiments

Naoki Chihara [1†]  Tatsushi Oka [2]  Yasuko Matsubara [1]  Yasushi Sakurai [1]  Shota Yasui [3]

## Abstract

We present a regression-adjustment framework designed for the estimation of longitudinal treatment effects in randomized experiments under static regimes. While regression-adjustment methods are useful for variance reduction in randomized experiments by using pre-treatment covariates, they usually focus only on average effects, from which we cannot obtain valuable insights into when the effects appear and how long they continue. To address this issue, we consider intermediate outcomes and evolving post-treatment covariates over time, and we represent such dynamic trajectories using transition kernels. Furthermore, we establish the asymptotic normality and the semiparametric efficiency bound for our estimator, enabling more powerful statistical inference. Simulation studies and empirical analysis using A/B test data from a streaming platform in Japan show the practical advantages of our method.

## 1. Introduction

Randomized experiments (Fisher, 1935) have been the gold standard for inferring causal effects and are widely employed across various domains, including medicine (Rubin, 1997) and economics (Duflo et al., 2007; Athey & Imbens, 2017). However, the statistical power is limited, and large sample sizes are needed to obtain reliable results, as reported in online experiments (Lewis & Rao, 2015). To address this challenge, regression adjustment has been widely used to reduce the variance of treatment effect estimators by utilizing auxiliary covariates collected in randomized experiments (Lin, 2013; Deng et al., 2013; Imbens & Rubin, 2015; Bloniarz et al., 2016; Oka et al., 2026).

Although regression adjustment under randomized experiments is widely employed for variance reduction in treatment effect estimation, standard methods typically utilize only pre-treatment covariates to estimate the average treatment effect (ATE). As a result, these methods overlook important insights into the temporal evolution of effects induced by early treatments. This limitation leads to a significant missed opportunity, especially in modern applications, where randomized experiments are often conducted over extended periods under static regimes. For example, the ATE does not capture the essential temporal dynamics of the response, such as whether the treatment yields an immediate impact, exhibits a delayed effect, or maintains persistence over time. Capturing these temporal dynamics requires incorporating information from intermediate outcomes and evolving post-treatment covariates. However, naively conditioning on such post-treatment variables allows for variance reduction but introduces post-treatment bias, distorting the target estimand even in randomized settings (Freedman, 2008a; Montgomery et al., 2018). For instance, in streaming services, user preferences dynamically evolve in response to recommendation algorithms. Adjusting for these evolving preferences as covariates blocks the causal pathway, thereby failing to capture the time-evolving impact of treatments on long-term engagement.

In this paper, we propose a regression adjustment method for the estimation of longitudinal treatment effects under randomized experiments. A key advantage of our framework is its ability to incorporate dynamic transitions in covariates induced by treatments into the estimation procedure. To account for temporal dynamics, we exploit transition kernels to represent how past treatments shape future histories and propagate outcome predictions forward along this kernel. This forward integration aggregates information contained in post-treatment covariates while preserving the marginal target estimand, thereby avoiding the post-treatment bias inherent in naive conditioning. As a by-product, this explicit modeling allows us to directly incorporate domain knowledge regarding system dynamics, such as seasonality or known exogenous shocks, into the transition mechanism.

In addition, the proposed method builds upon the Neyman-orthogonal moment conditions (Chernozhukov et al., 2018;

[†]Work done during research internship at CyberAgent AI Lab. [1]SANKEN, The University of Osaka, Osaka, Japan. [2]Keio University, Tokyo, Japan. [3]CyberAgent, Inc., Tokyo, Japan. Correspondence to: Naoki Chihara <naoki88@sanken.osaka-u.ac.jp>.

*Proceedings of the 43$^{rd}$ International Conference on Machine Learning*, Seoul, South Korea. PMLR 306, 2026. Copyright 2026 by the author(s).

2022), which provide robustness against first-order estimation errors in high-dimensional or complex nuisance components. These nuisance functions are estimated using flexible machine learning methods, making our method model-agnostic. Incorporating cross-fitting further strengthens robustness against estimation errors. With this design, we derive the asymptotic distribution of the proposed estimator and establish its semiparametric efficiency bound under mild regularity conditions. These results enable statistical inference, including standard error estimation and the construction of confidence intervals.

**Contributions.** The main contributions of this paper are: (i) we present the dynamic regression-adjustment method using transition kernels and forward integration under randomized experiments; (ii) we establish asymptotic normality and provide valid inference under cross-fitted machine learning nuisance estimation, generalizing beyond traditional settings in causal inference; (iii) we derive the semiparametric efficiency bound for our estimator and show that it attains this bound; (iv) simulation studies and empirical analyses demonstrated the practical advantages of our approach.

**Outline.** The rest of our paper is organized in a conventional format. After the introduction, we review related works in Section 2 and provide our problem definition in Section 3. Next, we introduce the dynamic regression-adjusted estimator and its estimation procedure in Section 4 and the asymptotic properties of our results in Section 5. We then provide our experimental results and discussion in Section 6, followed by a conclusion in Section 7. All of the proofs for our theoretical results are provided in Appendix C.

## 2. Related Work

**Longitudinal treatment effects.** Early seminal works in epidemiology and biostatistics established principled frameworks for estimating longitudinal treatment effects (Robins, 1986; Robins et al., 2000; Robins, 1994). Recently, many researchers in the machine learning community have been studying sequential decision making in adaptive experimental designs, such as multi-armed bandits (Lattimore & Szepesvári, 2020). The adaptive nature of these approaches poses significant challenges for statistical inference, and it has been reported that they may introduce bias and make it harder to estimate causal effects (Shin et al., 2021; Hadad et al., 2021). In this context, policy evaluation in contextual bandits and reinforcement learning (Dudik et al., 2011; Jiang & Li, 2016; Kallus & Uehara, 2020; Hadad et al., 2021), as well as dynamic treatment regimes (Murphy, 2003; Robins, 2004; Zhang et al., 2013; Lewis & Syrgkanis, 2021; Bradic et al., 2024), primarily aims to estimate the value of adaptive decision rules. Deep learning approaches have also been proposed for counterfactual prediction from temporal trajectories (Lim, 2018; Bica et al., 2020; Li et al., 2021;

Melnychuk et al., 2022; Frauen et al., 2025), while our focus is on estimators with formal statistical inference and efficiency analysis under randomized experiments. From a different perspective, surrogate index methods (Prentice, 1989; Athey et al., 2025) infer long-term treatment effects from short-term experimental results under strong surrogacy assumptions. Despite their respective strengths, the majority of these existing methods focus on estimating a single cumulative outcome, thereby obscuring the rich temporal dynamics of how treatment effects evolve at each specific time point. While the practical importance of tracking how treatment effects evolve over time has been widely recognized in large-scale online experiments (Hohnhold et al., 2015), formal efficiency theory for the full trajectory of marginal treatment effects in static randomized experiments remains limited.

**Regression adjustment.** Randomized experiments have been widely used for the unbiased identification of the parameter of interest, but they often suffer from relatively low statistical power, as reported in online experiments (Lewis & Rao, 2015). To tackle this challenge, regression adjustment using auxiliary covariates to improve precision in treatment effect estimation has been extensively studied (Fisher, 1925; Cochran, 1977; Yang & Tsiatis, 2001; Rosenbaum, 2002; Freedman, 2008b; Tsiatis et al., 2008; Rosenblum & van der Laan, 2010; Lin, 2013; Deng et al., 2013; Ghadiri et al., 2023). Recent work extends covariate adjustment to high-dimensional settings and modern machine learning, enabling variance reduction with many auxiliary variables (Wager et al., 2016; Bloniarz et al., 2016; Poyarkov et al., 2016; Guo et al., 2021; Lei & Ding, 2021; Masoero et al., 2023; Chiang et al., 2025). However, post-treatment covariates dynamically evolve in response to past treatments, and naive conditioning on them can distort the target estimand, even in randomized experiments (Freedman, 2008a; Montgomery et al., 2018). Our research tackles this difficulty using recursive forward integration based on transition kernels to exploit post-treatment histories for variance reduction while preserving the marginal estimand under longitudinal randomized experiments.

**Semiparametric estimation.** Our approach is grounded in semiparametric efficiency theory, which addresses the challenge of estimating low-dimensional parameters of interest in the presence of possibly infinite-dimensional nuisance components (Klaassen, 1987; Robinson, 1988; Bickel et al., 1993; Andrews, 1994; Newey, 1994; Robins & Rotnitzky, 1995). Building on this line of work, recent developments in double/debiased machine learning (DML) adapt semiparametric methods to accommodate flexible machine learning estimators for nuisance functions (Ichimura & Newey, 2022; Byambadalai et al., 2024; 2025b;a; Ahrens et al., 2025). Specifically, our work leverages Neyman-orthogonal moment conditions (Neyman, 1959; Chernozhukov et al.,

2022) combined with cross-fitting (Chernozhukov et al., 2018). This strategy mitigates the impact of slow convergence rates and overfitting of complex nuisance estimators, yielding valid asymptotic distributions.

To the best of our knowledge, this is the first work to develop a regression-adjustment estimator for longitudinal treatment effects in randomized experiments that explicitly leverages covariate transition modeling using transition kernels and to establish that our algorithm attains the semiparametric efficiency bound for our estimator under this formulation.

## 3. Problem Definition

First, we introduce key notations used in this paper. Please see Appendix A for details. We consider randomized controlled trials (RCTs) with longitudinal treatments $\{W_t\}_{t=1}^T$ under static regimes, where $T$ is the time horizon and $W_t \in \mathcal{W}_t := \{1, \dots, |\mathcal{W}_t|\}$ is the treatment assignment at time $t$. Let $Y_t \in \mathcal{Y}_t \subseteq \mathbb{R}$ denote the scalar-valued observed outcome of interest at time $t$ and $X_t \in \mathcal{X}_t \subseteq \mathbb{R}^d$ denote the covariates at time $t$. For each time $t$, we observe $n$ random samples $\{Z_{i,t}\}_{i=1}^n = \{(X_{i,t}, W_{i,t}, Y_{i,t})\}_{i=1}^n$ from a distribution on the product space $\mathcal{Z}_t := \mathcal{X}_t \times \mathcal{W}_t \times \mathcal{Y}_t$. We denote $\bar{X}_{i,t} = \{X_{i,1}, \dots, X_{i,t}\}$ with $\bar{W}_{i,t}, \bar{Y}_{i,t}$ defined analogously, and $Z_t \in \mathcal{Z}_t$ denotes the generic data point at time $t$. The probability of treatment assignment $\bar{w}_t$ is denoted as $\pi_{\bar{w}_t} = P(\bar{W}_{i,t} = \bar{w}_t)$ satisfying $\sum_{\bar{w}_t \in \bar{\mathcal{W}}_t} \pi_{\bar{w}_t} = 1$, while $n_{\bar{w}_t}$ indicates the number of observations in treatment history $\bar{w}_t$ satisfying $\sum_{\bar{w}_t \in \bar{\mathcal{W}}_t} n_{\bar{w}_t} = n$. Our work follows the potential outcome framework (Rubin, 1974; Imbens & Rubin, 2015) in accordance with conventional settings and we let $Y_t(\bar{w}_t)$ and $X_t(\bar{w}_{t-1})$ denote the potential outcome and covariate under longitudinal treatments $\bar{w}_t$.

**Remark 1** (Feasible longitudinal treatments $\bar{W}_t$). *Generally, the feasible longitudinal treatment set $\bar{\mathcal{W}}_t$ is the support of $\bar{W}_t$. For conventional designs, the cardinality $|\bar{\mathcal{W}}_t|$ is small (e.g., $|\bar{\mathcal{W}}_t| = 2$ in static A/B testing).*

Here, we provide the assumptions used in this paper.

**Assumption 1** (SUTVA). *No interference and consistency.*

**Assumption 2** (Independent units). *For any $t \in [T]$, the random samples $\{Z_{i,t}\}_{i=1}^n$ are i.i.d across units.*

**Assumption 3** (Randomization). *Treatment assignment is independent, i.e., $\{\bar{Y}_t(\bar{w}_t), \bar{X}_t(\bar{w}_{t-1})\} \perp\!\!\!\perp \bar{W}_t, \forall t \in [T]$.*

**Assumption 4** (Positivity). $0 < \pi_{\bar{w}_t} < 1, \forall \bar{w}_t \in \bar{\mathcal{W}}_t$.

### 3.1. Parameters of Interest

Our parameters of interest are the expected potential outcomes at time $t$, as follows:

$$\mu_{\bar{w}_t}(t) = \mathbb{E}[Y_t(\bar{w}_t)], \tag{1}$$

where $\bar{w}_t \in \bar{\mathcal{W}}_t$. Based on the parameters, we define the longitudinal average treatment effect (ATE) between treatments $\bar{w}_t, \bar{w}'_t$ as

$$\text{ATE}(t) = \mu_{\bar{w}_t}(t) - \mu_{\bar{w}'_t}(t). \tag{2}$$

Under Assumptions 1-4, the expected outcomes $\mu_{\bar{w}_t}(t)$ are identifiable without any covariates while potential outcomes $\{Y(\bar{w}_t)\}_{\bar{w}_t \in \bar{\mathcal{W}}_t}$ are unobserved variables. Specifically, a simple estimator for $\mu_{\bar{w}_t}(t)$ is given by

$$\widehat{\mu}_{\bar{w}_t}^{emp}(t) := \frac{1}{n_{\bar{w}_t}} \sum_{i: \bar{W}_{i,t} = \bar{w}_t} Y_{i,t}, \tag{3}$$

While this estimator is unbiased and consistent, we aim to enhance its precision using observed historical data.

## 4. Estimation Method

In this section, we present a new estimator that leverages observed historical data through covariate transition modeling. We also provide cross-fitted estimation and statistical inference procedures.

### 4.1. Regression-Adjusted Estimator

We begin by introducing the historical data $\bar{H}_t(\bar{w}_{t-1}) = (\bar{X}_t(\bar{w}_{t-1}), \bar{Y}_{t-1}(\bar{w}_{t-1})) \in \bar{\mathcal{H}}_t$. Note that the historical data at time $t = 1$ only has pre-treatment covariates, i.e., $H_1 = X_1$ and $Y_0 = 0$. We adopt a regression adjustment framework to incorporate historical data $\bar{H}_t(\bar{w}_{t-1})$ into an estimation procedure. For each time $t \in [T]$ and longitudinal treatment $\bar{w}_t \in \bar{\mathcal{W}}_t$, we define the mean regression function $m_{\bar{w}_t}^{(t)}(\bar{h}_t)$ as

$$m_{\bar{w}_t}^{(t)}(\bar{h}_t) = \mathbb{E}\big[Y_t(\bar{w}_t) \,|\, \bar{H}_t(\bar{w}_{t-1}) = \bar{h}_t\big]. \tag{4}$$

The conditional mean function can be estimated using various supervised learning methods, such as LASSO, random forests, boosted trees, or deep neural networks. Under Assumptions 1-4, we rewrite $\mu_{\bar{w}_t}(t)$ using the mean regression function $m_{\bar{w}_t}^{(t)}(\bar{h}_t)$ as

$$\mu_{\bar{w}_t}(t) = \int_{\bar{\mathcal{H}}_t} m_{\bar{w}_t}^{(t)}(\bar{h}_t) P_{\bar{H}_t}^{\bar{w}_{t-1}}(d\bar{h}_t), \tag{5}$$

where $P_{\bar{H}_t}^{\bar{w}_{t-1}}$ is the probability measure on $\bar{\mathcal{H}}_t$ under $\bar{w}_{t-1}$. A natural estimator of $\mu_{\bar{w}_t}(t)$ is given by

$$\begin{aligned}
\widehat{\mu}_{\bar{w}_t}^{adj}(t) = &\frac{1}{n_{\bar{w}_t}} \sum_{i: \bar{W}_{i,t} = \bar{w}_t} \left(Y_{i,t} - \widehat{m}_{\bar{w}_t}^{(t)}(\bar{H}_{i,t})\right) \\
&+ \frac{1}{n_{\bar{w}_{t-1}}} \sum_{i: \bar{W}_{i,t-1} = \bar{w}_{t-1}} \widehat{m}_{\bar{w}_t}^{(t)}(\bar{H}_{i,t}),
\end{aligned} \tag{6}$$

where $\widehat{m}_{\bar{w}_t}^{(t)}$ is an estimator for $m_{\bar{w}_t}^{(t)}$. The estimator takes the form of the well-known augmented inverse-propensity weighting estimator. However, the empirical support for $\bar{H}_t$ under the longitudinal treatments $\bar{w}_{t-1}$ can be sparse, which leads to a high-variance estimate of Eq. (6). In particular, when the units in the group $\{i : \bar{W}_{i,t-1} = \bar{w}_{t-1}\}$ coincide with those in $\{i : \bar{W}_{i,t} = \bar{w}_t\}$, we have $\widehat{\mu}_{\bar{w}_t}^{emp}(t) = \widehat{\mu}_{\bar{w}_t}^{adj}(t)$, which implies that no variance reduction is achieved.

## 4.2. Dynamic Regression-Adjusted Estimator

To mitigate the aforementioned problem, we complement Eq. (6) with a forward integration scheme over the dynamic trajectories in the historical data. Let $p_{\bar{w}_\tau}^{(\tau)}(dh_{\tau+1} \mid \bar{h}_\tau)$ be the transition kernel of $H_{\tau+1}$ given $\bar{h}_\tau$ under $\bar{w}_\tau$ and let $\widehat{p}_{\bar{w}_\tau}^{(\tau)}$ be an estimator for $p_{\bar{w}_\tau}^{(\tau)}$, where $\bar{w}_\tau := (w_1, \ldots, w_\tau)$ is the length-$\tau$ prefix of $\bar{w}_t$. The transition kernel can be learned using models that support conditional sampling, such as vector autoregressions, Gaussian processes, or deep neural networks. We decompose the probability measure using the transition kernels as follows:

$$P_{\bar{H}_t}^{\bar{w}_{t-1}}(d\bar{h}_t) = P_{H_1}(dh_1) \prod_{\tau=1}^{t-1} p_{\bar{w}_\tau}^{(\tau)}(dh_{\tau+1} \mid \bar{h}_\tau).$$

Substituting the above decomposition into Eq. (5), we obtain

$$\mu_{\bar{w}_t}(t) = \int_{\bar{\mathcal{H}}_t} m_{\bar{w}_t}^{(t)}(\bar{h}_t) P_{H_1}(dh_1) \prod_{\tau=1}^{t-1} p_{\bar{w}_\tau}^{(\tau)}(dh_{\tau+1} \mid \bar{h}_\tau).$$

We define the iterated conditional expectation $\Gamma_{\bar{w}_t}^{(1)}(h_1)$ under treatment history $\bar{w}_t$ as

$$\Gamma_{\bar{w}_t}^{(1)}(h_1) = \int_{\mathcal{H}_{2:t}} m_{\bar{w}_t}^{(t)}(\bar{h}_t) \prod_{\tau=1}^{t-1} p_{\bar{w}_\tau}^{(\tau)}(dh_{\tau+1} \mid \bar{h}_\tau), \quad (7)$$

and, recursively for $\tau = t - 1, \ldots, 1$,

$$\Gamma_{\bar{w}_t}^{(\tau)}(\bar{h}_\tau) = \int_{\mathcal{H}_{\tau+1}} \Gamma_{\bar{w}_t}^{(\tau+1)}(\bar{h}_{\tau+1}) p_{\bar{w}_\tau}^{(\tau)}(dh_{\tau+1} \mid \bar{h}_\tau), \quad (8)$$

and $\Gamma_{\bar{w}_t}^{(t)}(\bar{h}_t) = m_{\bar{w}_t}^{(t)}(\bar{h}_t)$. Intuitively, $\Gamma_{\bar{w}_t}^{(\tau)}(\bar{h}_\tau)$ is the conditional expectation of the terminal regression $m_{\bar{w}_t}^{(t)}(\bar{h}_t)$ after propagating the remaining future history from time $\tau + 1$ to $t$ according to the transition kernels under $\bar{w}_t$. Consequently, the target estimand satisfies $\mu_{\bar{w}_t}(t) = \mathbb{E}[\Gamma_{\bar{w}_t}^{(1)}(X_1)]$. Then, the resulting estimator augments the residual average with a fully marginalized iterated conditional expectation:

$$
\begin{aligned}
\widehat{\mu}_{\bar{w}_t}^{dy\text{-}adj}(t) = {} & \frac{1}{n_{\bar{w}_t}} \sum_{i:\bar{W}_{i,t}=\bar{w}_t} \left( Y_{i,t} - \widehat{\Gamma}_{\bar{w}_t}^{(t)}(\bar{H}_{i,t}) \right) \\
& + \frac{1}{n} \sum_{i=1}^{n} \widehat{A}_{\bar{w}_t}^{(t)}(\bar{H}_{i,t}) + \frac{1}{n} \sum_{i=1}^{n} \widehat{\Gamma}_{\bar{w}_t}^{(1)}(X_{i,1}),
\end{aligned}
\tag{9}
$$

where $\widehat{\Gamma}_{\bar{w}_t}^{(1)}$ is an estimator for $\Gamma_{\bar{w}_t}^{(1)}$ and $\widehat{A}_{\bar{w}_t}^{(t)}$ is an estimator for the auxiliary correction term $A_{\bar{w}_t}^{(t)}$ defined as follows:

$$
\begin{aligned}
A_{\bar{w}_t}^{(t)}(\bar{H}_t) = {} & \sum_{\tau=1}^{t-1} \frac{\mathbb{1}\{\bar{W}_\tau = \bar{w}_\tau\}}{\pi_{\bar{w}_\tau}} \left\{ \Gamma_{\bar{w}_t}^{(\tau+1)}(\bar{H}_{\tau+1}) \right. \\
& \left. - \int_{\mathcal{H}_{\tau+1}} \Gamma_{\bar{w}_t}^{(\tau+1)}(\bar{H}_\tau, h_{\tau+1}) p_{\bar{w}_\tau}^{(\tau)}(dh_{\tau+1} \mid \bar{H}_\tau) \right\}.
\end{aligned}
\tag{10}
$$

The second term in Eq. (9) compensates for discrepancies between realized and one-step transitions via the auxiliary term $A_{\bar{w}_t}^{(t)}$, and the third integrates $\widehat{m}_{\bar{w}_t}$ along the transition under $\bar{w}_t$. The transition $\widehat{p}_{\bar{w}_\tau}^{(\tau)}$ supplies a forward model for the trajectory of covariates, enabling a stable approximation to Eq. (5). The conditional regression functions $m_{\bar{w}_t}$ and the transition kernels $p_{\bar{w}_\tau}^{(\tau)}$ are treated as nuisance functions.

**Remark 2** (Role of transition kernels $p_{\bar{w}_\tau}^{(\tau)}$). *Modeling covariate transition dynamics is significant to utilize post-treatment covariates, as simple adjustment leads to the cancellation issue described in Eq. (6). The transition kernel enables us to calculate the expected trajectory and isolate innovations, i.e., the stochastic deviations of realized covariates from their expectations. By capturing these innovations in the auxiliary correction term $A_{\bar{w}_t}^{(t)}(\bar{H}_t)$, we extract new information resolved during the experiment that is orthogonal to pre-treatment covariates, thereby achieving variance reduction without bias.*

## 4.3. Moment Condition Problem

We rewrite our estimation problem as a moment condition problem. This formulation is crucial for establishing the asymptotic properties of our estimator when nuisance functions are estimated via machine learning (ML). We define $m_t := (m_{\bar{w}_t}^{(t)}(\cdot))_{\bar{w}_t \in \bar{\mathcal{W}}_t}$ and $p_t := (p_{\bar{w}_t})_{\bar{w}_t \in \bar{\mathcal{W}}_t}$, where $p_{\bar{w}_t} := \{p_{\bar{w}_\tau}^{(\tau)}\}_{\tau=1}^{t-1}$. In addition, let $\theta_t = (\mu_{\bar{w}_t}(t))_{\bar{w}_t \in \bar{\mathcal{W}}_t}$ be the target estimand. We define the moment functions as

$$\psi_t^\pi(\bar{Z}_t; \theta_t, m_t, p_t) = \left( \psi_{\bar{w}_t}^\pi(\bar{Z}_t; \theta_t, m_t, p_t) \right)_{\bar{w}_t \in \bar{\mathcal{W}}_t}.$$

In this equation, for each $\bar{w}_t \in \bar{W}_t$,

$$
\begin{aligned}
\psi_{\bar{w}_t}^\pi := {} & \frac{\mathbb{1}\{\bar{W}_t = \bar{w}_t\} \cdot (Y_t - \Gamma_{\bar{w}_t}^{(t)}(\bar{H}_t))}{\pi_{\bar{w}_t}} \\
& + A_{\bar{w}_t}^{(t)}(\bar{H}_t) + \Gamma_{\bar{w}_t}^{(1)}(X_1) - \mu_{\bar{w}_t}(t),
\end{aligned}
\tag{11}
$$

where $\mathbb{1}\{\cdot\}$ represents the indicator function. The following lemma states what moment conditions are implied by our setup with a randomized experiment.

**Lemma 1** (Moment Conditions). *Under Assumptions 1-4, we have the following moment conditions at any time $t$:*

$$\mathbb{E}[\psi_t^\pi(\bar{Z}_t; \theta_t, m_t, p_t)] = 0, \tag{12}$$

*where $\psi_{\bar{w}_t}^\pi(\bar{Z}_t; \theta_t, m_t, p_t)$ is given in Eq. (11).*

The key to valid inference with ML estimators lies in the robustness of this moment condition to local perturbations in the nuisance functions.

**Lemma 2** (Neyman orthogonality). *Let $\eta_t = \{m_t, p_t\}$ be the nuisance functions and let $\eta$ be any admissible perturbation in the same function class. Then, for any $t \in [T]$,*

$$\frac{\partial}{\partial r} \mathbb{E}\big[\psi_t^\pi(\bar{Z}_t; \theta_t, \eta_t + r(\eta - \eta_t))\big]\Big|_{r=0} = 0,$$

*where $r \in \mathbb{R}$ lies in a neighborhood containing $0$.*

Neyman orthogonality implies that the moment condition is first-order robustness of the moment condition to errors in the nuisance estimates. Leveraging this property, coupled with cross-fitting, allows us to obtain asymptotic normality of the dynamic regression-adjusted estimator under mild conditions, despite using ML models for the nuisance functions.

**Lemma 3** (Sample moment condition). *For any $t \in [T]$, the proposed dynamic regression-adjusted estimator $\widehat{\theta}_t :=$ $(\widehat{\mu}_{\bar{w}_t}^{dy\text{-}adj}(t))_{\bar{w}_t \in \bar{\mathcal{W}}_t}$, where $\widehat{\mu}_{\bar{w}_t}^{dy\text{-}adj}(t)$ is defined in Eq. (9), is uniquely obtained as the solution to the following sample moment condition:*

$$\frac{1}{n}\sum_{i=1}^{n} \psi_t^{\widehat{\pi}}(\bar{Z}_{i,t}; \widehat{\theta}_t, \widehat{m}_t, \widehat{p}_t) = 0, \tag{13}$$

*where $\widehat{m}_t$ and $\widehat{p}_t$ denote the set of nuisance functions estimated via cross-fitted ML models produced by Algorithm 1, and $\psi_t^{\widehat{\pi}}$ denotes the feasible score obtained from $\psi_{\bar{w}_t}^{\pi}$ in Eq. (11) by replacing $\pi_{\bar{w}_t}$ with its empirical analogs $\widehat{\pi}_{\bar{w}_t} := n_{\bar{w}_t}/n$.*

This lemma shows that the solution to Eq. (13) coincides with the vector whose components are the dynamic regression-adjusted estimators $\widehat{\mu}_{\bar{w}_t}^{dy\text{-}adj}(t)$ defined in Eq. (9). Lemmas 2 and 3 ensure that the estimator $\widehat{\mu}_{\bar{w}_t}^{dy\text{-}adj}(t)$ is insensitive to the estimation errors in the nuisance functions.

## 4.4. Estimation Procedure

Here, we explain our algorithm for computing the dynamic regression-adjusted estimator $\widehat{\mu}_{\bar{w}_t}^{dy\text{-}adj}(t)$ defined in Eq. (9). Algorithm 1 shows an overview of our estimation procedure. To ensure the validity of our asymptotic results and avoid overfitting biases, we employ an $L$-fold cross-fitting strategy. This procedure decouples nuisance function estimation from moment condition evaluation. The estimation proceeds in two main stages:

**Step 1. Nuisance estimation.** We randomly partition the observation indices into $L$ disjoint folds. For each fold $\ell$, we utilize the units in the remaining $L-1$ folds as the training set. On this training set, we fit the supervised learning model

---

**Algorithm 1** Dynamic regression-adjusted estimator

**Input:** (a) Observed data $\bar{Z}_{i,T} = \{(\bar{X}_{i,T}, \bar{W}_{i,T}, \bar{Y}_{i,T})\}_{i=1}^{n}$
       (b) Number of folds $L$
       (c) Learning models $\mathcal{M}, \mathcal{T}$
       (d) Monte Carlo samples $S$
**Output:** Adjusted Estimator $\{\widehat{\theta}_1, \ldots, \widehat{\theta}_T\}$
1: Create $\bar{H}_{i,t} \leftarrow (\bar{X}_{i,t}, \bar{Y}_{i,t-1})$ for all $(i,t) \in [n] \times [T]$
2: Randomly split the units into $L$ folds
3: /* Step 1. NUISANCEESTIMATION */
4: **for** $\ell \in [L], t \in [T], \bar{w}_t \in \bar{\mathcal{W}}_t$ **do**
5:    Train $\mathcal{M}$ on data with $\bar{w}_t$, excluding fold $\ell$
6:    Obtain out-of-fold predictions $\widehat{m}_{\bar{w}_t}^{(t)}(\bar{H}_{i,t})$
7: **end for**
8: **for** $\ell \in [L], \tau \in [T-1], \bar{w}_\tau \in \bar{\mathcal{W}}_\tau$ **do**
9:    Train $\mathcal{T}$ on data with $\bar{w}_\tau$, excluding fold $\ell$
10:   Obtain out-of-fold samplers $\widehat{p}_{\bar{w}_\tau}^{(\tau)}(\bar{H}_{i,\tau})$
11: **end for**
12: /* Step 2. RECURSIVEINTEGRATION */
13: **for** $t \in [T], \bar{w}_t \in \bar{\mathcal{W}}_t$ **do**
14:   Initialize $\widehat{\Gamma}_{\bar{w}_t}^{(t)}(\bar{H}_{i,t}) \leftarrow \widehat{m}_{\bar{w}_t}^{(t)}(\bar{H}_{i,t})$
15:   **for** $\tau = t-1, \ldots, 1$ **do**
16:      Compute $\widehat{\Gamma}_{\bar{w}_t}^{(\tau)}(\bar{H}_{i,\tau})$ according to Eq. (8) using Monte Carlo sampling with $S$ draws
17:   **end for**
18:   Compute $\widehat{A}_{\bar{w}_t}^{(t)}(\bar{H}_{i,t})$ according to Eq. (10)
19:   Compute $\widehat{\mu}_{\bar{w}_t}^{dy\text{-}adj}(t)$ according to Eq. (9)
20: **end for**
21: **return** $\{\widehat{\theta}_1, \ldots, \widehat{\theta}_T\}$

---

$\mathcal{M}$ to estimate the conditional mean function $\widehat{m}_{\bar{w}_t}^{(t)}$ and the transition model $\mathcal{T}$ to learn the transition kernel $\widehat{p}_{\bar{w}_\tau}^{(\tau)}$. These trained models are then applied to the units in fold $\ell$ to generate out-of-sample predictions and transition estimates.

**Step 2. Recursive integration.** We calculate the integral over the high-dimensional historical covariate space defined in Eq. (8). However, since analytical integration is often intractable, we approximate it using Monte Carlo integration. Specifically, for each unit in fold $\ell$, we draw $S$ samples from the estimated transition kernel $p_{\bar{w}_\tau}^{(\tau)}$ to recursively compute $\widehat{\Gamma}_{\bar{w}_t}^{(\tau)}$ backwards from $\tau = t-1$ down to $1$.

Lastly, these components are combined to compute the auxiliary correction term $\widehat{A}_{\bar{w}_t}^{(t)}$ and the target estimator $\mu_{\bar{w}_t}^{dy\text{-}adj}(t)$.

## 4.5. Statistical Inference

Our estimator is easy to use for statistical inference, such as standard error estimation and the construction of confidence intervals, because it is based on Neyman-orthogonal moment conditions in Lemma 2 and has an explicit influ-

ence function in Eq. (11). In this section, we briefly explain two practical inference methods for the target parameter $\theta_t = (\mu_{\bar{w}_t}(t))_{\bar{w}_t \in \bar{\mathcal{W}}_t}$ and for contrasts such as longitudinal ATEs. The asymptotic validity of these procedures is established in the next section.

**Analytical variance estimation.** The asymptotic normality result in Theorem 4 allows us to estimate standard errors using the sample variance of the estimated influence functions. Since the moment function $\psi_t^\pi$ derived in Eq. (11) coincides with the efficient influence function (Theorem 5), the asymptotic covariance matrix $\Sigma_t$ is consistently estimated by the empirical second moment of the scores:

$$\widehat{\Sigma}_t = \frac{1}{n} \sum_{i=1}^n \psi_t^{\widehat{\pi}}(\bar{Z}_{i,t}; \widehat{\theta}_t, \widehat{m}_t, \widehat{p}_t) \psi_t^{\widehat{\pi}}(\bar{Z}_{i,t}; \widehat{\theta}_t, \widehat{m}_t, \widehat{p}_t)^\top$$

Using this consistent variance estimator, a $(1 - \alpha)$ Wald-type confidence interval for a linear contrast $c^\top \theta_t$ can be constructed as

$$\left[ c^\top \widehat{\theta}_t \pm z_{1-\alpha/2} \sqrt{\frac{c^\top \widehat{\Sigma}_t c}{n}} \right],$$

where $z_{1-\alpha/2}$ denotes the $(1-\alpha/2)$-quantile of the standard normal distribution.

**Multiplier bootstrap.** While the analytical approach yields closed-form standard errors, it relies on asymptotic approximations that may be inaccurate in finite-sample settings. The multiplier bootstrap (Gine & Zinn, 1984) provides an alternative approximation that can improve finite-sample accuracy relative to the normal approximation. Importantly, in our semiparametric setting, this method avoids the computationally expensive step of refitting nuisance models for each bootstrap iteration (Chernozhukov et al., 2018). Specifically, we first draw multipliers $\{\xi_i^{(b)}\}_{i=1}^n$ for $b \in [B]$ independently of data, from a distribution with mean zero and unit variance (e.g., standard normal or Rademacher). Then, for each $t \in [T]$, we compute

$$G_t^{(b)} := \frac{1}{\sqrt{n}} \sum_{i=1}^n \xi_i^{(b)} \psi_t^{\widehat{\pi}}(\bar{Z}_{i,t}; \widehat{\theta}_t, \widehat{m}_t, \widehat{p}_t).$$

Conditional on the observed data, the empirical distribution of $\{G_t^{(b)}\}_{b=1}^B$ serves as an approximation to the law of $\sqrt{n}(\widehat{\theta}_t - \theta_t)$. A two-sided $(1 - \alpha)$ confidence interval for a scalar contrast is then given by

$$\left[ c^\top \widehat{\theta}_t - \frac{\widehat{q}_{1-\alpha/2}}{\sqrt{n}}, \ c^\top \widehat{\theta}_t - \frac{\widehat{q}_{\alpha/2}}{\sqrt{n}} \right],$$

where $\widehat{q}_\alpha$ is the $\alpha$-th empirical quantile of $\{c^\top G_t^{(b)}\}_{b=1}^B$. This bootstrap is computationally attractive in our setting because it operates only on the already computed influence function evaluations and therefore avoids repeating the training of the nuisance learners.

## 5. Asymptotic Distribution

In this section, we derive the asymptotic distribution of the proposed estimator, which enables statistical inference and the construction of confidence intervals. Additionally, we establish the semiparametric efficiency bound for our dynamic regression-adjusted estimator and demonstrate that our estimator achieves this bound under the specified assumptions. We begin by introducing additional assumptions to formalize our results.

**Assumption 5** (Finite moments of the moment function)**.**

$$\exists \delta > 0 \ \text{ s.t. } \ \sup_{t \in [T]} \mathbb{E}\Big[\big\|\psi_t^\pi(\bar{Z}_t; \theta_t, m_t, p_t)\big\|_2^{2+\delta}\Big] < \infty.$$

**Assumption 6** (Rate of convergence of cross-fitted nuisance functions)**.** *For any $t \in [T]$, the cross-fitted estimators of nuisance functions obtained by Algorithm 1 satisfy*

$$\|\widehat{m}_t - m_t\|_{P,2} + \|\widehat{p}_t - p_t\|_{P,2} = o_p(n^{-1/4}),$$

*where $\|\cdot\|_{P,2}$ denotes the $L_2$-norm with respect to the true distribution $P$ of the relevant arguments.*

**Assumption 7** (Uniform stability)**.** *For any $t \in [T]$, the construction of the forward operators $\Gamma_{\bar{w}_t}^{(\tau)}$ and the auxiliary correction term $A_{\bar{w}_t}^{(t)}$ is uniformly continuous with respect to the nuisance functions $\eta_t$ in the $L_2$-norm, admitting a modulus of continuity $\omega$.*

Assumption 5 rules out extremely heavy-tailed outcomes and allows us to apply standard limit theorems. Assumption 6 requires that the cross-fitted nuisance functions become sufficiently accurate as the sample size grows. Note that transition kernels are viewed as maps from the conditioning history to the vector space of finite signed measures, equipped with the variation norm. Assumption 7 states that small estimation errors in the nuisance functions do not get amplified when constructing $\widehat{\Gamma}_{\bar{w}_t}^{(\tau)}$ and $\widehat{A}_{\bar{w}_t}^{(t)}$.

We now establish the weak convergence of our proposed estimator in the following theorem, which serves as the theoretical foundation for statistical inference.

**Theorem 4** (Asymptotic normality)**.** *Under Assumptions 1-7, for any $t \in [T]$, the dynamic regression-adjusted estimator obtained by Algorithm 1 satisfies*

$$\sqrt{n}(\widehat{\theta}_t - \theta_t) \rightsquigarrow \mathcal{N}(0, \Sigma_t),$$

*where $\Sigma_t = \mathrm{Var}\big(\psi_t^\pi(\bar{Z}_t; \theta_t, m_t, p_t)\big).$*

Theorem 4 and Lemmas 1-3 show that the moment function $\psi_t^\pi(\bar{Z}_t)$ is the influence function. The next theorem reveals that $\psi_t^\pi(\bar{Z}_t)$ is also the efficient influence function.

**Theorem 5** (Semiparametric efficiency bound)**.** *Under Assumptions 1-5, for any $t \in [T]$, the semiparametric efficiency*

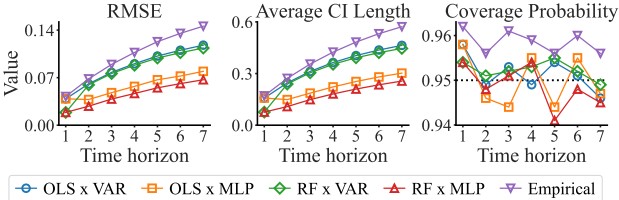

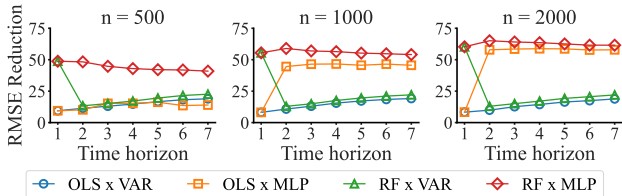

*Figure 1.* **Statistical properties of different estimators** on synthetic data with a sample size of $n = 1000$ over 1000 simulations. Any adjustment achieves smaller RMSE and average 95% CI length while maintaining the nominal coverage probability. Appendix F provides additional results for other sample sizes.

*Figure 2.* **Root mean squared error (RMSE) reduction in %** of adjusted estimators compared to empirical ones with different sample sizes $n \in \{500, 1000, 2000\}$. RF-MLP adjustment consistently achieves variance reduction across time horizons, with performance improving as sample size increases.

bound for $\theta_t$ is $\Sigma_t = \mathrm{Var}\big(\psi_t^\pi(\bar{Z}_t; \theta_t, m_t, p_t)\big)$. *Moreover, if Assumptions 6-7 also hold, the dynamic regression-adjusted estimator $\widehat{\theta}_t$ attains the semiparametric efficiency bound.*

**Corollary 6** (Variance reduction). *Under Assumptions 1-7, for any $t \in [T]$ and $\bar{w}_t \in \bar{\mathcal{W}}_t$, we have*

$$\mathrm{Var}\Big(\widehat{\mu}_{\bar{w}_t}^{emp}(t)\Big) \geq \mathrm{Var}\Big(\widetilde{\mu}_{\bar{w}_t}^{dy\text{-}adj}(t)\Big),$$

*where $\widetilde{\mu}_{\bar{w}_t}^{dy\text{-}adj}(t)$ is the dynamic regression-adjusted estimator that incorporates known adjustment terms.*

Theorem 5 and Corollary 6 demonstrate that the dynamic regression-adjusted estimator is asymptotically normal and semiparametrically efficient, reaching the efficiency bound.

## 6. Experiments

We conducted simulation study to validate our theoretical results and demonstrated the effectiveness of our estimator using real-world A/B test data collected from a large-scale streaming platform in Japan.

### 6.1. Simulation Study

**Setup.** We explain our synthetic data generating process. We generate the outcome $Y_{i,t}$ and covariate $X_{i,t}$ for the $i$-th unit at time point $t$ according to the following process:

$$\begin{aligned} Y_{i,t} &= g(X_{i,t}, Y_{i,t-1}) + \tau(t)W_{i,t} + \varepsilon_{i,t}, \\ dX_{i,t} &= Q \cdot f(Q^\top X_{i,t}, Y_{i,t-1}, W_{i,t})dt + \sigma_X dB_t, \end{aligned} \quad (14)$$

where the initial states $X_{i,1} \sim \mathcal{N}(F, \sigma_0^2 I_d)$ and $Y_{i,0} = 0$, the error terms $\varepsilon_{i,t} \sim \mathcal{N}(0, \sigma_Y^2)$, $B_t$ is a standard Wiener process representing system noise, and $Q$ is a random orthogonal matrix sampled uniformly from the Haar measure. Here, let $g(\cdot)$ be a nonlinear outcome function depending only on a subset of covariates, and let $\tau(t)$ be a time-varying treatment effect. The covariate transition $f(\cdot)$ is constructed based on the rotated forced Lorenz-96 model (Karimi & Paul, 2010), which is commonly used as a benchmark. This

function includes external time-dependent forcing $F_t$ coupled with both historical outcomes and treatment assignments, which together shape the evolution of the covariates over time. In addition, we drew a subject-level treatment indicator $W_i \sim \mathrm{Bernoulli}(0.5)$ independently across $i$ and set $W_{i,t} = W_i$ for all $t \in [T]$, which follows a standard A/B testing setup. Detailed descriptions of the data generating process are provided in Appendix E.2. This design incorporates nonlinear dependencies, irrelevant covariates, influence from past treatments, and time-varying treatment effects, so the synthetic data generated by this process reflects the complexities of real-world scenarios.

We used synthetic data of size $n \in \{500, 1000, 2000\}$ and length $T = 7$ with 10-dimensional covariates and estimated the longitudinal ATE defined by Eq. (2) with 1000 simulations. We approximated the ground-truth values using Monte Carlo sampling with $10^6$ draws. We used an OLS model (linear) and an RF model (nonlinear) as supervised learning models $\mathcal{M}$, and a VAR model (linear) and a Gaussian MLP model (nonlinear) as transition learning models $\mathcal{T}$. For instance, the estimator combining a RF model and Gaussian MLP model is referred to as RF-MLP adjustment. All adjusted estimators used 5-fold cross-fitting. Our source code for the simulation study is publicly available at: https://github.com/C-Naoki/dynamic-adjustment.

**Result.** Figure 1 shows the statistical properties (RMSE, average 95% confidence interval (CI) length, and its coverage probability) of different estimators using synthetic data with a sample size of $n = 1000$, based on 1000 simulation runs. Additional results for other sample sizes are found in Appendix F. The pointwise confidence intervals are calculated using sample estimates of the asymptotic variance. All adjusted estimators exhibit lower RMSE and shorter average 95% confidence interval lengths compared to the empirical estimator while maintaining the nominal coverage probability. In particular, RF-MLP adjustment achieves the best performance, demonstrating the benefits of incorporating machine learning techniques. OLS-MLP adjustment yields performance comparable to RF-MLP adjustment because

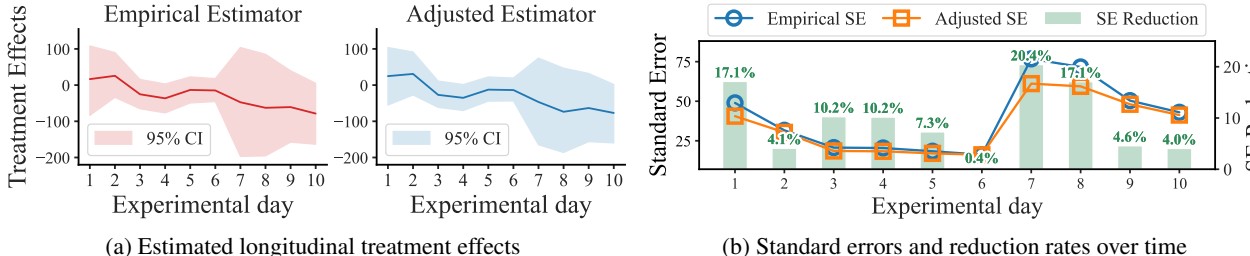

(a) Estimated longitudinal treatment effects

(b) Standard errors and reduction rates over time

*Figure 3.* **Practicality of our framework on real-world A/B test data from a large-scale streaming platform.** (a) Estimated trajectories of longitudinal treatment effects on daily user viewing time. The A/B test compares content-based recommendations (treatment) against interaction-based ones (control) over a 10-day period. Our dynamic regression adjustment yields narrower 95% pointwise confidence intervals (shaded areas) compared to the unadjusted empirical estimator. (b) Comparison of standard errors and the corresponding reduction rates. Our method reduces standard errors (SE) by approximately 0.4% to 20.4% across the experimental period, demonstrating the practical advantage of incorporating covariate transition dynamics for efficient inference.

the lagged data itself contains a certain amount of information, allowing even a simple linear baseline to capture the underlying trend to a large extent. In contrast, RF-VAR adjustment offers only a subtle improvement because a VAR model fails to capture the complex nonlinear dynamics, resulting in sample trajectories that deviate significantly from the true data distribution. Figure 2 shows RMSE reduction in % relative to the empirical estimator for different sample sizes $n \in \{500, 1000, 2000\}$. Across all sample sizes, dynamic regression adjustment consistently yields substantial precision gains with performance improving as sample size $n$ increases. These results validate the theoretical discussion presented in Corollary 6, and emphasize the value of flexible dynamic regression adjustment in improving finite-sample efficiency for longitudinal treatment effect estimation.

### 6.2. Real-world Datasets

**Setup.** In this section, the proposed dynamic regression adjustment framework is applied to evaluate the results of a longitudinal A/B test at a streaming platform in Japan, using non-public proprietary data provided by the company. The A/B test was conducted to investigate how different types of up-next content recommendations influenced user viewing time over a 10-day period. The logged data on the platform is organized as panel data, with each observation showing the daily viewing behavior during the experiment. Users were randomly assigned to one of two groups. The control group received recommendations based on user interaction data, such as past clicks, while the treatment group received recommendations based on content-level similarity derived from item features. The outcome variable is the daily viewing time for a specific series, and the covariate variables are the daily viewing times for the five series, including lagged outcome variables. To ensure reliable analysis, we selected a total of 6,802 users with sufficient observation coverage during the experimental period. For regression adjustment, we used an RF model and a zero-inflated log-normal (ZILN) MLP model based on the domain knowledge that users did

not watch the target series daily, leading to a large number of zero observations. We used 5-fold cross-fitting as well as the simulation studies.

**Result.** Figure 3 (a) visualizes the estimated trajectories of the longitudinal treatment effects on daily viewing time over the 10-day experiment. The left panel displays the results from the empirical estimator, while the right panel presents those from our proposed dynamic regression-adjusted estimator. The shaded regions represent the 95% pointwise confidence intervals computed using multiplier bootstrap (Gine & Zinn, 1984) with 1000 repetitions. We can see that the treatment effect initially showed slightly positive values but subsequently decreased. This pattern may reflect that users initially engaged with the new recommendations but did not sustain their viewing time, possibly due to a mismatch with their longitudinal preferences. Figure 3 (b) quantifies the corresponding precision gains in terms of the standard errors (SE) and the reduction rates over time. Specifically, the regression adjustment reduces the standard errors by approximately 0.4% to 20.4% across the experimental period. These empirical findings demonstrate the practical utility of incorporating covariate transition dynamics to enhance statistical power in large-scale online experiments.

## 7. Conclusion

We present a regression adjustment framework for the estimation of longitudinal treatment effects under randomized experiments. By explicitly modeling covariate dynamics via transition kernels and employing recursive forward integration, our method successfully extracts information from evolving post-treatment histories without distorting our target estimand. Moreover, we derive several theoretical foundations of our proposed estimator, including asymptotic normality (Theorem 4) and a semiparametric efficiency bound (Theorem 5). Simulation studies and empirical validations using A/B test data from a large-scale streaming platform demonstrated the practical advantages of our approach.

**Limitations.** Our approach has some limitations, which are interesting to tackle in the future. First, this paper considers experimental data under perfect compliance and no interference. Such conditions are realistic in standard A/B tests, but these assumptions may be restrictive in other contexts, such as covariate-adaptive randomization (CAR) (Tsiatis et al., 2008) and observational settings. In these scenarios, time-varying confounding and various latent regimes (Saggioro et al., 2020; Rahmani & Frossard, 2025; Chihara et al., 2025; Mameche et al., 2025) may further complicate the estimation of nuisance components. In addition, any learned components would be treated as additional nuisances, whose uncertainty is propagated to inference. Second, adapting our framework for distributional treatment effects (DTE) (Doksum, 1974; Lehmann & D'Abrera, 2006) provides richer insights into heterogeneous impacts than solely focusing on overall average effects. Lastly, addressing irregular sampling intervals and missing data through continuous-time modeling (Chen et al., 2018) may broaden empirical applicability.

## Impact Statement

The goal of this paper is to advance the field of machine learning. There are many potential societal consequences of our work. Here, we highlight key points to ensure our method works well. At first, our framework builds on the Neyman-orthogonal moment conditions, which provide first-order robustness against small errors in nuisance estimation. However, severe model misspecification can still yield unexpected or unreliable results, making the rigorous validation of nuisance functions essential. We recommend a step-by-step approach to model selection and validation. When modeling the nuisance functions $m_t$ and $p_t$, it is best to establish a stable foundation by starting with simple linear baselines. If the underlying system dynamics turn out to be complex or the linear models underperform, practitioners should then scale up to more flexible, nonlinear models. Furthermore, while the theoretical assumptions cannot be directly evaluated on real-world data, the cross-fitting procedure offers an empirical diagnostic. Practitioners should closely monitor out-of-fold predictive performance (e.g., out-of-fold RMSE). Achieving high predictive accuracy without severe overfitting on the validation folds serves as a highly reliable, practical proxy for confirming that the necessary structural assumptions are reasonably satisfied. Also, we emphasize that our work focuses on static regimes (i.e., pre-specified treatment sequences). Therefore, adaptive experiments, such as contextual bandits, fall outside the scope of our method. Applying our framework to such settings could lead to substantial bias. The recursive integration and Monte Carlo sampling used in our algorithm are often computationally intensive, particularly for long time horizons, large sample sizes, and high-dimensional covariates. If the

data is too large, our method may be impractical due to limited computational budgets. Lastly, although the proposed estimator is asymptotically normal, finite-sample bias can cause the coverage of confidence intervals to deviate from nominal levels. To improve finite-sample accuracy, the multiplier bootstrap is often an effective approach, but it still relies on empirical evaluations of the influence functions. Consequently, if the nuisance components are poorly estimated due to insufficient sample sizes, the bootstrap cannot fully correct for the resulting inference errors.

## Acknowledgments

We would like to thank the anonymous referees for their valuable and helpful comments. This work was partly supported by "Program for Leading Graduate Schools" of the Osaka University, Japan, JSPS KAKENHI Grant-in-Aid for Scientific Research Number JP25KJ1729, JP26H02499, JST CREST JPMJCR23M3, JST K Program JPMJKP25Y6, JST COI-NEXT JPMJPF2009, JST COI-NEXT JPMJPF2115, Future Social Value Co-Creation Project - Osaka University.

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

# Appendix

# A. Notation and Terminology

The main symbols we use in this paper are described in Table 1.

*Table 1.* Symbols and definitions.

| Symbol | Definition |
|---|---|
| $T$ | Time horizon length |
| $n$ | Sample size |
| $d$ | Number of covariate dimensions |
| $X_{i,t}$ | Covariates of $i$-th unit at time $t$ |
| $W_{i,t}$ | Treatment indicator of $i$-th unit at time $t$ |
| $Y_{i,t}$ | Outcome of $i$-th unit at time $t$ |
| $\{\bar{Z}_{i,t}\}_{i=1}^n$ | Observed data, i.e., $\bar{Z}_{i,t} = (\bar{X}_{i,t}, \bar{W}_{i,t}, \bar{Y}_{i,t})$ |
| $Y_{i,t}(\bar{w}_t)$ | Potential outcome for treatment group $\bar{w}_t$ of $i$-th unit at time $t$ |
| $\bar{H}_t(\bar{w}_{t-1})$ | Historical data for treatment group $\bar{w}_{t-1}$ up until time $t$ |
| $\pi_{\bar{w}_t}$ | Treatment assignment probability for treatment group $\bar{w}_t$, i.e., $\pi_{\bar{w}_t} = P(\bar{W}_t = \bar{w}_t)$ |
| $n_{\bar{w}_t}$ | Sample size in treatment group $\bar{w}_t$ |
| $m_{\bar{w}_t}^{(t)}(\bar{h}_t)$ | Conditional mean function, i.e., $m_{\bar{w}_t}^{(t)}(\bar{h}_t) = \mathbb{E}[Y_t(\bar{w}_t) \mid \bar{H}_t(\bar{w}_{t-1}) = \bar{h}_t]$ |
| $p_{\bar{w}_\tau}^{(\tau)}(\cdot \mid \bar{h}_\tau)$ | Transition kernel of $H_{\tau+1}$ given $\bar{h}_\tau$ for treatment history $\bar{w}_\tau$, i.e., $h_{\tau+1} \sim p_{\bar{w}_\tau}^{(\tau)}(\cdot \mid \bar{h}_\tau)$ |
| $\Gamma_{\bar{w}_t}^{(\tau)}(\bar{h}_\tau)$ | Iterated conditional expectation |
| $\psi_t(\bar{Z}_t)$ | Moment function |
| $A_{\bar{w}_t}^{(t)}(\bar{H}_t)$ | Auxiliary correction term |
| $\theta_t$ | Target estimand, i.e., $\theta_t = (\mu_{\bar{w}_t}(t))_{\bar{w}_t \in \bar{\mathcal{W}}_t}$ |

# B. Preliminaries

### B.1. Empirical Process Theory

We collect here standard notation from empirical process theory and $L_2(P)$-norm conventions used in the proofs. Let $(\Omega, \mathcal{F}, \mathbb{P})$ be a probability space, and let $(\mathcal{G}, \|\cdot\|_\mathcal{G})$ be a normed vector space. Let $Z$ be a generic observation taking values in a measurable space $(\mathcal{Z}, \mathcal{A})$ with law $P := \mathbb{P} \circ Z^{-1}$. For a measurable map $f : \mathcal{Z} \to \mathcal{G}$, define

$$\|f\|_{P,2} := \left( \int_\mathcal{Z} \|f(z)\|_\mathcal{G}^2 P(dz) \right)^{1/2},$$

whenever the right-hand side is finite. Equivalently, $f \in L_2(P, \mathcal{G})$ if $\|f\|_{P,2} < \infty$. The range norm $\|\cdot\|_\mathcal{G}$ is part of the definition. For instance, in the case of $\mathcal{G} = \mathbb{R}^k$, we use the Euclidean norm as the range norm, so that

$$\|f\|_{P,2} = \left( \int_\mathcal{Z} \|f(z)\|_2^2 P(dz) \right)^{1/2}.$$

For a finite collection $f = (f_j)_{j \in \mathcal{J}}$, where $f_j : \mathcal{Z}_j \to \mathcal{G}_j$ and $\mathcal{J}$ are finite, we use the product convention

$$\|f\|_{P,2} := \left( \sum_{j \in \mathcal{J}} \|f_j\|_{P_j,2}^2 \right)^{1/2},$$

where $P = (P_j)_{j \in \mathcal{J}}$. Since $\mathcal{J}$ is finite, replacing this sum with any equivalent finite product norm does not affect asymptotic rate conditions.

We next specify the convention for measure-valued maps. For two probability measures $\mathbb{P}_1$ and $\mathbb{P}_2$ on $(\Omega, \mathcal{F})$, total-variation distance is written by $d_{\mathrm{TV}}(\mathbb{P}_1, \mathbb{P}_2) = \sup_{A \in \mathcal{F}} |\mathbb{P}_1(A) - \mathbb{P}_2(A)|$. Then, for kernel-valued maps $z \mapsto K_1(\cdot \mid z)$ and

$z \mapsto K_2(\cdot \mid z)$, we have

$$\|K_1 - K_2\|_{P,2} := \left( \int_{\mathcal{Z}} d_{\mathrm{TV}}(K_1(\cdot \mid z), K_2(\cdot \mid z))^2 P(dz) \right)^{1/2}.$$

Let $(Z_i)_{i=1}^n$ be i.i.d. copies of $Z$. For any measurable scalar function $f : \mathcal{Z} \to \mathbb{R}$, we write

$$Pf := \mathbb{E}[f(Z)] = \int_{\mathcal{Z}} f(z) P(dz) = \int_{\Omega} f(Z(\omega)) \mathbb{P}(d\omega), \quad P_n f := \frac{1}{n} \sum_{i=1}^n f(Z_i),$$

and define the empirical process $\mathbb{G}_n$ by $\mathbb{G}_n f := \sqrt{n}(P_n - P)f$ and $\|f\|_{P,2} := (Pf^2)^{1/2}$ for the $L_2$-norm of $f$.

The next lemma provides a simple variance bound for a single empirical process and will be used in Theorem 4.

**Lemma 7** (Conditional variance bound for a single empirical process). *Let $f : \mathcal{Z} \to \mathbb{R}$ be any measurable function and satisfy $\|f\|_{P,2}^2 < \infty$, where $\|f\|_{P,2} = (Pf^2)^{1/2}$. Then*

$$\mathbb{E}\big[(\mathbb{G}_n f)^2\big] \le \|f\|_{P,2}^2 \quad a.s.$$

*Proof.* By definition of the empirical process $\mathbb{G}_n$,

$$
\begin{aligned}
\mathbb{E}\big[(\mathbb{G}_n f)^2\big] = \mathbb{E}\big[(\sqrt{n}(P_n - P)f)^2\big] &= \mathbb{E}\left[ \left( \frac{1}{\sqrt{n}} \sum_{i=1}^n (f(Z_i) - Pf) \right)^2 \right] \\
&= \frac{1}{n} \sum_{i=1}^n \mathbb{E}\big[(f(Z_i) - Pf)^2\big] + \frac{1}{n} \sum_{i \ne j} \mathbb{E}[(f(Z_i) - Pf)(f(Z_j) - Pf)] \\
&= \frac{1}{n} \sum_{i=1}^n \mathbb{E}\big[(f(Z_i) - Pf)^2\big] + \frac{1}{n} \sum_{i \ne j} \mathbb{E}[(f(Z_i) - Pf)]\, \mathbb{E}[(f(Z_j) - Pf)] \\
&= \frac{1}{n} \sum_{i=1}^n \mathbb{E}\big[(f(Z_i) - Pf)^2\big] = \mathbb{E}\big[(f(Z) - Pf)^2\big] = \mathrm{Var}(f(Z)) \\
&\le \mathbb{E}\big[f^2(Z)\big] = \|f\|_{P,2}^2
\end{aligned}
$$

Hence, the desired result is obtained. $\square$

## B.2. Semiparametric Efficiency Theory

In this section, we provide a brief review of semiparametric theory, focusing on the definitions of the influence function and semiparametric efficiency. We consider a generic observation $Z$ taking values in a measurable space $(\mathcal{Z}, \mathcal{A})$ and let $\mathcal{P}$ be a set of all probability measures on $\mathcal{Z}$. We specify a statistical model $\mathcal{M} \subset \mathcal{P}$, which is a (possibly infinite-dimensional) collection of candidate distributions. We assume that the model $\mathcal{M}$ is dominated by a common $\sigma$-finite measure $\mu$ (e.g., the Lebesgue measure), such that every distribution $P \in \mathcal{M}$ admits a probability density function $p = dP/d\mu$. We assume that the data generating process follows a distribution $P \in \mathcal{M}$.

In semiparametric theory, we are interested in a low-dimensional parameter $\theta \in \mathbb{R}$ (e.g., mean, variance, and average treatment effect), while leaving the rest of the data generating mechanism unrestricted and treating them as nuisance components. A convenient way to formalize this idea is to introduce a *statistical functional* $T : \mathcal{M} \to \mathbb{R}$, which maps each data distribution $P \in \mathcal{M}$ to the corresponding target parameter $\theta$, i.e., $\theta = T(P)$. Correspondingly, an estimator $\widehat{\theta}_n$ is often conceptually motivated by the plug-in estimator $\widehat{\theta}_n = T(P_n)$, assuming the domain of $T$ can be extended to include the empirical distribution $P_n$. While this provides an intuitive baseline, rigorous estimation requires analyzing how sensitive the target parameter is to local perturbations within the model $\mathcal{M}$.

To describe this strictly, we introduce the geometric structure of the model. For any $P \in \mathcal{M}$, let $L^2(P)$ denote the Hilbert space of square-integrable functions with respect to $P$, equipped with the inner product $\langle f, g \rangle_{L^2(P)} = \mathbb{E}[f(Z)g(Z)]$. We further define the subspace of zero-mean functions as $L_0^2(P) := \{ f \in L^2(P) : \mathbb{E}[f(Z)] = 0 \}$. Based on these spaces, we provide the following definitions.

**Definition 1** (Tangent space). *Let $\mathcal{M}$ be the statistical model and let $P \in \mathcal{M}$ be any distribution. The **tangent space** $\mathcal{T}(P) \subset L_0^2(P)$ of the model at $P$ is defined as the closed linear span of all score functions $S(Z) = \partial_\alpha \log p_\alpha(Z)\big|_{\alpha=0}$ associated with regular parametric submodels[1] $\{P_\alpha : \alpha \in (-\varepsilon, \varepsilon)\} \subset \mathcal{M}$ passing through $P$.*

**Definition 2** (Pathwise differentiability). *Let $T : \mathcal{M} \to \mathbb{R}$ be a statistical functional and let $P$ be the true data distribution. Let $\mathcal{T}(P) \subset L_0^2(P)$ denote the tangent space of the model at $P$. The functional $T$ is said to be **pathwise differentiable** at $P$ relative to the tangent space $\mathcal{T}(P)$ if there exists a continuous linear functional $T_P : \mathcal{T}(P) \to \mathbb{R}$ such that for every regular parametric submodel $\{P_\alpha : \alpha \in (-\varepsilon, \varepsilon)\} \subset \mathcal{M}$ passing through $P$ at $\alpha = 0$ with score function $S(Z)$, the following equation holds $\partial_\alpha T(P_\alpha)\big|_{\alpha=0} = T_P(S)$. Here, the linear map $T_P$ is called the pathwise derivative of $T$ at $P$.*

**Definition 3** (Gradient). *Let $T : \mathcal{M} \to \mathbb{R}$ be a statistical functional and let $P$ be the true distribution. Let $\mathcal{T}(P) \subset L_0^2(P)$ denote the tangent space of the model at $P$. A function $\phi(\cdot; P) \in L_0^2(P)$ is referred to as a **gradient** of $T$ at $P$ if it represents the pathwise derivative $T_P$ via the inner product in $L^2(P)$. That is, for every regular parametric submodel $\{P_\alpha : \alpha \in (-\varepsilon, \varepsilon)\} \subset \mathcal{M}$ passing through $P$ with score function $S \in \mathcal{T}(P)$, we have $\partial_\alpha T(P_\alpha)\big|_{\alpha=0} = \mathbb{E}[\phi(Z; P)S(Z)]$.*

**Definition 4** (Influence function). *Consider an estimator $\widehat{\theta}_n$ for $\theta = T(P)$. The estimator is said to be asymptotically linear if it satisfies the following expansion:*

$$\sqrt{n}(\widehat{\theta}_n - \theta) = \frac{1}{\sqrt{n}} \sum_{i=1}^{n} \psi(Z_i; P) + o_p(1),$$

*where $\psi(\cdot; P) \in L_0^2(P)$ is a zero-mean square-integrable function. The function $\psi$ is called the **influence function** of the estimator $\widehat{\theta}_n$. For any regular asymptotically linear estimator, its influence function $\psi$ must be a gradient as defined in Definition 3.*

**Definition 5** (Efficient influence function). *Among all possible gradients for the functional $T$ at $P$, the **efficient influence function** (or canonical gradient), denoted by $\tilde{\phi}(\cdot; P)$, is the unique gradient that lies in the tangent space $\mathcal{T}(P)$; that is, $\tilde{\phi} \in \mathcal{T}(P)$. The estimator whose influence function equals the efficient influence function achieves the minimum asymptotic variance among all regular estimators (semiparametric efficiency bound).*

Intuitively, the score function $S(Z)$ is the sensitivity of the log-likelihood to infinitesimal changes in the parameter $\alpha$ and the pathwise derivative $T_P$ is the sensitivity of the target parameter to changes in the distribution. Then, the influence function quantifies the effect of an infinitesimal contamination at point $z$ on the estimate of the target parameter. The efficient influence function is unique (almost surely) and equals the orthogonal projection of any gradient $\phi(\cdot; P)$ onto the tangent space $\mathcal{T}(P)$ (equivalently, any other gradient differs from $\tilde{\phi}$ by an element of $\mathcal{T}(P)^\perp$).

In the context of our paper, the moment function $\psi_t^\pi(\bar{Z}_t; \theta_t, m_t, p_t)$ derived in Eq. (11) is equivalent to the efficient influence function for our target parameter $\theta_t$ (with the above definitions understood component-wise for the finite-dimensional vector $\theta_t$).

## C. Proofs

We provide proofs of main results and technical lemmas.

### C.1. Proof of Lemma 1

**Lemma 1** (Moment Conditions). *Under Assumptions 1-4, we have the following moment conditions at any time $t$:*

$$\mathbb{E}[\psi_t^\pi(\bar{Z}_t; \theta_t, m_t, p_t)] = 0, \tag{12}$$

*where $\psi_{\bar{w}_t}^\pi(\bar{Z}_t; \theta_t, m_t, p_t)$ is given in Eq. (11).*

*Proof.* It is sufficient to derive that $\mathbb{E}\big[\psi_{\bar{w}_t}^\pi(\bar{Z}_t; \theta_t, m_t, p_t)\big] = 0$ for each $\bar{w}_t$. We decompose it via the linearity of expectation.

$$\mathbb{E}\big[\psi_{\bar{w}_t}^\pi(\bar{Z}_t; \theta_t, m_t, p_t)\big] = \underbrace{\mathbb{E}\left[\frac{\mathbb{1}\{\bar{W}_t = \bar{w}_t\} \cdot (Y_t - \Gamma_{\bar{w}_t}^{(t)}(\bar{H}_t))}{\pi_{\bar{w}_t}}\right]}_{\text{(I)}} + \underbrace{\mathbb{E}\Big[A_{\bar{w}_t}^{(t)}(\bar{H}_t; m_{\bar{w}_t}, \{p_{\bar{w}_\tau}\}_{\tau=1}^{t-1})\Big]}_{\text{(II)}} + \underbrace{\mathbb{E}\Big[\Gamma_{\bar{w}_t}^{(1)}(X_1) - \mu_{\bar{w}_t}(t)\Big]}_{\text{(III)}}$$

---

[1] Regular parametric submodels satisfy the differentiable in quadratic mean (DQM) condition (van der Vaart, 1998).

First term (I):

$$
\mathbb{E}\left[\frac{\mathbb{1}\{\bar{W}_t = \bar{w}_t\} \cdot (Y_t - \Gamma^{(t)}_{\bar{w}_t}(\bar{H}_t))}{\pi_{\bar{w}_t}}\right] = \mathbb{E}\left[\frac{\mathbb{1}\{\bar{W}_t = \bar{w}_t\} \cdot (Y_t(\bar{w}_t) - \Gamma^{(t)}_{\bar{w}_t}(\bar{H}_t))}{\pi_{\bar{w}_t}}\right]
$$

$$
= \mathbb{E}\left[\mathbb{E}\left[\frac{\mathbb{1}\{\bar{W}_t = \bar{w}_t\} \cdot (Y_t(\bar{w}_t) - \Gamma^{(t)}_{\bar{w}_t}(\bar{H}_t))}{\pi_{\bar{w}_t}} \,\Bigg|\, \bar{H}_t, \bar{W}_t\right]\right]
$$

$$
= \mathbb{E}\left[\frac{\mathbb{1}\{\bar{W}_t = \bar{w}_t\}}{\pi_{\bar{w}_t}} \cdot \left\{\mathbb{E}\left[Y_t(\bar{w}_t)\mid \bar{H}_t, \bar{W}_t = \bar{w}_t\right] - \Gamma^{(t)}_{\bar{w}_t}(\bar{H}_t)\right\}\right]
$$

$$
= \mathbb{E}\left[\frac{\mathbb{1}\{\bar{W}_t = \bar{w}_t\}}{\pi_{\bar{w}_t}} \cdot \left\{\mathbb{E}\left[Y_t(\bar{w}_t)\mid \bar{H}_t(\bar{w}_{t-1}), \bar{W}_t = \bar{w}_t\right] - m^{(t)}_{\bar{w}_t}(\bar{H}_t(\bar{w}_{t-1}))\right\}\right]
$$

$$
= \mathbb{E}\left[\frac{\mathbb{1}\{\bar{W}_t = \bar{w}_t\}}{\pi_{\bar{w}_t}} \cdot \left\{\mathbb{E}\left[Y_t(\bar{w}_t)\mid \bar{H}_t(\bar{w}_{t-1})\right] - m^{(t)}_{\bar{w}_t}(\bar{H}_t(\bar{w}_{t-1}))\right\}\right]
$$

$$
= 0
$$

Second term (II):

$$
\mathbb{E}\left[A^{(t)}_{\bar{w}_t}\right] = \mathbb{E}\left[\sum_{\tau=1}^{t-1} \frac{\mathbb{1}\{\bar{W}_\tau = \bar{w}_\tau\}}{\pi_{\bar{w}_\tau}}\left\{\Gamma^{\tau+1}_{\bar{w}_t}(\bar{H}_{\tau+1}) - \int_{\mathcal{H}_{\tau+1}} \Gamma^{\tau+1}_{\bar{w}_t}(\bar{H}_\tau, h_{\tau+1}) p^{(\tau)}_{\bar{w}_\tau}(dh_{\tau+1}\mid \bar{H}_\tau)\right\}\right]
$$

$$
= \sum_{\tau=1}^{t-1} \mathbb{E}\left[\frac{\mathbb{1}\{\bar{W}_\tau = \bar{w}_\tau\}}{\pi_{\bar{w}_\tau}}\left\{\Gamma^{\tau+1}_{\bar{w}_t}(\bar{H}_{\tau+1}) - \mathbb{E}\left[\Gamma^{\tau+1}_{\bar{w}_t}(\bar{H}_{\tau+1})\mid \bar{H}_\tau, \bar{W}_\tau = \bar{w}_\tau\right]\right\}\right]
$$

$$
= \sum_{\tau=1}^{t-1} \mathbb{E}\left[\mathbb{E}\left[\frac{\mathbb{1}\{\bar{W}_\tau = \bar{w}_\tau\}}{\pi_{\bar{w}_\tau}}\left\{\Gamma^{\tau+1}_{\bar{w}_t}(\bar{H}_{\tau+1}) - \mathbb{E}\left[\Gamma^{\tau+1}_{\bar{w}_t}(\bar{H}_{\tau+1})\mid \bar{H}_\tau, \bar{W}_\tau = \bar{w}_\tau\right]\right\}\,\Bigg|\, \bar{H}_\tau, \bar{W}_\tau\right]\right]
$$

$$
= \sum_{\tau=1}^{t-1} \mathbb{E}\left[\frac{\mathbb{1}\{\bar{W}_\tau = \bar{w}_\tau\}}{\pi_{\bar{w}_\tau}}\left\{\mathbb{E}\left[\Gamma^{\tau+1}_{\bar{w}_t}(\bar{H}_{\tau+1})\mid \bar{H}_\tau, \bar{W}_\tau = \bar{w}_\tau\right] - \mathbb{E}\left[\Gamma^{\tau+1}_{\bar{w}_t}(\bar{H}_{\tau+1})\mid \bar{H}_\tau, \bar{W}_\tau = \bar{w}_\tau\right]\right\}\right]
$$

$$
= 0
$$

Third term (III):

$$
\mathbb{E}\left[\Gamma^{(1)}_{\bar{w}_t}(X_1) - \mu_{\bar{w}_t}(t)\right] = \mathbb{E}\left[\Gamma^{(1)}_{\bar{w}_t}(X_1)\right] - \mu_{\bar{w}_t}(t) = 0
$$

Hence, the desired result is obtained because we have (I) + (II) + (III) = 0. $\qquad\square$

### C.2. Proof of Lemma 2

**Lemma 2** (Neyman orthogonality). *Let $\eta_t = \{m_t, p_t\}$ be the nuisance functions and let $\eta$ be any admissible perturbation in the same function class. Then, for any $t \in [T]$,*

$$
\frac{\partial}{\partial r}\mathbb{E}\left[\psi^\pi_t(\bar{Z}_t; \theta_t, \eta_t + r(\eta - \eta_t))\right]\bigg|_{r=0} = 0,
$$

*where $r \in \mathbb{R}$ lies in a neighborhood containing 0.*

*Proof.* Fix $t \in [T]$ and $\bar{w}_t \in \bar{\mathcal{W}}_t$. It suffices to prove the claim component-wise for $\psi^\pi_{\bar{w}_t}$. Let $r \mapsto \eta_r = (m_r, p_r)$ be an admissible path in the nuisance space, defined in a neighborhood of 0, with $\eta_0 = \eta_t$. Define

$$
\Phi_{\bar{w}_t}(r) := \mathbb{E}\left[\psi^\pi_{\bar{w}_t}(\bar{Z}_t; \theta_t, m_r, p_r)\right].
$$

For each $r$, let $\Gamma^{(\tau)}_{\bar{w}_t,r}$ and $A^{(t)}_{\bar{w}_t,r}$ be the quantities defined in Eq. (8) and Eq. (10), respectively, with $(m^{(t)}_{\bar{w}_t}, \{p^{(\tau)}_{\bar{w}_\tau}\}^{t-1}_{\tau=1})$ replaced by $(m^{(t)}_{\bar{w}_t,r}, \{p^{(\tau)}_{\bar{w}_\tau,r}\}^{t-1}_{\tau=1})$. In particular,

$$
\Gamma^{(t)}_{\bar{w}_t,r}(\bar{h}_t) = m^{(t)}_{\bar{w}_t,r}(\bar{h}_t),
$$

and, recursively for $\tau = t - 1, \ldots, 1$,

$$\Gamma_{\bar{w}_t, r}^{(\tau)}(\bar{h}_\tau) = \int_{\mathcal{H}_{\tau+1}} \Gamma_{\bar{w}_t, r}^{(\tau+1)}(\bar{h}_\tau, h_{\tau+1}) p_{\bar{w}_\tau, r}^{(\tau)}(dh_{\tau+1} \mid \bar{h}_\tau).$$

The target parameter $\mu_{\bar{w}_t}(t)$ and the assignment probabilities $\pi_{\bar{w}_t}$ do not depend on $r$.

We first record the inverse-probability identity used below. By consistency and randomization, for any integrable measurable function $f$ on $\bar{\mathcal{H}}_{\tau+1}$,

$$\mathbb{E}\left[\frac{\mathbb{1}\{\bar{W}_\tau = \bar{w}_\tau\}}{\pi_{\bar{w}_\tau}} f(\bar{H}_{\tau+1})\right] = \int_{\bar{\mathcal{H}}_{\tau+1}} f(\bar{h}_{\tau+1}) P_{\bar{H}_{\tau+1}}^{\bar{w}_\tau}(d\bar{h}_{\tau+1}),$$

and, for any integrable measurable function $g$ on $\bar{\mathcal{H}}_\tau$,

$$\mathbb{E}\left[\frac{\mathbb{1}\{\bar{W}_\tau = \bar{w}_\tau\}}{\pi_{\bar{w}_\tau}} g(\bar{H}_\tau)\right] = \int_{\bar{\mathcal{H}}_\tau} g(\bar{h}_\tau) P_{\bar{H}_\tau}^{\bar{w}_{\tau-1}}(d\bar{h}_\tau),$$

where $P_{\bar{H}_1}^{\bar{w}_0}$ is understood as the marginal law $P_{H_1}$ of the baseline history. These identities follow, for example, from

$$\mathbb{E}\left[\frac{\mathbb{1}\{\bar{W}_\tau = \bar{w}_\tau\}}{\pi_{\bar{w}_\tau}} f(\bar{H}_{\tau+1})\right] = \mathbb{E}\left[\frac{\mathbb{1}\{\bar{W}_\tau = \bar{w}_\tau\}}{\pi_{\bar{w}_\tau}} f(\bar{H}_{\tau+1}(\bar{w}_\tau))\right]$$
$$= \mathbb{E}\left[f(\bar{H}_{\tau+1}(\bar{w}_\tau))\right],$$

and the second identity is shown analogously. The same consistency and randomization argument applied to the outcome term gives

$$\mathbb{E}\left[\frac{\mathbb{1}\{\bar{W}_t = \bar{w}_t\}}{\pi_{\bar{w}_t}} Y_t\right] = \mu_{\bar{w}_t}(t),$$

and applying the inverse-probability identity to the adjustment term gives

$$\mathbb{E}\left[\frac{\mathbb{1}\{\bar{W}_t = \bar{w}_t\}}{\pi_{\bar{w}_t}} \Gamma_{\bar{w}_t, r}^{(t)}(\bar{H}_t)\right] = \int_{\bar{\mathcal{H}}_t} \Gamma_{\bar{w}_t, r}^{(t)}(\bar{h}_t) P_{\bar{H}_t}^{\bar{w}_{t-1}}(d\bar{h}_t).$$

Next, by the same identity and by the definition of $A_{\bar{w}_t, r}^{(t)}$,

$$\mathbb{E}\left[A_{\bar{w}_t, r}^{(t)}\right] = \sum_{\tau=1}^{t-1}\left[\int_{\bar{\mathcal{H}}_{\tau+1}} \Gamma_{\bar{w}_t, r}^{(\tau+1)}(\bar{h}_{\tau+1}) P_{\bar{H}_{\tau+1}}^{\bar{w}_\tau}(d\bar{h}_{\tau+1})\right.$$
$$\left. - \int_{\bar{\mathcal{H}}_\tau}\left\{\int_{\mathcal{H}_{\tau+1}} \Gamma_{\bar{w}_t, r}^{(\tau+1)}(\bar{h}_\tau, h_{\tau+1}) p_{\bar{w}_\tau, r}^{(\tau)}(dh_{\tau+1} \mid \bar{h}_\tau)\right\} P_{\bar{H}_\tau}^{\bar{w}_{\tau-1}}(d\bar{h}_\tau)\right].$$

By the recursive definition of $\Gamma_{\bar{w}_t, r}^{(\tau)}$, the inner integral in the second term equals $\Gamma_{\bar{w}_t, r}^{(\tau)}(\bar{h}_\tau)$. Hence,

$$\mathbb{E}\left[A_{\bar{w}_t, r}^{(t)}\right] = \sum_{\tau=1}^{t-1}\left[\int_{\bar{\mathcal{H}}_{\tau+1}} \Gamma_{\bar{w}_t, r}^{(\tau+1)}(\bar{h}_{\tau+1}) P_{\bar{H}_{\tau+1}}^{\bar{w}_\tau}(d\bar{h}_{\tau+1}) - \int_{\bar{\mathcal{H}}_\tau} \Gamma_{\bar{w}_t, r}^{(\tau)}(\bar{h}_\tau) P_{\bar{H}_\tau}^{\bar{w}_{\tau-1}}(d\bar{h}_\tau)\right].$$

Finally,

$$\mathbb{E}\left[\Gamma_{\bar{w}_t, r}^{(1)}(X_1)\right] = \int_{\mathcal{H}_1} \Gamma_{\bar{w}_t, r}^{(1)}(h_1) P_{H_1}(dh_1).$$

Combining the preceding displays, we obtain

$$\Phi_{\bar{w}_t}(r) = \mu_{\bar{w}_t}(t) - \int_{\bar{\mathcal{H}}_t} \Gamma_{\bar{w}_t, r}^{(t)}(\bar{h}_t) P_{\bar{H}_t}^{\bar{w}_{t-1}}(d\bar{h}_t)$$

$$+ \sum_{\tau=1}^{t-1} \left[ \int_{\bar{\mathcal{H}}_{\tau+1}} \Gamma_{\bar{w}_t, r}^{(\tau+1)}(\bar{h}_{\tau+1}) P_{\bar{H}_{\tau+1}}^{\bar{w}_\tau}(d\bar{h}_{\tau+1}) - \int_{\bar{\mathcal{H}}_\tau} \Gamma_{\bar{w}_t, r}^{(\tau)}(\bar{h}_\tau) P_{\bar{H}_\tau}^{\bar{w}_{\tau-1}}(d\bar{h}_\tau) \right]$$

$$+ \int_{\mathcal{H}_1} \Gamma_{\bar{w}_t, r}^{(1)}(h_1) P_{H_1}(dh_1) - \mu_{\bar{w}_t}(t)$$

$$= 0.$$

The last equality follows by exact telescoping of the finite sum. Therefore, $\Phi_{\bar{w}_t}(r) = 0$ for every admissible $r$ in a neighborhood of zero. In particular,

$$\frac{\partial}{\partial r} \mathbb{E}\left[\psi_{\bar{w}_t}^\pi(\bar{Z}_t; \theta_t, m_r, p_r)\right]\bigg|_{r=0} = \Phi_{\bar{w}_t}'(0) = 0.$$

Since the argument holds for every $\bar{w}_t \in \bar{\mathcal{W}}_t$, the vector-valued moment function also satisfies

$$\frac{\partial}{\partial r} \mathbb{E}\left[\psi_t^\pi(\bar{Z}_t; \theta_t, m_r, p_r)\right]\bigg|_{r=0} = 0.$$

The desired result is obtained. □

## C.3. Proof of Lemma 3

**Lemma 3** (Sample moment condition). *For any $t \in [T]$, the proposed dynamic regression-adjusted estimator $\widehat{\theta}_t :=$ $(\widehat{\mu}_{\bar{w}_t}^{dy\text{-}adj}(t))_{\bar{w}_t \in \bar{\mathcal{W}}_t}$, where $\widehat{\mu}_{\bar{w}_t}^{dy\text{-}adj}(t)$ is defined in Eq. (9), is uniquely obtained as the solution to the following sample moment condition:*

$$\frac{1}{n} \sum_{i=1}^n \psi_t^{\widehat{\pi}}(\bar{Z}_{i,t}; \widehat{\theta}_t, \widehat{m}_t, \widehat{p}_t) = 0, \tag{13}$$

*where $\widehat{m}_t$ and $\widehat{p}_t$ denote the set of nuisance functions estimated via cross-fitted ML models produced by Algorithm 1, and $\psi_t^{\widehat{\pi}}$ denotes the feasible score obtained from $\psi_{\bar{w}_t}^\pi$ in Eq. (11) by replacing $\pi_{\bar{w}_t}$ with its empirical analogs $\widehat{\pi}_{\bar{w}_t} := n_{\bar{w}_t}/n$.*

*Proof.* For any $t \in [T]$ and $\bar{w}_t \in \bar{\mathcal{W}}_t$ with $n_{\bar{w}_t} > 0$, we substitute the estimators $\widehat{m}_{\bar{w}_t}^{(t)}$ for $m_{\bar{w}_t}^{(t)}$, $\widehat{p}_{\bar{w}_\tau}^{(\tau)}$ for $p_{\bar{w}_\tau}^{(\tau)}$, and $n_{\bar{w}_t}/n = \widehat{\pi}_{\bar{w}_t}$ in this empirical moment function, and evaluate it using the empirical probability measure.

$$P_n \psi_{\bar{w}_t}^{\widehat{\pi}} = \frac{1}{n} \sum_{i=1}^n \psi_{\bar{w}_t}^{\widehat{\pi}}(\bar{Z}_{i,t}; \widehat{\theta}_t, \widehat{m}_t, \widehat{p}_t)$$

$$= \frac{1}{n} \sum_{i=1}^n \left[ \frac{\mathbb{1}\{\bar{W}_{i,t} = \bar{w}_t\} \cdot (Y_{i,t} - \widehat{\Gamma}_{\bar{w}_t}^{(t)}(\bar{H}_{i,t}))}{\widehat{\pi}_{\bar{w}_t}} + \widehat{A}_{\bar{w}_t}^{(t)}(\bar{H}_{i,t}) + \widehat{\Gamma}_{\bar{w}_t}^{(1)}(X_{i,1}) \right] - \widehat{\mu}_{\bar{w}_t}^{dy\text{-}adj}(t)$$

$$= \frac{1}{n_{\bar{w}_t}} \sum_{i:\bar{W}_{i,t}=\bar{w}_t} (Y_{i,t} - \widehat{\Gamma}_{\bar{w}_t}^{(t)}(\bar{H}_{i,t})) + \frac{1}{n} \sum_{i=1}^n \left[ \widehat{A}_{\bar{w}_t}^{(t)}(\bar{H}_{i,t}) + \widehat{\Gamma}_{\bar{w}_t}^{(1)}(X_{i,1}) \right] - \widehat{\mu}_{\bar{w}_t}^{dy\text{-}adj}(t).$$

Imposing $P_n \psi_{\bar{w}_t}^{\widehat{\pi}} = 0$, we have a unique solution:

$$\widehat{\mu}_{\bar{w}_t}^{dy\text{-}adj}(t) = \frac{1}{n_{\bar{w}_t}} \sum_{i:\bar{W}_{i,t}=\bar{w}_t} (Y_{i,t} - \widehat{\Gamma}_{\bar{w}_t}^{(t)}(\bar{H}_{i,t})) + \frac{1}{n} \sum_{i=1}^n \left[ \widehat{A}_{\bar{w}_t}^{(t)}(\bar{H}_{i,t}) + \widehat{\Gamma}_{\bar{w}_t}^{(1)}(X_{i,1}) \right],$$

which is equivalent to the dynamic regression-adjusted estimator defined in Eq. (9). □

## C.4. Proof of Theorem 4

**Theorem 4** (Asymptotic normality). *Under Assumptions 1-7, for any $t \in [T]$, the dynamic regression-adjusted estimator obtained by Algorithm 1 satisfies*

$$\sqrt{n}(\widehat{\theta}_t - \theta_t) \rightsquigarrow \mathcal{N}(0, \Sigma_t),$$

*where $\Sigma_t = \text{Var}\left(\psi_t^\pi(\bar{Z}_t; \theta_t, m_t, p_t)\right)$.*

*Proof.* Throughout this proof, we consider the exact recursive-integration version of Algorithm 1, where the recursive integrals in Eq. (8) are evaluated exactly. The feasible score $\psi^{\widehat{\pi}}$ is understood as the score obtained by replacing every treatment-path probability appearing in $\psi^{\pi}_{\bar{w}_t}$, including the prefix probabilities in the auxiliary correction term, by its empirical counterpart. All arguments below are on the event that the empirical probabilities appearing in the proof are positive; under Assumptions 2 and 4 and fixed feasible treatment histories, this event has probability tending to one. Recalling that $\theta_t = (\mu_{\bar{w}_t}(t))_{\bar{w}_t \in \mathcal{W}_t}$, it suffices to derive an asymptotic distribution of each component $\widehat{\mu}^{dy\text{-}adj}_{\bar{w}_t}(t) - \mu_{\bar{w}_t}(t)$ and stack these results over $\bar{w}_t \in \mathcal{W}_t$. Our proof follows the four steps below:

1. We derive an asymptotic linear representation of the dynamic regression-adjusted estimator, showing that for each treatment path $\bar{w}_t$, we express $\sqrt{n}\big(\widehat{\mu}^{dy\text{-}adj}_{\bar{w}_t}(t) - \mu_{\bar{w}_t}(t)\big)$ as an average of the score plus two remainder terms.

2. We then show that the empirical process remainder term is $o_p(1)$ by combining the stability properties of the forward operator with the convergence rates of the nuisance estimators and standard moment bounds for the empirical process.

3. Next, we exploit the Neyman orthogonality (Lemma 2) of the score and the Lipschitz continuity of the moment map in the nuisance functions to prove that the bias remainder is $o_p(1)$.

4. Lastly, since the score has mean zero (Lemma 1) and finite $(2 + \delta)$-moments (Assumption 5), the multivariate central limit theorem applied to the stacked scores $\psi^{\pi}_t$ yields the asserted asymptotic normality of $\sqrt{n}(\widehat{\theta}_t - \theta_t)$.

**Step 1. Linear representation of our estimator $\widehat{\mu}^{dy\text{-}adj}_{\bar{w}_t}(t)$.** From Lemma 3, our estimator $\widehat{\theta}_t = (\widehat{\mu}^{dy\text{-}adj}_{\bar{w}_t}(t))_{\bar{w}_t \in \mathcal{W}_t}$ is the solution to the following equation

$$\frac{1}{n}\sum_{i=1}^n \psi^{\widehat{\pi}}_t(\bar{Z}_{i,t}; \widehat{\theta}_t, \widehat{m}_t, \widehat{p}_t) = P_n \psi^{\widehat{\pi}}_t(\bar{Z}_{i,t}; \widehat{\theta}_t, \widehat{m}_t, \widehat{p}_t) = 0$$

Equivalently, for each longitudinal path $\bar{w}_t \in \bar{\mathcal{W}}_t$,

$$P_n \psi^{\widehat{\pi}}_{\bar{w}_t}(\bar{Z}_{i,t}; \widehat{\theta}_t, \widehat{m}_t, \widehat{p}_t) = 0.$$

We decompose the above formula as follows:

$$P_n \psi^{\widehat{\pi}}_{\bar{w}_t}(\bar{Z}_{i,t}; \widehat{\theta}_t, \widehat{m}_t, \widehat{p}_t) = P_n \psi^{\pi}_{\bar{w}_t}(\bar{Z}_{i,t}; \theta_t, m_t, p_t) + P_n \Delta\psi_{\bar{w}_t}(\bar{Z}_t) + P_n\{\psi^{\widehat{\pi}}_{\bar{w}_t}(\bar{Z}_{i,t}; \widehat{\theta}_t, \widehat{m}_t, \widehat{p}_t) - \psi^{\widehat{\pi}}_{\bar{w}_t}(\bar{Z}_{i,t}; \theta_t, \widehat{m}_t, \widehat{p}_t)\},$$

where $\Delta\psi_{\bar{w}_t}(\bar{Z}_t) = \psi^{\widehat{\pi}}_{\bar{w}_t}(\bar{Z}_{i,t}; \theta_t, \widehat{m}_t, \widehat{p}_t) - \psi^{\pi}_{\bar{w}_t}(\bar{Z}_{i,t}; \theta_t, m_t, p_t)$ represents the perturbation term induced by replacing the true nuisance functions and treatment-path probabilities with their estimates. Next, we consider the third term of the RHS.

$$\psi^{\widehat{\pi}}_{\bar{w}_t}(\bar{Z}_{i,t}; \widehat{\theta}_t, \widehat{m}_t, \widehat{p}_t) - \psi^{\widehat{\pi}}_{\bar{w}_t}(\bar{Z}_{i,t}; \theta_t, \widehat{m}_t, \widehat{p}_t) = -(\widehat{\mu}^{dy\text{-}adj}_{\bar{w}_t}(t) - \mu_{\bar{w}_t}(t))$$
$$P_n\{\psi^{\widehat{\pi}}_{\bar{w}_t}(\bar{Z}_{i,t}; \widehat{\theta}_t, \widehat{m}_t, \widehat{p}_t) - \psi^{\widehat{\pi}}_{\bar{w}_t}(\bar{Z}_{i,t}; \theta_t, \widehat{m}_t, \widehat{p}_t)\} = -(\widehat{\mu}^{dy\text{-}adj}_{\bar{w}_t}(t) - \mu_{\bar{w}_t}(t)).$$

The second equality holds because the RHS does not depend on $\bar{Z}_t$. Then

$$\widehat{\mu}^{dy\text{-}adj}_{\bar{w}_t}(t) - \mu_{\bar{w}_t}(t) = P_n \psi^{\pi}_{\bar{w}_t}(\bar{Z}_{i,t}; \theta_t, m_t, p_t) + P_n \Delta\psi_{\bar{w}_t}(\bar{Z}_t)$$
$$\sqrt{n}(\widehat{\mu}^{dy\text{-}adj}_{\bar{w}_t}(t) - \mu_{\bar{w}_t}(t)) = \mathbb{G}_n \psi^{\pi}_{\bar{w}_t}(\bar{Z}_{i,t}; \theta_t, m_t, p_t) + \sqrt{n}P_n\Delta\psi_{\bar{w}_t}(\bar{Z}_t)$$
$$= \mathbb{G}_n \psi^{\pi}_{\bar{w}_t}(\bar{Z}_{i,t}; \theta_t, m_t, p_t) + \underbrace{\sqrt{n}(P_n - P)\Delta\psi_{\bar{w}_t}(\bar{Z}_t)}_{\text{empirical process remainder } R_1} + \underbrace{\sqrt{n}P\Delta\psi_{\bar{w}_t}(\bar{Z}_t)}_{\text{bias remainder } R_2},$$

where $\sqrt{n}P_n\psi^{\pi}_{\bar{w}_t}(\cdot) = \mathbb{G}_n\psi^{\pi}_{\bar{w}_t}(\cdot)$ because of Lemma 1. In what follows, the perturbation $\Delta\psi_{\bar{w}_t}$ is defined using the cross-fitted nuisance estimators obtained by Algorithm 1. Specifically, for each unit $i$ in a fold $\ell$, we evaluate $\psi^{\widehat{\pi}}_{\bar{w}_t}(\bar{Z}_{i,t})$ with $(\widehat{m}_t, \widehat{p}_t) = (\widehat{m}^{(-\ell)}_t, \widehat{p}^{(-\ell)}_t)$. Thus, $R_1$ is the empirical process remainder term due to sampling variability under cross-fitting, and $R_2$ is the bias remainder term induced by plugging in the estimated nuisance functions and empirical treatment-path probabilities. We will show that $R_1 = o_p(1)$ and $R_2 = o_p(1)$.

**Step 2. Empirical process remainder term $R_1$.** First, we exploit the cross-fitting in Algorithm 1 to decompose $R_1$ into fold-specific terms $R_{1,\ell}$ that allow for conditional independence arguments. Let $I_1, \ldots, I_L$ be the random partition of $n$

random samples into $L$ folds used in Algorithm 1, and $n_\ell = |I_\ell|$ be the sample size in a fold $I_\ell$. For each fold $\ell$, we consider $\{\bar{Z}_{i,t} : i \in I_\ell^c\}$ as the training set; otherwise it is treated as the evaluation set, where $I_\ell^c := \bigcup_{k \neq \ell} I_k$ is the complementary set of $I_\ell$. First, we fit the nuisance estimators $(\widehat{m}_t^{(-\ell)}, \widehat{p}_t^{(-\ell)})$ via the learning models $\mathcal{M}$ and $\mathcal{T}$ using only the training set. For later use, define the fold-specific one-step innovation in the auxiliary correction by

$$\widehat{D}_{\tau,\bar{w}_t}^{(-\ell)}(\bar{H}_{\tau+1}) := \widehat{\Gamma}_{\bar{w}_t}^{(\tau+1),(-\ell)}(\bar{H}_{\tau+1}) - \int_{\mathcal{H}_{\tau+1}} \widehat{\Gamma}_{\bar{w}_t}^{(\tau+1),(-\ell)}(\bar{H}_\tau, h_{\tau+1})\widehat{p}_{\bar{w}_\tau}^{(\tau),(-\ell)}(dh_{\tau+1} \,|\, \bar{H}_\tau), \quad \tau = 1, \ldots, t-1.$$

Let $\widehat{A}_{\pi,\bar{w}_t}^{(t),(-\ell)}$ be the auxiliary correction constructed from these innovations with the true prefix normalizers $\{\pi_{\bar{w}_\tau}\}_{\tau=1}^{t-1}$, and let $\widehat{A}_{\widehat{\pi},\bar{w}_t}^{(t),(-\ell)}$ be its feasible counterpart with $\{\widehat{\pi}_{\bar{w}_\tau}\}_{\tau=1}^{t-1}$:

$$\widehat{A}_{\pi,\bar{w}_t}^{(t),(-\ell)}(\bar{H}_t) := \sum_{\tau=1}^{t-1} \frac{\mathbb{1}\{\bar{W}_\tau = \bar{w}_\tau\}}{\pi_{\bar{w}_\tau}} \widehat{D}_{\tau,\bar{w}_t}^{(-\ell)}(\bar{H}_{\tau+1}), \qquad \widehat{A}_{\widehat{\pi},\bar{w}_t}^{(t),(-\ell)}(\bar{H}_t) := \sum_{\tau=1}^{t-1} \frac{\mathbb{1}\{\bar{W}_\tau = \bar{w}_\tau\}}{\widehat{\pi}_{\bar{w}_\tau}} \widehat{D}_{\tau,\bar{w}_t}^{(-\ell)}(\bar{H}_{\tau+1}),$$

with the convention that these sums are zero when $t = 1$. We use the same notation without the superscript $(-\ell)$ for the resulting global cross-fitted quantity, evaluated fold by fold. Thus, $\psi_{\bar{w}_t}^\pi(\cdot; \theta_t, \widehat{m}_t^{(-\ell)}, \widehat{p}_t^{(-\ell)})$ uses $\widehat{A}_{\pi,\bar{w}_t}^{(t),(-\ell)}$, whereas $\psi_{\bar{w}_t}^{\widehat{\pi}}(\cdot; \theta_t, \widehat{m}_t^{(-\ell)}, \widehat{p}_t^{(-\ell)})$ uses $\widehat{A}_{\widehat{\pi},\bar{w}_t}^{(t),(-\ell)}$ and the empirical normalizer $\widehat{\pi}_{\bar{w}_t}$ in the top-level residual term. Next, for each unit $i \in I_\ell$ in the evaluation set of fold $\ell$, we decompose the perturbation term as follows:

$$\Delta\psi_{\bar{w}_t}^{(-\ell)} = \Delta_\eta \psi_{\bar{w}_t}^{(-\ell)} + \Delta_\pi \psi_{\bar{w}_t}^{(-\ell)},$$
$$\Delta_\eta \psi_{\bar{w}_t}^{(-\ell)}(\bar{Z}_t) := \psi_{\bar{w}_t}^\pi(\bar{Z}_t; \theta_t, \widehat{m}_t^{(-\ell)}, \widehat{p}_t^{(-\ell)}) - \psi_{\bar{w}_t}^\pi(\bar{Z}_t; \theta_t, m_t, p_t),$$
$$\Delta_\pi \psi_{\bar{w}_t}^{(-\ell)}(\bar{Z}_t) := \psi_{\bar{w}_t}^{\widehat{\pi}}(\bar{Z}_t; \theta_t, \widehat{m}_t^{(-\ell)}, \widehat{p}_t^{(-\ell)}) - \psi_{\bar{w}_t}^\pi(\bar{Z}_t; \theta_t, \widehat{m}_t^{(-\ell)}, \widehat{p}_t^{(-\ell)}).$$

Accordingly, write $R_1 = R_{1,\eta} + R_{1,\pi}$.

We first control $R_{1,\eta}$. By the construction of the cross-fitted scores, for each unit $i \in I_\ell$, the nuisance perturbation term is given by

$$\Delta_\eta \psi_{\bar{w}_t,i}^{(-\ell)} := \Delta_\eta \psi_{\bar{w}_t}^{(-\ell)}(\bar{Z}_{i,t}) = \Delta_\eta \psi_{\bar{w}_t}(\bar{Z}_{i,t}).$$

Moreover, the global cross-fitted nuisance estimators $(\widehat{m}_t, \widehat{p}_t)$ are obtained by setting $(\widehat{m}_t, \widehat{p}_t) = (\widehat{m}_t^{(-\ell)}, \widehat{p}_t^{(-\ell)})$ on the evaluation fold $I_\ell$. Consequently, the empirical average of $\Delta_\eta \psi_{\bar{w}_t}$ can be written as

$$P_n \Delta_\eta \psi_{\bar{w}_t} = \frac{1}{n} \sum_{\ell=1}^{L} \sum_{i \in I_\ell} \Delta_\eta \psi_{\bar{w}_t}^{(-\ell)}(\bar{Z}_{i,t}) = \sum_{\ell=1}^{L} \frac{n_\ell}{n} P_{n,\ell} \Delta_\eta \psi_{\bar{w}_t}^{(-\ell)},$$

where $P_{n,\ell}$ is the empirical measure on $\{\bar{Z}_{i,t} : i \in I_\ell\}$. Also, by the definition of $\Delta_\eta \psi_{\bar{w}_t}$, the population expectation $P \Delta_\eta \psi_{\bar{w}_t}$ is written as a weighted average of $P \Delta_\eta \psi_{\bar{w}_t}^{(-\ell)}$ as follows:

$$P \Delta_\eta \psi_{\bar{w}_t} = \sum_{\ell=1}^{L} \frac{n_\ell}{n} P \Delta_\eta \psi_{\bar{w}_t}^{(-\ell)},$$

Therefore, the nuisance empirical process term $R_{1,\eta}$ can be written as

$$R_{1,\eta} = \sqrt{n}(P_n - P)\Delta_\eta \psi_{\bar{w}_t} = \sum_{\ell=1}^{L} \sqrt{\frac{n_\ell}{n}} \mathbb{G}_{n,\ell} \Delta_\eta \psi_{\bar{w}_t}^{(-\ell)} = \sum_{\ell=1}^{L} R_{1,\eta,\ell},$$

where $\mathbb{G}_{n,\ell} f := \sqrt{n_\ell}(P_{n,\ell} - P) f$ is the empirical process based on the subsample $\{\bar{Z}_{i,t} : i \in I_\ell\}$, and

$$R_{1,\eta,\ell} := \sqrt{\frac{n_\ell}{n}} \mathbb{G}_{n,\ell} \Delta_\eta \psi_{\bar{w}_t}^{(-\ell)} = \frac{1}{\sqrt{n}} \sum_{i \in I_\ell} \left\{ \Delta_\eta \psi_{\bar{w}_t,i}^{(-\ell)} - P \Delta_\eta \psi_{\bar{w}_t}^{(-\ell)} \right\},$$

and $\mathbb{G}_{n,\ell}$ is the empirical process based on the subsample $\{\bar{Z}_{i,t} : i \in I_\ell\}$.

Next, we control each fold-specific term $R_{1,\eta,\ell}$ through a conditional variance argument (Lemma 7) that exploits the independence between the training and evaluation samples induced by cross-fitting. For a given fold $\ell$, let $\mathcal{G}_\ell$ be the $\sigma$-algebra generated by the training set $\{\bar{Z}_{i,t} : i \in I_\ell^c\}$ together with all the randomness used to construct $(\widehat{m}_t^{(-\ell)}, \widehat{p}_t^{(-\ell)})$. Conditional on $\mathcal{G}_\ell$, the function $\Delta_\eta \psi_{\bar{w}_t}^{(-\ell)}(\cdot)$ is deterministic, while the evaluation sample $\{\bar{Z}_{i,t} : i \in I_\ell\}$ is i.i.d. with law $P$ and independent of $\mathcal{G}_\ell$ by Assumption 2. Therefore, the assumptions of Lemma 7 are satisfied for the conditional law of $\{\bar{Z}_{i,t} : i \in I_\ell\}$ given $\mathcal{G}_\ell$, and its variance bound applied with $\mathbb{E}[\cdot]$ replaced by $\mathbb{E}[\cdot \mid \mathcal{G}_\ell]$. Hence, applying Lemma 7 to the empirical process $\mathbb{G}_{n,\ell}$ on fold $\ell$ yields

$$\mathbb{E}\big[R_{1,\eta,\ell}^2 \mid \mathcal{G}_\ell\big] = \frac{n_\ell}{n}\mathbb{E}\Big[(\mathbb{G}_{n,\ell}\Delta_\eta\psi_{\bar{w}_t}^{(-\ell)})^2 \;\big|\; \mathcal{G}_\ell\Big] \le \frac{n_\ell}{n}\Big\|\Delta_\eta\psi_{\bar{w}_t}^{(-\ell)}\Big\|_{P,2}^2 \quad a.s.$$

Now, fix $\varepsilon > 0$ and an arbitrary threshold $a > 0$. By the conditional Chebyshev inequality,

$$P(|R_{1,\eta,\ell}| > \varepsilon \mid \mathcal{G}_\ell) \le \frac{1}{\varepsilon^2}\mathbb{E}\big[R_{1,\eta,\ell}^2 \mid \mathcal{G}_\ell\big] \le \frac{1}{\varepsilon^2}\frac{n_\ell}{n}\Big\|\Delta_\eta\psi_{\bar{w}_t}^{(-\ell)}\Big\|_{P,2}^2. \tag{C.1}$$

Decomposing with respect to $\left\{\Big\|\Delta_\eta\psi_{\bar{w}_t}^{(-\ell)}\Big\|_{P,2} \le a\right\}$, we have

$$P\left(|R_{1,\eta,\ell}| > \varepsilon\right) = P\left(|R_{1,\eta,\ell}| > \varepsilon, \Big\|\Delta_\eta\psi_{\bar{w}_t}^{(-\ell)}\Big\|_{P,2} > a\right) + P\left(|R_{1,\eta,\ell}| > \varepsilon, \Big\|\Delta_\eta\psi_{\bar{w}_t}^{(-\ell)}\Big\|_{P,2} \le a\right).$$

Using the basic inequality $P(A \cap B) \le P(A)$, the first term is bounded by

$$P\left(|R_{1,\eta,\ell}| > \varepsilon, \Big\|\Delta_\eta\psi_{\bar{w}_t}^{(-\ell)}\Big\|_{P,2} > a\right) \le P\left(\Big\|\Delta_\eta\psi_{\bar{w}_t}^{(-\ell)}\Big\|_{P,2} > a\right).$$

For the second term, we use the conditional bound and Eq. (C.1)

$$\begin{aligned}
P\left(|R_{1,\eta,\ell}| > \varepsilon, \Big\|\Delta_\eta\psi_{\bar{w}_t}^{(-\ell)}\Big\|_{P,2} \le a\right) &= \mathbb{E}\left[\mathbb{1}\left\{\Big\|\Delta_\eta\psi_{\bar{w}_t}^{(-\ell)}\Big\|_{P,2} \le a\right\}\mathbb{1}\left\{|R_{1,\eta,\ell}| > \varepsilon\right\}\right] \\
&= \mathbb{E}\left[\mathbb{E}\left[\mathbb{1}\left\{\Big\|\Delta_\eta\psi_{\bar{w}_t}^{(-\ell)}\Big\|_{P,2} \le a\right\}\mathbb{1}\left\{|R_{1,\eta,\ell}| > \varepsilon\right\}\;\Big|\;\mathcal{G}_\ell\right]\right] \\
&= \mathbb{E}\left[\mathbb{1}\left\{\Big\|\Delta_\eta\psi_{\bar{w}_t}^{(-\ell)}\Big\|_{P,2} \le a\right\}\mathbb{E}[\mathbb{1}\left\{|R_{1,\eta,\ell}| > \varepsilon\right\} \mid \mathcal{G}_\ell]\right] \\
&= \mathbb{E}\left[\mathbb{1}\left\{\Big\|\Delta_\eta\psi_{\bar{w}_t}^{(-\ell)}\Big\|_{P,2} \le a\right\}P\left(|R_{1,\eta,\ell}| > \varepsilon \mid \mathcal{G}_\ell\right)\right] \\
&\le \frac{1}{\varepsilon^2}\frac{n_\ell}{n}\mathbb{E}\left[\mathbb{1}\left\{\Big\|\Delta_\eta\psi_{\bar{w}_t}^{(-\ell)}\Big\|_{P,2} \le a\right\}\Big\|\Delta_\eta\psi_{\bar{w}_t}^{(-\ell)}\Big\|_{P,2}^2\right] \\
&\le \frac{1}{\varepsilon^2}\frac{n_\ell}{n}a^2.
\end{aligned}$$

This final bound follows immediately from the fact that $\|\Delta_\eta\psi_{\bar{w}_t}^{(-\ell)}\|_{P,2}^2$ is at most $a^2$ whenever the indicator function is true. Combining these bounds yields, for every $\varepsilon > 0$ and $a > 0$,

$$P\left(|R_{1,\eta,\ell}| > \varepsilon\right) \le P\left(\Big\|\Delta_\eta\psi_{\bar{w}_t}^{(-\ell)}\Big\|_{P,2} > a\right) + \frac{1}{\varepsilon^2}\frac{n_\ell}{n}a^2. \tag{C.2}$$

This formula implies that the tail probability of $R_{1,\eta,\ell}$ is controlled by the $L_2$-norm of the nuisance perturbation $\Delta_\eta\psi_{\bar{w}_t}^{(-\ell)}$. In particular, for any fixed $\varepsilon > 0$, we have $P(|R_{1,\eta,\ell}| > \varepsilon) \to 0$ as soon as $\|\Delta_\eta\psi_{\bar{w}_t}^{(-\ell)}\|_{P,2} \to 0$ in probability.

Here, we derive a bound for the aggregate nuisance empirical process term $R_{1,\eta} = \sum_{\ell=1}^L R_{1,\eta,\ell}$. First, by applying the triangle inequality, we bound the magnitude of the sum by the sum of magnitudes:

$$|R_{1,\eta}| = \left|\sum_{\ell=1}^L R_{1,\eta,\ell}\right| \le \sum_{\ell=1}^L |R_{1,\eta,\ell}|.$$

Next, observe that the event $\{|R_{1,\eta}| > \varepsilon\}$ implies that the sum of the magnitudes exceeds. For this to occur, at least one of the fold-specific terms $|R_{1,\eta,\ell}|$ must exceed the average contribution $\varepsilon/L$. This leads to the following set inclusion:

$$\{|R_{1,\eta}| > \varepsilon\} \subseteq \bigcup_{\ell=1}^{L} \left\{|R_{1,\eta,\ell}| > \frac{\varepsilon}{L}\right\}$$

Therefore, applying the union bound yields

$$P(|R_{1,\eta}| > \varepsilon) \leq \sum_{\ell=1}^{L} P\left(|R_{1,\eta,\ell}| > \frac{\varepsilon}{L}\right)$$

Applying Eq. (C.2) with $\varepsilon$ replaced by $\varepsilon/L$ yields, for any $a > 0$,

$$\begin{aligned}
P(|R_{1,\eta}| > \varepsilon) &\leq \sum_{\ell=1}^{L} P\left(\left\|\Delta_\eta \psi_{\bar{w}_t}^{(-\ell)}\right\|_{P,2} > a\right) + \frac{L^2}{\varepsilon^2} a^2 \sum_{\ell=1}^{L} \frac{n_\ell}{n} \\
&\leq \sum_{\ell=1}^{L} P\left(\left\|\Delta_\eta \psi_{\bar{w}_t}^{(-\ell)}\right\|_{P,2} > a\right) + \frac{L^2}{\varepsilon^2} a^2.
\end{aligned}$$

(C.3)

Finally, we show that the nuisance perturbation term $\max_{\ell \in [L]} \|\Delta_\eta \psi_{\bar{w}_t}^{(-\ell)}\|_{P,2} \to 0$ in probability as $n \to \infty$. From the definition of $\psi_{\bar{w}_t}^\pi$, we have for any $\bar{Z}_t$,

$$\Delta_\eta \psi_{\bar{w}_t}(\bar{Z}_t) = -\frac{\mathbb{1}\{\bar{W}_t = \bar{w}_t\}}{\pi_{\bar{w}_t}} \left\{\widehat{\Gamma}_{\bar{w}_t}^{(t)}(\bar{H}_t) - \Gamma_{\bar{w}_t}^{(t)}(\bar{H}_t)\right\} + \left\{\widehat{A}_{\pi,\bar{w}_t}^{(t)}(\bar{H}_t) - A_{\bar{w}_t}^{(t)}(\bar{H}_t)\right\} + \left\{\widehat{\Gamma}_{\bar{w}_t}^{(1)}(X_1) - \Gamma_{\bar{w}_t}^{(1)}(X_1)\right\}.$$

Hence, by the triangle inequality and the positivity Assumption 4, we obtain

$$\|\Delta_\eta \psi_{\bar{w}_t}\|_{P,2} \lesssim \left\|\widehat{\Gamma}_{\bar{w}_t}^{(t)}(\bar{H}_t) - \Gamma_{\bar{w}_t}^{(t)}(\bar{H}_t)\right\|_{P,2} + \left\|\widehat{A}_{\pi,\bar{w}_t}^{(t)}(\bar{H}_t) - A_{\bar{w}_t}^{(t)}(\bar{H}_t)\right\|_{P,2} + \left\|\widehat{\Gamma}_{\bar{w}_t}^{(1)}(X_1) - \Gamma_{\bar{w}_t}^{(1)}(X_1)\right\|_{P,2},$$

where $A \lesssim B$ denotes that there exists a constant $C$ such that $A \leq C \cdot B$. By Assumption 7, the RHS is further bounded by

$$\|\Delta_\eta \psi_{\bar{w}_t}\|_{P,2} \lesssim \omega(\|\widehat{m}_t - m_t\|_{P,2} + \|\widehat{p}_t - p_t\|_{P,2}),$$

which is $o_p(1)$ by Assumption 6. Recalling the cross-fitting in Algorithm 1, every training set $I_\ell^c$ approximately contains $(1 - 1/L)n$ samples, i.e., its sample sizes are of order $n$. Hence, the same argument applies to the fold-specific nuisance estimators $(\widehat{m}_t^{(-\ell)}, \widehat{p}_t^{(-\ell)})$. Specifically, we obtain $\|\Delta_\eta \psi_{\bar{w}_t}^{(-\ell)}\|_{P,2} = o_p(1)$ for each fold $\ell \in [L]$. Since $L$ is fixed, this implies

$$\max_{\ell \in [L]} \left\|\Delta_\eta \psi_{\bar{w}_t}^{(-\ell)}\right\|_{P,2} \xrightarrow{p} 0.$$

In conclusion, with the bound in Eq. (C.3) and the choice of a deterministic sequence $a_n \downarrow 0$ sufficiently slowly, we obtain $P(|R_{1,\eta}| > \varepsilon) \to 0$ for every $\varepsilon > 0$, that is,

$$R_{1,\eta} = \sqrt{n}(P_n - P)\Delta_\eta \psi_{\bar{w}_t} = o_p(1).$$

It remains to control $R_{1,\pi}$. By the definitions of $\widehat{A}_{\pi,\bar{w}_t}^{(t),(-\ell)}$ and $\widehat{A}_{\widehat{\pi},\bar{w}_t}^{(t),(-\ell)}$,

$$\Delta_\pi \psi_{\bar{w}_t}^{(-\ell)}(\bar{Z}_t) = (\widehat{\pi}_{\bar{w}_t}^{-1} - \pi_{\bar{w}_t}^{-1})\mathbb{1}\{\bar{W}_t = \bar{w}_t\}\left\{Y_t - \widehat{\Gamma}_{\bar{w}_t}^{(t),(-\ell)}(\bar{H}_t)\right\} + \sum_{\tau=1}^{t-1}(\widehat{\pi}_{\bar{w}_\tau}^{-1} - \pi_{\bar{w}_\tau}^{-1})\mathbb{1}\{\bar{W}_\tau = \bar{w}_\tau\}\widehat{D}_{\tau,\bar{w}_t}^{(-\ell)}(\bar{H}_{\tau+1}).$$

Thus,

$$R_{1,\pi} = (\widehat{\pi}_{\bar{w}_t}^{-1} - \pi_{\bar{w}_t}^{-1})\sum_{\ell=1}^{L} \sqrt{\frac{n_\ell}{n}}\mathbb{G}_{n,\ell}\left[\mathbb{1}\{\bar{W}_t = \bar{w}_t\}\left\{Y_t - \widehat{\Gamma}_{\bar{w}_t}^{(t),(-\ell)}(\bar{H}_t)\right\}\right]$$

$$+ \sum_{\tau=1}^{t-1}(\widehat{\pi}_{\bar{w}_\tau}^{-1} - \pi_{\bar{w}_\tau}^{-1}) \sum_{\ell=1}^{L} \sqrt{\frac{n_\ell}{n}} \mathbb{G}_{n,\ell} \left[ \mathbb{1}\{\bar{W}_\tau = \bar{w}_\tau\}\widehat{D}_{\tau,\bar{w}_t}^{(-\ell)}(\bar{H}_{\tau+1}) \right].$$

By Assumptions 2 and 4, and since the number of feasible treatment histories and the horizon are fixed,

$$\max_{\tau \leq t} \left|\widehat{\pi}_{\bar{w}_\tau}^{-1} - \pi_{\bar{w}_\tau}^{-1}\right| = O_p(n^{-1/2}).$$

The top-level and prefix empirical-process sums in the last display are $O_p(1)$ by the same conditional variance bound and Assumptions 5-7. Hence $R_{1,\pi} = o_p(1)$. Combining $R_{1,\eta} = o_p(1)$ and $R_{1,\pi} = o_p(1)$, we obtain

$$R_1 = \sqrt{n}(P_n - P)\Delta\psi_{\bar{w}_t} = o_p(1).$$

This establishes the desired control over the empirical process remainder term.

**Step 3. Bias remainder term $R_2$.** We now control the population bias remainder. By the cross-fitted construction,

$$P\Delta\psi_{\bar{w}_t} = \sum_{\ell=1}^{L} \frac{n_\ell}{n}P\Delta\psi_{\bar{w}_t}^{(-\ell)},$$

where

$$P\Delta\psi_{\bar{w}_t}^{(-\ell)} = P\left[ \psi_{\bar{w}_t}^{\widehat{\pi}}(\bar{Z}_t; \theta_t, \widehat{m}_t^{(-\ell)}, \widehat{p}_t^{(-\ell)}) - \psi_{\bar{w}_t}^{\pi}(\bar{Z}_t; \theta_t, m_t, p_t) \right].$$

For each fold $\ell$, the exact telescoping calculation in the proof of Lemma 2 can be applied directly to the cross-fitted nuisance estimates $(\widehat{m}_t^{(-\ell)}, \widehat{p}_t^{(-\ell)})$, because these estimates are constructed on the training fold and then evaluated on the held-out fold. In particular,

$$P\left[ \frac{\mathbb{1}\{\bar{W}_t = \bar{w}_t\}}{\pi_{\bar{w}_t}} \left\{Y_t - \widehat{\Gamma}_{\bar{w}_t}^{(t),(-\ell)}(\bar{H}_t)\right\} + \sum_{\tau=1}^{t-1} \frac{\mathbb{1}\{\bar{W}_\tau = \bar{w}_\tau\}}{\pi_{\bar{w}_\tau}} \widehat{D}_{\tau,\bar{w}_t}^{(-\ell)}(\bar{H}_{\tau+1}) + \widehat{\Gamma}_{\bar{w}_t}^{(1),(-\ell)}(X_1) - \mu_{\bar{w}_t}(t) \right] = 0.$$

Likewise, by Lemma 1,

$$P\left[ \frac{\mathbb{1}\{\bar{W}_t = \bar{w}_t\}}{\pi_{\bar{w}_t}} \left\{Y_t - \Gamma_{\bar{w}_t}^{(t)}(\bar{H}_t)\right\} + A_{\bar{w}_t}^{(t)}(\bar{H}_t) + \Gamma_{\bar{w}_t}^{(1)}(X_1) - \mu_{\bar{w}_t}(t) \right] = 0.$$

Hence, the contribution to $P\Delta\psi_{\bar{w}_t}^{(-\ell)}$ comes from replacing the true treatment-path probabilities by their empirical analogs in the top-level residual term and in the prefix normalizers of the auxiliary correction term. Therefore,

$$P\Delta\psi_{\bar{w}_t}^{(-\ell)} = (\widehat{\pi}_{\bar{w}_t}^{-1} - \pi_{\bar{w}_t}^{-1})P\left[ \mathbb{1}\{\bar{W}_t = \bar{w}_t\}\left\{Y_t - \widehat{\Gamma}_{\bar{w}_t}^{(t),(-\ell)}(\bar{H}_t)\right\} \right]$$

$$+ \sum_{\tau=1}^{t-1}(\widehat{\pi}_{\bar{w}_\tau}^{-1} - \pi_{\bar{w}_\tau}^{-1})P\left[ \mathbb{1}\{\bar{W}_\tau = \bar{w}_\tau\}\widehat{D}_{\tau,\bar{w}_t}^{(-\ell)}(\bar{H}_{\tau+1}) \right].$$

Since

$$P\left[ \mathbb{1}\{\bar{W}_t = \bar{w}_t\}\left\{Y_t - \Gamma_{\bar{w}_t}^{(t)}(\bar{H}_t)\right\} \right] = 0,$$

we have

$$P\left[ \mathbb{1}\{\bar{W}_t = \bar{w}_t\}\left\{Y_t - \widehat{\Gamma}_{\bar{w}_t}^{(t),(-\ell)}(\bar{H}_t)\right\} \right] = P\left[ \mathbb{1}\{\bar{W}_t = \bar{w}_t\}\left\{\Gamma_{\bar{w}_t}^{(t)}(\bar{H}_t) - \widehat{\Gamma}_{\bar{w}_t}^{(t),(-\ell)}(\bar{H}_t)\right\} \right].$$

By Cauchy-Schwarz and Assumptions 6 and 7,

$$\left|P\left[ \mathbb{1}\{\bar{W}_t = \bar{w}_t\}\left\{\Gamma_{\bar{w}_t}^{(t)}(\bar{H}_t) - \widehat{\Gamma}_{\bar{w}_t}^{(t),(-\ell)}(\bar{H}_t)\right\} \right]\right| \leq \|\widehat{\Gamma}_{\bar{w}_t}^{(t),(-\ell)} - \Gamma_{\bar{w}_t}^{(t)}\|_{P,2} = o_p(1).$$

For the prefix terms, define the true one-step innovation

$$D_{\tau,\bar{w}_t}(\bar{H}_{\tau+1}) := \Gamma_{\bar{w}_t}^{(\tau+1)}(\bar{H}_{\tau+1}) - \int_{\mathcal{H}_{\tau+1}} \Gamma_{\bar{w}_t}^{(\tau+1)}(\bar{H}_\tau, h_{\tau+1})p_{\bar{w}_\tau}^{(\tau)}(dh_{\tau+1} \mid \bar{H}_\tau).$$

By the conditional-mean property of the true transition kernel, $P[\mathbb{1}\{\bar{W}_\tau = \bar{w}_\tau\}D_{\tau,\bar{w}_t}(\bar{H}_{\tau+1})] = 0$. Therefore, by Cauchy-Schwarz and the same stability argument used for the auxiliary correction term,

$$\left| P\left[\mathbb{1}\{\bar{W}_\tau = \bar{w}_\tau\}\widehat{D}^{(-\ell)}_{\tau,\bar{w}_t}(\bar{H}_{\tau+1})\right]\right| = \left| P\left[\mathbb{1}\{\bar{W}_\tau = \bar{w}_\tau\}\left\{\widehat{D}^{(-\ell)}_{\tau,\bar{w}_t}(\bar{H}_{\tau+1}) - D_{\tau,\bar{w}_t}(\bar{H}_{\tau+1})\right\}\right]\right|$$

$$\leq \left\|\widehat{D}^{(-\ell)}_{\tau,\bar{w}_t} - D_{\tau,\bar{w}_t}\right\|_{P,2} = o_p(1),$$

uniformly over the finitely many $\tau = 1, \ldots, t - 1$, by Assumptions 6 and 7. Moreover, by Assumptions 2 and 4, and since the number of feasible treatment histories and the horizon are fixed,

$$\max_{\tau \leq t}|\widehat{\pi}_{\bar{w}_\tau} - \pi_{\bar{w}_\tau}| = O_p(n^{-1/2}), \quad \sqrt{n}\max_{\tau \leq t}|\widehat{\pi}^{-1}_{\bar{w}_\tau} - \pi^{-1}_{\bar{w}_\tau}| = O_p(1).$$

Combining the previous formulations and using that $L$ and $t$ are fixed, we obtain

$$R_2 = \sqrt{n}\sum_{\ell=1}^{L}\frac{n_\ell}{n}P\Delta\psi^{(-\ell)}_{\bar{w}_t} = O_p(1)o_p(1) = o_p(1).$$

This shows that the bias remainder term is negligible. The argument uses the telescoping structure of the moment function and the root-$n$ consistency of the empirical treatment path proportions.

**Step 4. Derivation of asymptotic normality.** For any $t \in [T]$, substituting the bounds for the remainder terms $R_1 = o_p(1)$ and $R_2 = o_p(1)$ into the asymptotic linear representation derived in Step 1 yields

$$\sqrt{n}(\widehat{\theta}_t - \theta_t) = \frac{1}{\sqrt{n}}\sum_{i=1}^{n}\psi^\pi_t(\bar{Z}_{i,t};\theta_t, m_t, p_t) + o_p(1).$$

Here, $\psi^\pi_t(\bar{Z}_{i,t};\theta_t, m_t, p_t)$ is a vector of influence functions defined component-wise for each treatment history $\bar{w}_t \in \bar{\mathcal{W}}_t$ as shown in Eq. (11). The random vectors $\{\psi^\pi_t(\bar{Z}_{i,t};\theta_t, m_t, p_t)\}^n_{i=1}$ are independent and identically distributed (i.i.d.) by Assumption 2.

Furthermore, Lemma 1 ensures that these influence functions have zero mean, i.e., $\mathbb{E}\left[\psi^\pi_t(\bar{Z}_t;\theta_t, m_t, p_t)\right] = 0$. Assumption 5 guarantees the existence of finite second moments for the moment function, ensuring that the covariance matrix $\Sigma_t = \text{Var}\left(\psi^\pi_t(\bar{Z}_t;\theta_t, m_t, p_t)\right)$ is well-defined and finite.

Therefore, by applying the Multivariate Central Limit Theorem (CLT) to the leading term, we obtain:

$$\frac{1}{\sqrt{n}}\sum_{i=1}^{n}\psi^\pi_t(\bar{Z}_{i,t};\theta_t, m_t, p_t) \xrightarrow{d} \mathcal{N}(0, \Sigma_t),$$

as $n \to \infty$. Finally, by Slutsky's theorem, the addition of the $o_p(1)$ remainder term does not affect the asymptotic distribution. Consequently, we conclude that:

$$\sqrt{n}(\widehat{\theta}_t - \theta_t) \xrightarrow{d} \mathcal{N}(0, \Sigma_t).$$

Hence, the desired result is obtained. $\qquad\qquad\square$

## C.5. Proof of Theorem 5

**Theorem 5** (Semiparametric efficiency bound). *Under Assumptions 1-5, for any $t \in [T]$, the semiparametric efficiency bound for $\theta_t$ is $\Sigma_t = \text{Var}\left(\psi^\pi_t(\bar{Z}_t;\theta_t, m_t, p_t)\right)$. Moreover, if Assumptions 6-7 also hold, the dynamic regression-adjusted estimator $\widehat{\theta}_t$ attains the semiparametric efficiency bound.*

*Proof.* First, we characterize the tangent space. For any $t \in [T]$, the joint density of the observed random variables $\bar{Z}_t = (\bar{X}_t, \bar{W}_t, \bar{Y}_t)$ can be written as:

$$p(\bar{z}_t) = \prod_{\tau=1}^{t}p(x_\tau \mid \bar{z}_{\tau-1})p(w_\tau \mid \bar{z}_{\tau-1}, x_\tau)p(y_\tau \mid \bar{z}_{\tau-1}, x_\tau, w_\tau)$$

$$= \pi_{\bar{w}_t} \prod_{\tau=1}^{t} p(x_\tau \mid \bar{z}_{\tau-1}) p(y_\tau \mid \bar{h}_\tau, \bar{w}_\tau),$$

where $\bar{z}_0$ is an empty set, the equality $\prod_{\tau=1}^{t} p(w_\tau \mid \bar{z}_{\tau-1}, x_\tau) = \pi_{\bar{w}_t}$ follows from the known fixed randomization law under Assumption 3, and $(\bar{z}_{\tau-1}, x_\tau, w_\tau) = (\bar{h}_\tau, \bar{w}_\tau)$. Since the assignment probabilities are fixed by the randomized design, the treatment assignment law contributes no nuisance tangent direction. Here, let $\mathscr{T}_t$ be the tangent space at time $t$. The tangent directions arise only from the conditional covariate laws $p(x_\tau \mid \bar{z}_{\tau-1})$ and the conditional outcome laws $p(y_\tau \mid \bar{h}_\tau, \bar{w}_\tau)$. Specifically, $\mathscr{T}_t$ is the closed linear span of the following sum of square-integrable martingale differences:

$$\mathscr{T}_t = \mathrm{cl}_{L_2(P)} \left( \left\{ \sum_{\tau=1}^{t} \left[ \mathscr{S}_{X,\tau}(\bar{Z}_{\tau-1}, X_\tau) + \mathscr{S}_{Y,\tau}(\bar{H}_\tau, \bar{W}_\tau, Y_\tau) \right] : \mathscr{S}_{X,\tau} \in \mathscr{T}_{X,\tau}, \mathscr{S}_{Y,\tau} \in \mathscr{T}_{Y,\tau} \right\} \right).$$

In this equation, each subspace is defined as follows, with $\bar{Z}_0$ interpreted as the empty history:

$$\mathscr{T}_{X,\tau} := \left\{ s_X(\bar{Z}_{\tau-1}, X_\tau) \in L_2(P) : \mathbb{E}\left[ s_X(\bar{Z}_{\tau-1}, X_\tau) \mid \bar{Z}_{\tau-1} \right] = 0 \right\},$$
$$\mathscr{T}_{Y,\tau} := \left\{ s_Y(\bar{H}_\tau, \bar{W}_\tau, Y_\tau) \in L_2(P) : \mathbb{E}\left[ s_Y(\bar{H}_\tau, \bar{W}_\tau, Y_\tau) \mid \bar{H}_\tau, \bar{W}_\tau \right] = 0 \right\}.$$

We next show that the efficient influence function for $\theta_t$ coincides with the score $\psi_t^\pi(\bar{Z}_t; \theta_t, m_t, p_t)$ introduced in Section 4.3, and hence that the semiparametric efficiency bound is given by $\Sigma_t = \mathrm{Var}\left( \psi_t^\pi(\bar{Z}_t; \theta_t, m_t, p_t) \right)$ whenever the displayed covariance is finite. For notational convenience, fix $t \in [T]$ and a treatment history $\bar{w}_t \in \bar{\mathcal{W}}_t$, and work with the scalar component $\mu_{\bar{w}_t}(t) \in \theta_t$.

**Step 1. Membership of $\psi_{\bar{w}_t}^\pi$ in the tangent space $\mathscr{T}_t$.** We decompose the moment function $\psi_{\bar{w}_t}^\pi$ into time indexed martingale differences that match the structure of the tangent space. First, define

$$\mathscr{S}_{X,1}(X_1) := \Gamma_{\bar{w}_t}^{(1)}(X_1) - \mu_{\bar{w}_t}(t),$$

which is measurable with respect to $\sigma(X_1)$. Since $\bar{Z}_0$ is the empty history, we compute

$$\mathbb{E}\left[ \mathscr{S}_{X,1}(X_1) \mid \bar{Z}_0 \right] = \mathbb{E}\left[ \Gamma_{\bar{w}_t}^{(1)}(X_1) - \mu_{\bar{w}_t}(t) \right] = 0,$$

so that $\mathscr{S}_{X,1} \in \mathscr{T}_{X,1}$.

Next, we decompose each one-step innovation in the auxiliary correction term into an outcome component at time $\tau$ and a covariate-transition component at time $\tau + 1$. For $\tau = 1, \ldots, t-1$, define

$$G_{\tau+1} := \Gamma_{\bar{w}_t}^{(\tau+1)}(\bar{H}_{\tau+1}),$$
$$C_\tau(\bar{Z}_\tau) := \mathbb{E}\left[ G_{\tau+1} \mid \bar{Z}_\tau \right],$$
$$M_\tau(\bar{H}_\tau) := \mathbb{E}\left[ G_{\tau+1} \mid \bar{H}_\tau, \bar{W}_\tau = \bar{w}_\tau \right].$$

By the definition of the transition kernel,

$$M_\tau(\bar{H}_\tau) = \int_{\mathcal{H}_{\tau+1}} \Gamma_{\bar{w}_t}^{(\tau+1)}(\bar{H}_\tau, h_{\tau+1}) p_{\bar{w}_\tau}^{(\tau)}(dh_{\tau+1} \mid \bar{H}_\tau).$$

Hence, the auxiliary correction term can be written as

$$A_{\bar{w}_t}^{(t)}(\bar{H}_t) = \sum_{\tau=1}^{t-1} \frac{\mathbb{1}\{\bar{W}_\tau = \bar{w}_\tau\}}{\pi_{\bar{w}_\tau}} \left\{ G_{\tau+1} - M_\tau(\bar{H}_\tau) \right\}$$

$$= \sum_{\tau=1}^{t-1} \mathscr{S}_{Y,\tau}^A(\bar{H}_\tau, \bar{W}_\tau, Y_\tau) + \sum_{\tau=1}^{t-1} \mathscr{S}_{X,\tau+1}^A(\bar{Z}_\tau, X_{\tau+1}),$$

where

$$\mathscr{S}_{Y,\tau}^A(\bar{H}_\tau, \bar{W}_\tau, Y_\tau) := \frac{\mathbb{1}\{\bar{W}_\tau = \bar{w}_\tau\}}{\pi_{\bar{w}_\tau}} \left\{ C_\tau(\bar{Z}_\tau) - M_\tau(\bar{H}_\tau) \right\},$$

$$\mathscr{S}^A_{X,\tau+1}(\bar{Z}_\tau, X_{\tau+1}) := \frac{\mathbb{1}\{\bar{W}_\tau = \bar{w}_\tau\}}{\pi_{\bar{w}_\tau}} \left\{ G_{\tau+1} - C_\tau(\bar{Z}_\tau) \right\}.$$

The first component is measurable with respect to $\sigma(\bar{H}_\tau, \bar{W}_\tau, Y_\tau)$, since $\bar{Z}_\tau = (\bar{H}_\tau, \bar{W}_\tau, Y_\tau)$. Moreover,

$$\mathbb{E}\left[\mathscr{S}^A_{Y,\tau}(\bar{H}_\tau, \bar{W}_\tau, Y_\tau) \,|\, \bar{H}_\tau, \bar{W}_\tau\right] = \frac{\mathbb{1}\{\bar{W}_\tau = \bar{w}_\tau\}}{\pi_{\bar{w}_\tau}} \left\{ \mathbb{E}\left[C_\tau(\bar{Z}_\tau) \,|\, \bar{H}_\tau, \bar{W}_\tau\right] - M_\tau(\bar{H}_\tau) \right\} = 0,$$

because on the event $\{\bar{W}_\tau = \bar{w}_\tau\}$,

$$\mathbb{E}\left[C_\tau(\bar{Z}_\tau) \,|\, \bar{H}_\tau, \bar{W}_\tau = \bar{w}_\tau\right] = \mathbb{E}\left[G_{\tau+1} \,|\, \bar{H}_\tau, \bar{W}_\tau = \bar{w}_\tau\right] = M_\tau(\bar{H}_\tau).$$

Thus, $\mathscr{S}^A_{Y,\tau} \in \mathscr{T}_{Y,\tau}$. The second component is measurable with respect to $\sigma(\bar{Z}_\tau, X_{\tau+1})$, since $G_{\tau+1} = \Gamma^{(\tau+1)}_{\bar{w}_t}(\bar{H}_{\tau+1})$ is a function of $(\bar{Z}_\tau, X_{\tau+1})$. Moreover,

$$\mathbb{E}\left[\mathscr{S}^A_{X,\tau+1}(\bar{Z}_\tau, X_{\tau+1}) \,|\, \bar{Z}_\tau\right] = \frac{\mathbb{1}\{\bar{W}_\tau = \bar{w}_\tau\}}{\pi_{\bar{w}_\tau}} \left\{ \mathbb{E}\left[G_{\tau+1} \,|\, \bar{Z}_\tau\right] - C_\tau(\bar{Z}_\tau) \right\} = 0,$$

so that $\mathscr{S}^A_{X,\tau+1} \in \mathscr{T}_{X,\tau+1}$.

Finally, define

$$\mathscr{S}_{Y,t}(\bar{H}_t, \bar{W}_t, Y_t) := \frac{\mathbb{1}\{\bar{W}_t = \bar{w}_t\}}{\pi_{\bar{w}_t}} \left\{ Y_t - \Gamma^{(t)}_{\bar{w}_t}(\bar{H}_t) \right\}.$$

This component is measurable with respect to $\sigma(\bar{H}_t, \bar{W}_t, Y_t)$. We compute the expectation of both sides

$$\mathbb{E}\left[\mathscr{S}_{Y,t}(\bar{H}_t, \bar{W}_t, Y_t) \,|\, \bar{H}_t, \bar{W}_t\right] = \frac{\mathbb{1}\{\bar{W}_t = \bar{w}_t\}}{\pi_{\bar{w}_t}} \left\{ \mathbb{E}\left[Y_t \,|\, \bar{H}_t, \bar{W}_t = \bar{w}_t\right] - \Gamma^{(t)}_{\bar{w}_t}(\bar{H}_t) \right\} = 0,$$

so that $\mathscr{S}_{Y,t} \in \mathscr{T}_{Y,t}$. Hence, the moment function $\psi^\pi_{\bar{w}_t}(\bar{Z}_t)$ is described as

$$\psi^\pi_{\bar{w}_t}(\bar{Z}_t) = \mathscr{S}_{X,1}(X_1) + \mathscr{S}_{Y,t}(\bar{H}_t, \bar{W}_t, Y_t) + \sum_{\tau=1}^{t-1} \left\{ \mathscr{S}^A_{Y,\tau}(\bar{H}_\tau, \bar{W}_\tau, Y_\tau) + \mathscr{S}^A_{X,\tau+1}(\bar{Z}_\tau, X_{\tau+1}) \right\}.$$

Therefore, we have $\psi^\pi_{\bar{w}_t}(\bar{Z}_t) \in \mathscr{T}_t$. Since this argument applies to every scalar component $\psi^\pi_{\bar{w}_t}$ of the vector-valued score, the vector-valued score $\psi^\pi_t(\bar{Z}_t)$ lies component-wise in the tangent space $\mathscr{T}_t$.

**Step 2. Identification of $\psi^\pi_t$ as the efficient influence function.** We now show that the moment function $\psi^\pi_t(\bar{Z}_t)$ is in fact the efficient influence function for the target parameter $\theta_t$ in the semiparametric model defined above. Recall from Lemma 1, for the true distribution $P$, the true target parameter $\theta_t$, and the true nuisance functions $\eta_t = (m_t, p_t)$, we have the population moment condition

$$\mathbb{E}\left[\psi^\pi_t(\bar{Z}_t; \theta_t, \eta_t)\right] = 0.$$

Also, by construction of the model, this moment condition continues to hold along any regular parametric submodel of $P$ that preserves the randomized assignment law. Let $\{P_\alpha : \alpha \in (-\varepsilon, \varepsilon)\}$ be an arbitrary regular one-dimensional parametric submodel of the data generating process, dominated by $P = P_0$, with score function

$$S(\bar{Z}_t) = \frac{\partial}{\partial \alpha} \log p_\alpha(\bar{Z}_t) \Big|_{\alpha=0} \in \mathscr{T}_t,$$

where $p_\alpha$ denotes the density of $P_\alpha$. For each $\alpha$, let $\theta_t(\alpha) := \theta_t(P_\alpha)$ and $\eta_t(\alpha) := \eta_t(P_\alpha)$ be the target and nuisance functions implied by $P_\alpha$. By construction of the model and Lemma 1, these satisfy

$$\mathbb{E}_\alpha\left[\psi^\pi_t(\bar{Z}_t; \theta_t(\alpha), \eta_t(\alpha))\right] = 0, \quad \forall \alpha \in (-\varepsilon, \varepsilon).$$

Define

$$F(\alpha) := \mathbb{E}_\alpha \left[ \psi_t^\pi(\bar{Z}_t; \theta_t(\alpha), \eta_t(\alpha)) \right].$$

Then $F(\alpha) \equiv 0$ for all $\alpha$, so in particular $F'(0) = 0$. Here, we now expand $F'(0)$ using the chain rule in the three arguments: the target parameter $\theta_t$, the nuisance functions $\eta_t$, and the underlying distribution $P_\alpha$. By standard semiparametric calculus,

$$
\begin{aligned}
F'(0) &= \frac{\partial}{\partial \alpha} \left\{ \int \psi_t^\pi(\bar{z}_t; \theta_t(\alpha), \eta_t(\alpha)) p_\alpha(\bar{z}_t) \nu(d\bar{z}_t) \right\} \bigg|_{\alpha=0} \\
&= \int \frac{\partial}{\partial \alpha} \psi_t^\pi(\bar{z}_t; \theta_t(\alpha), \eta_t(\alpha)) \bigg|_{\alpha=0} p(\bar{z}_t) \nu(d\bar{z}_t) + \int \psi_t^\pi(\bar{z}_t; \theta_t(0), \eta_t(0)) \frac{\partial}{\partial \alpha} p_\alpha(\bar{z}_t) \bigg|_{\alpha=0} \nu(d\bar{z}_t) \\
&= \int \left\{ \partial_\theta \psi_t^\pi(\bar{z}_t; \theta_t, \eta_t) \theta_t'(0) + \partial_\eta \psi_t^\pi(\bar{z}_t; \theta_t, \eta_t)[\eta_t'(0)] \right\} p(\bar{z}_t) \nu(d\bar{z}_t) \\
&\quad + \int \psi_t^\pi(\bar{z}_t; \theta_t, \eta_t) S(\bar{z}_t) p(\bar{z}_t) \nu(d\bar{z}_t) \\
&= \underbrace{\mathbb{E}\left[ \psi_t^\pi(\bar{Z}_t; \theta_t, \eta_t) S(\bar{Z}_t) \right]}_{\text{variation of } P} + \underbrace{\mathbb{E}\left[ \partial_\theta \psi_t^\pi(\bar{Z}_t; \theta_t, \eta_t) \right] \theta_t'(0)}_{\text{variation of } \theta_t} + \underbrace{\mathbb{E}\left[ \partial_\eta \psi_t^\pi(\bar{Z}_t; \theta_t, \eta_t)[\eta_t'(0)] \right]}_{\text{variation of } \eta_t},
\end{aligned}
$$

where

$$\theta_t'(0) = \frac{d}{d\alpha} \theta_t(\alpha) \bigg|_{\alpha=0}, \quad \eta_t'(0) = \frac{d}{d\alpha} \eta_t(\alpha) \bigg|_{\alpha=0}.$$

First, recalling that $\psi_t^\pi = (\psi_{\bar{w}_t}^\pi)_{\bar{w}_t \in \bar{\mathcal{W}}_t}$ and each element $\psi_{\bar{w}_t}^\pi$ is

$$\psi_{\bar{w}_t}^\pi(\bar{Z}_t; \theta_t, m_t, p_t) = \frac{\mathbb{1}\{\bar{W}_t = \bar{w}_t\} \cdot (Y_t - \Gamma_{\bar{w}_t}^{(t)}(\bar{H}_t))}{\pi_{\bar{w}_t}} + A_{\bar{w}_t}^{(t)}(\bar{H}_t) + \Gamma_{\bar{w}_t}^{(1)}(X_1) - \mu_{\bar{w}_t}(t).$$

Hence, we have $\partial_{\mu_{\bar{w}_t}(t)} \psi_{\bar{w}_t}^\pi = -1$, so $\mathbb{E}\left[ \partial_\theta \psi_t^\pi(\bar{Z}_t; \theta_t, \eta_t) \right] = -I$.

Next, by Lemma 2, for any admissible direction $h$ in the nuisance function class,

$$\frac{\partial}{\partial r} \mathbb{E}\left[ \psi_t^\pi(\bar{Z}; \theta_t, \eta_t + rh) \right] \bigg|_{r=0} = \mathbb{E}\left[ \partial_\eta \psi_t^\pi(\bar{Z}; \theta_t, \eta_t)[h] \right] = 0.$$

Applying this with $h = \eta_t'(0)$ yields

$$\mathbb{E}\left[ \partial_\eta \psi_t^\pi(\bar{Z}; \theta_t, \eta_t)[\eta_t'(0)] \right] = 0.$$

Lastly, substituting the two results above into $F'(0) = 0$, we obtain

$$\theta_t'(0) = \mathbb{E}\left[ \psi_t^\pi(\bar{Z}_t; \theta_t, \eta_t) S(\bar{Z}_t) \right],$$

for every regular submodel score $S(\cdot) \in \mathcal{T}_t$. Therefore, $\psi_t^\pi$ is a gradient of $\theta_t$. Combining this gradient representation with Step 1, $\psi_t^\pi$ is the unique gradient lying in the tangent space $\mathcal{T}_t$, and hence it is the efficient influence function. Consequently, the semiparametric efficiency bound for $\theta_t$ is the covariance of this efficient influence function:

$$\Sigma_t = \text{Var}\left( \psi_t^\pi(\bar{Z}_t; \theta_t, m_t, p_t) \right).$$

**Step 3. Efficiency bound and attainment.** We finally establish the semiparametric efficiency bound and show that the dynamic regression-adjusted estimator attains it. Under Assumptions Assumptions 5-7, Theorem 4 provides the asymptotic linear expansion

$$\sqrt{n}(\widehat{\theta}_t - \theta_t) = \frac{1}{\sqrt{n}} \sum_{i=1}^n \psi_t^\pi(\bar{Z}_{i,t}; \theta_t, m_t, p_t) + o_p(1),$$

so $\widehat{\theta}_t$ is regular and asymptotically linear with the influence function equal to the efficient influence function $\psi_t^\pi$. Consequently, its asymptotic covariance equals $\text{Var}\left( \psi_t^\pi(\bar{Z}_t) \right)$, i.e., it achieves the semiparametric efficiency bound $\Sigma_t$. Hence, we have the desired result.

$\square$

### C.6. Proof of Corollary 6

**Corollary 6** (Variance reduction). *Under Assumptions 1-7, for any $t \in [T]$ and $\bar{w}_t \in \bar{\mathcal{W}}_t$, we have*

$$\mathrm{Var}\left(\widehat{\mu}_{\bar{w}_t}^{emp}(t)\right) \geq \mathrm{Var}\left(\widetilde{\mu}_{\bar{w}_t}^{dy\text{-}adj}(t)\right),$$

*where $\widetilde{\mu}_{\bar{w}_t}^{dy\text{-}adj}(t)$ is the dynamic regression-adjusted estimator that incorporates known adjustment terms.*

*Proof.* Consider the empirical estimator $\widehat{\mu}_{\bar{w}_t}^{emp}(t)$ defined in Eq. (3). Under Assumptions 1-4, $\widehat{\mu}_{\bar{w}_t}^{emp}(t)$ is an unbiased estimator of $\mu_{\bar{w}_t}(t)$. Then, it admits an influence function representation with influence function:

$$\psi_{\bar{w}_t}^{emp}(\bar{Z}_t) = \frac{\mathbb{1}\{\bar{W}_t = \bar{w}_t\}}{\pi_{\bar{w}_t}}(Y_t - \mu_{\bar{w}_t}(t)).$$

Let $\psi_t^{emp} = (\psi_{\bar{w}_t}^{emp})_{\bar{w}_t \in \bar{\mathcal{W}}_t}$ and denote $\Sigma_t^{emp} = \mathrm{Var}\left(\psi_t^{emp}(\bar{Z}_t)\right)$, so that $\Sigma_t^{emp}$ is the asymptotic covariance matrix of $\sqrt{n}(\widehat{\theta}_t^{emp} - \theta_t)$, where $\widehat{\theta}_t^{emp} = (\widehat{\mu}_{\bar{w}_t}^{emp})_{\bar{w}_t \in \bar{\mathcal{W}}_t}$.

By Theorem 5, $\psi_t^{\pi}(\bar{Z}_t; \theta_t, m_t, p_t)$ is the efficient influence function for the vector parameter $\theta_t = (\mu_{\bar{w}_t}(t))_{\bar{w}_t \in \bar{\mathcal{W}}_t}$, and therefore $\Sigma_t = \mathrm{Var}\left(\psi_t^{\pi}(\bar{Z}_t; \theta_t, m_t, p_t)\right)$ is the semiparametric efficiency bound. Moreover, the oracle dynamic regression-adjusted estimator $\widetilde{\mu}_{\bar{w}_t}^{dy\text{-}adj}(t)$ is regular and asymptotically linear with influence function $\psi_t^{\pi}$. Therefore, its asymptotic covariance equals $\Sigma_t$.

Since $\Sigma_t$ is the semiparametric efficiency bound for $\theta_t$, any regular estimator of $\theta_t$ must have an asymptotic covariance matrix that dominates $\Sigma_t$ in the Loewner order $\Sigma_t^{emp} - \Sigma_t \succeq 0$. Finally, using the asymptotic linear expansions as mentioned above, this inequality translates into the claimed variance reduction for the leading $1/n$ term of the estimator variance. $\qquad\square$

## D. Sufficient Conditions for Assumptions

Assumptions 6 and 7 are stated as high-level regularity conditions on the full nuisance vector. This section gives primitive sufficient conditions under which these assumptions hold. The conditions below are not necessary. Rather, they should be read as a practical checklist for applying the proposed estimator safely. In particular, Neyman orthogonality removes first-order local sensitivity to nuisance estimation errors, but it does not protect the estimator against gross misspecification, unstable transition simulation, insufficient treatment-path sample sizes, or applying the method outside static randomized regimes.

We first introduce operator notation. For a transition kernel $p_{\bar{w}_\tau}^{(\tau)}$, define the corresponding one-step integration operator by

$$(Q_{\bar{w}_\tau}^{(\tau)} f)(\bar{h}_\tau) = \int_{\mathcal{H}_{\tau+1}} f(\bar{h}_\tau, h_{\tau+1}) p_{\bar{w}_\tau}^{(\tau)}(dh_{\tau+1} \,|\, \bar{h}_\tau).$$

### D.1. Primitive Conditions

The following primitive conditions are sufficient for the stability and smoothness assumptions used in Section 5. Here, the transition-kernel error is measured by the same total-variation-based $L_2(P)$ norm defined in Appendix B.1.

(C1) The horizon $T$ and the number of feasible treatment histories $|\bar{\mathcal{W}}_t|$ are fixed. Moreover,

$$\underline{\pi}_t := \min_{\tau \leq t} \min_{\bar{w}_\tau \in \bar{\mathcal{W}}_\tau} \pi_{\bar{w}_\tau} > 0.$$

Since $|\bar{\mathcal{W}}_t|$ is finite, this is implied by Assumption 4 for each fixed $t$.

(C2) The outcome has a finite $(2 + \delta)$-moment for some $\delta > 0$:

$$\sup_{\tau \leq T} \mathbb{E}\left[|Y_\tau|^{2+\delta}\right] < \infty.$$

The true and estimated regression functions used in the recursive adjustment are uniformly bounded:

$$\sup_{\bar{w}_t, t, \bar{h}_t} |m^{(t)}_{\bar{w}_t}(\bar{h}_t)| \leq M_m, \qquad \sup_{\bar{w}_t, t, \bar{h}_t} |\widehat{m}^{(t)}_{\bar{w}_t}(\bar{h}_t)| \leq M_m.$$

The second bound can always be enforced by clipping the regression learner. If the outcome is uniformly bounded, then the first bound is automatically satisfied by the conditional mean regression function. For example, random forests with bounded outcomes satisfy the estimated-regression bound automatically because their predictions are convex averages of observed outcomes. With unbounded outcomes, the same bound can be enforced by clipping the predictions.

(C3) The true and estimated transition kernels are stable under the total-variation perturbation metric used in Appendix B.1. Specifically, for every pair of bounded measurable functions $f$ and $\widetilde{f}$ with $\|f\|_\infty \vee \|\widetilde{f}\|_\infty \leq M_f$,

$$\|\widehat{Q}^{(\tau)}_{\bar{w}_\tau} f - Q^{(\tau)}_{\bar{w}_\tau} \widetilde{f}\|_{P,2} \leq C_Q \left\{ \|f - \widetilde{f}\|_{P,2} + 2M_f \|\widehat{p}^{(\tau)}_{\bar{w}_\tau} - p^{(\tau)}_{\bar{w}_\tau}\|_{P,2} \right\}.$$

In particular,

$$\|(\widehat{Q}^{(\tau)}_{\bar{w}_\tau} - Q^{(\tau)}_{\bar{w}_\tau}) f\|_{P,2} \leq 2C_Q \|f\|_\infty \|\widehat{p}^{(\tau)}_{\bar{w}_\tau} - p^{(\tau)}_{\bar{w}_\tau}\|_{P,2}.$$

The first display follows from the contraction property of the true conditional-expectation operator, up to the fixed constants induced by the finite treatment-history and positivity conditions in (C1), and the second display follows directly from $|\int f \, d(\widehat{p}^{(\tau)}_{\bar{w}_\tau} - p^{(\tau)}_{\bar{w}_\tau})| \leq 2\|f\|_\infty d_{\mathrm{TV}}(\widehat{p}^{(\tau)}_{\bar{w}_\tau}, p^{(\tau)}_{\bar{w}_\tau})$. Therefore, small total-variation errors in the transition kernel do not get amplified in the fixed-horizon recursive construction, provided that the functions propagated through the transition operators are uniformly bounded as in (C2). When the kernels admit densities with respect to a common dominating measure, the total-variation term can be checked through $\frac{1}{2}\int |\widehat{p} - p|$ conditionally on the history. In applications where $p^{(\tau)}_{\bar{w}_\tau}$ is implemented through a conditional sampler, the same condition can be empirically verified through the induced one-step output law, rather than through an explicit density.

## D.2. Assumption 6

First, we consider the regression nuisance $m^{(t)}_{\bar{w}_t}$. Suppose $m^{(t)}_{\bar{w}_t}$ depends on only $s_m$ active coordinates of $\bar{H}_t$ and is Hölder smooth with smoothness $\beta_m$ in those active coordinates. If the supervised learner satisfies the standard nonparametric root-risk bound

$$\|\widehat{m}^{(t)}_{\bar{w}_t} - m^{(t)}_{\bar{w}_t}\|_{P,2} = O_p(n_{\bar{w}_t}^{-\frac{\beta_m}{2\beta_m + s_m}} \ell_n),$$

where $\ell_n$ is a polylogarithmic factor, then

$$\|\widehat{m}^{(t)}_{\bar{w}_t} - m^{(t)}_{\bar{w}_t}\|_{P,2} = o_p(n^{-1/4}),$$

whenever $\beta_m > s_m/2$, since $n_{\bar{w}_t}/n \to_p \pi_{\bar{w}_t} > 0$ holds under positivity and fixed feasible treatment histories. This condition formalizes the usual *low effective dimension plus sufficient smoothness* requirement. Honest random forests and generalized random forests are designed as adaptive local averaging estimators that exploit such low-dimensional heterogeneity and admit formal large-sample theory under regularity conditions (Athey et al., 2019). Therefore, random forests provide a natural choice for $m^{(t)}_{\bar{w}_t}$ when the regression signal is sparse, smooth, or approximately low-dimensional. In practice, out-of-fold prediction errors on each treatment path provide useful diagnostics for the plausibility and finite-sample stability of this risk condition, although they do not, by themselves, certify the asymptotic $L_2(P)$ rate.

For the transition nuisance $p^{(\tau)}_{\bar{w}_\tau}$, consider the following structural assumption. Suppose

$$H_{\tau+1} = a_{\tau, \bar{w}_\tau}(\bar{H}_\tau) + B_{\tau, \bar{w}_\tau}(\bar{H}_\tau)\varepsilon_\tau, \qquad \varepsilon_\tau \sim \nu_0,$$

where $\nu_0$ has a sufficiently regular density, such as a standard Gaussian density, and the associated location-scale family is locally Lipschitz in total variation (equivalently, in $L_1$ for densities) on compact nondegenerate parameter sets. Assume that $B_{\tau, \bar{w}_\tau}(\cdot)$ is represented by an identifiable scale parameterization, such as a Cholesky factor with positive diagonal entries, and that the singular values of both the true and fitted scale maps are uniformly bounded between two positive constants. If the conditional location map $a_{\tau, \bar{w}_\tau}$ and scale map $B_{\tau, \bar{w}_\tau}$ are Hölder or compositional Hölder functions with effective

dimension $s_p$ and smoothness $\beta_p$, and if they are estimated componentwise by sparse $\rho$-MLP estimators with suitably chosen architectures and negligible optimization error, then the standard sparse deep neural network (DNN) prediction bounds give

$$||\widehat{a}_{\tau,\bar{w}_\tau} - a_{\tau,\bar{w}_\tau}||_{P,2} + ||\widehat{B}_{\tau,\bar{w}_\tau} - B_{\tau,\bar{w}_\tau}||_{P,2} = O_p(n_{\bar{w}_\tau}^{-\frac{\beta_p}{2\beta_p + s_p}} \ell_n),$$

where $\ell_n$ is a polylogarithmic factor. Such rates are available, up to logarithmic factors, for ReLU networks over compositional Hölder classes (Schmidt-Hieber, 2020), for continuous piecewise-linear and locally quadratic activations over Hölder classes, including Swish $\rho(x) = x/(1 + e^{-x})$ (Ohn & Kim, 2019), and for sparse-penalized/adaptive DNN estimators and compositional extensions, with the locally quadratic compositional case subject to the additional smoothness caveat discussed by Ohn & Kim (2022).

Then, this rate is $o_p(n^{-1/4})$ whenever $\beta_p > s_p/2$ since $n_{\bar{w}_\tau}/n \to_p \pi_{\bar{w}_\tau} > 0$ under positivity and fixed feasible treatment histories. Therefore, the induced transition kernels satisfy

$$||\widehat{p}_{\bar{w}_\tau}^{(\tau)} - p_{\bar{w}_\tau}^{(\tau)}||_{P,2} \lesssim ||\widehat{a}_{\tau,\bar{w}_\tau} - a_{\tau,\bar{w}_\tau}||_{P,2} + ||\widehat{B}_{\tau,\bar{w}_\tau} - B_{\tau,\bar{w}_\tau}||_{P,2} = o_p(n^{-1/4}),$$

where the first inequality follows from the total-variation-local Lipschitzness of the regular nondegenerate location-scale family in its parameters. Consequently, Gaussian $\rho$-MLP transition models provide a natural choice for $p_{\bar{w}_\tau}^{(\tau)}$ when the conditional transition dynamics are smooth, nonlinear, and approximately low-dimensional. The same reasoning applies to other regular conditional density families whose parameter-to-law maps are locally Lipschitz in total variation on compact subsets of the nondegenerate parameter space. Since $T$ and the feasible treatment histories are fixed, these componentwise bounds imply $||\widehat{p}_t - p_t||_{P,2} = o_p(n^{-1/4})$.

### D.3. Assumption 7

We verify that conditions (C1)–(C3), with the total-variation norm convention in Appendix B.1, imply Assumption 7. The verification is deterministic: once the terminal regression functions are bounded and the one-step transition operators are stable in the total-variation metric, the backward recursion can only propagate perturbations through finitely many sums and products. This is the precise sense in which the recursive construction does not amplify nuisance estimation errors under a fixed horizon.

**Lemma 8** (Primitive sufficient conditions for uniform stability). *Suppose (C1)–(C3) hold. Then, for any $t \in [T]$, there exists a finite constant $C_t < \infty$ such that, for any treatment history $\bar{w}_t \in \bar{\mathcal{W}}_t$ and any two nuisance collections*

$$\eta = (m_{\bar{w}_t}^{(t)}, \{p_{\bar{w}_\tau}^{(\tau)}\}_{\tau=1}^{t-1}), \qquad \widetilde{\eta} = (\widetilde{m}_{\bar{w}_t}^{(t)}, \{\widetilde{p}_{\bar{w}_\tau}^{(\tau)}\}_{\tau=1}^{t-1}),$$

*that satisfy (C2)–(C3), the corresponding recursively defined quantities satisfy*

$$\max_{1 \leq \tau \leq t} \left\|\widetilde{\Gamma}_{\bar{w}_t}^{(\tau)} - \Gamma_{\bar{w}_t}^{(\tau)}\right\|_{P,2} + \left\|\widetilde{A}_{\bar{w}_t}^{(t)} - A_{\bar{w}_t}^{(t)}\right\|_{P,2} \leq C_t \left\{\left\|\widetilde{m}_{\bar{w}_t}^{(t)} - m_{\bar{w}_t}^{(t)}\right\|_{P,2} + \left(\sum_{\tau=1}^{t-1} \left\|\widetilde{p}_{\bar{w}_\tau}^{(\tau)} - p_{\bar{w}_\tau}^{(\tau)}\right\|_{P,2}^2\right)^{1/2}\right\}.$$

*Consequently, Assumption 7 holds with the linear modulus of continuity $\omega(r) = Cr$ for some finite constant $C < \infty$, uniformly over the finitely many $t \leq T$ and $\bar{w}_t \in \bar{\mathcal{W}}_t$.*

*Proof.* Fix $t \in [T]$ and $\bar{w}_t \in \bar{\mathcal{W}}_t$. Let $\widetilde{Q}_{\bar{w}_\tau}^{(\tau)}$ denote the one-step integration operator generated by $\widetilde{p}_{\bar{w}_\tau}^{(\tau)}$. Define

$$\epsilon_m := \left\|\widetilde{m}_{\bar{w}_t}^{(t)} - m_{\bar{w}_t}^{(t)}\right\|_{P,2}, \qquad \epsilon_{p,\tau} := \left\|\widetilde{p}_{\bar{w}_\tau}^{(\tau)} - p_{\bar{w}_\tau}^{(\tau)}\right\|_{P,2},$$

and

$$\rho_t(\widetilde{\eta}, \eta) := \epsilon_m + \left(\sum_{\tau=1}^{t-1} \epsilon_{p,\tau}^2\right)^{1/2}.$$

By (C2), the terminal functions $\Gamma_{\bar{w}_t}^{(t)} = m_{\bar{w}_t}^{(t)}$ and $\widetilde{\Gamma}_{\bar{w}_t}^{(t)} = \widetilde{m}_{\bar{w}_t}^{(t)}$ are uniformly bounded by $M_m$. Since each lower-order $\Gamma$ is obtained by integrating a bounded function with respect to a probability kernel,

$$\sup_{1 \leq \tau \leq t} \left\|\Gamma_{\bar{w}_t}^{(\tau)}\right\|_\infty \leq M_m, \qquad \sup_{1 \leq \tau \leq t} \left\|\widetilde{\Gamma}_{\bar{w}_t}^{(\tau)}\right\|_\infty \leq M_m.$$

Let

$$\Delta_\tau := \left\| \widetilde{\Gamma}^{(\tau)}_{\bar{w}_t} - \Gamma^{(\tau)}_{\bar{w}_t} \right\|_{P,2}.$$

At the terminal step, $\Delta_t = \epsilon_m$. For $\tau = t-1, \ldots, 1$, condition (C3) gives

$$\Delta_\tau = \left\| \widetilde{Q}^{(\tau)}_{\bar{w}_\tau} \widetilde{\Gamma}^{(\tau+1)}_{\bar{w}_t} - Q^{(\tau)}_{\bar{w}_\tau} \Gamma^{(\tau+1)}_{\bar{w}_t} \right\|_{P,2} \le C_Q \left\{ \Delta_{\tau+1} + 2 M_m \epsilon_{p,\tau} \right\}.$$

Iterating this finite recursion yields

$$\Delta_\tau \le C_Q^{t-\tau} \epsilon_m + 2 M_m \sum_{s=\tau}^{t-1} C_Q^{s-\tau+1} \epsilon_{p,s}.$$

Since $t$ is fixed, the right-hand side is bounded by $C_{\Gamma,t} \rho_t(\widetilde{\eta}, \eta)$ for a finite constant $C_{\Gamma,t}$ depending only on $t$, $C_Q$, and $M_m$. Hence,

$$\max_{1 \le \tau \le t} \left\| \widetilde{\Gamma}^{(\tau)}_{\bar{w}_t} - \Gamma^{(\tau)}_{\bar{w}_t} \right\|_{P,2} \le C_{\Gamma,t} \rho_t(\widetilde{\eta}, \eta).$$

It remains to control the auxiliary correction term. Define the one-step innovations

$$D^{\eta}_{\tau,\bar{w}_t}(\bar{H}_{\tau+1}) := \Gamma^{(\tau+1)}_{\bar{w}_t}(\bar{H}_{\tau+1}) - (Q^{(\tau)}_{\bar{w}_\tau} \Gamma^{(\tau+1)}_{\bar{w}_t})(\bar{H}_\tau),$$

$$D^{\widetilde{\eta}}_{\tau,\bar{w}_t}(\bar{H}_{\tau+1}) := \widetilde{\Gamma}^{(\tau+1)}_{\bar{w}_t}(\bar{H}_{\tau+1}) - (\widetilde{Q}^{(\tau)}_{\bar{w}_\tau} \widetilde{\Gamma}^{(\tau+1)}_{\bar{w}_t})(\bar{H}_\tau).$$

By the triangle inequality and (C3),

$$\left\| D^{\widetilde{\eta}}_{\tau,\bar{w}_t} - D^{\eta}_{\tau,\bar{w}_t} \right\|_{P,2} \le \Delta_{\tau+1} + \left\| \widetilde{Q}^{(\tau)}_{\bar{w}_\tau} \widetilde{\Gamma}^{(\tau+1)}_{\bar{w}_t} - Q^{(\tau)}_{\bar{w}_\tau} \Gamma^{(\tau+1)}_{\bar{w}_t} \right\|_{P,2}$$

$$\le (1 + C_Q) \Delta_{\tau+1} + 2 C_Q M_m \epsilon_{p,\tau}.$$

Using positivity in (C1) and the fact that indicators are bounded by one,

$$\left\| \widetilde{A}^{(t)}_{\bar{w}_t} - A^{(t)}_{\bar{w}_t} \right\|_{P,2} \le \sum_{\tau=1}^{t-1} \frac{1}{\pi_{\bar{w}_\tau}} \left\| D^{\widetilde{\eta}}_{\tau,\bar{w}_t} - D^{\eta}_{\tau,\bar{w}_t} \right\|_{P,2}$$

$$\le \frac{1}{\underline{\pi}_t} \sum_{\tau=1}^{t-1} \left\{ (1 + C_Q) \Delta_{\tau+1} + 2 C_Q M_m \epsilon_{p,\tau} \right\}.$$

Combining this display with the bound on $\Delta_\tau$ gives

$$\left\| \widetilde{A}^{(t)}_{\bar{w}_t} - A^{(t)}_{\bar{w}_t} \right\|_{P,2} \le C_{A,t} \rho_t(\widetilde{\eta}, \eta)$$

for some finite constant $C_{A,t}$. Taking $C_t = C_{\Gamma,t} + C_{A,t}$ proves the displayed bound. Since $T$ and $|\mathcal{W}_t|$ are fixed, taking the maximum over all relevant $t$ and $\bar{w}_t$ gives a finite global constant $C$ and therefore the modulus $\omega(r) = Cr$. □

Combining Lemma 8 with Assumption 6 gives the operational implication used in the proof of Theorem 4. If

$$\|\widehat{m}_t - m_t\|_{P,2} + \|\widehat{p}_t - p_t\|_{P,2} = o_p(n^{-1/4}),$$

then, for each fixed $t$,

$$\max_{\bar{w}_t \in \mathcal{W}_t} \left[ \max_{1 \le \tau \le t} \left\| \widehat{\Gamma}^{(\tau)}_{\bar{w}_t} - \Gamma^{(\tau)}_{\bar{w}_t} \right\|_{P,2} + \left\| \widehat{A}^{(t)}_{\pi,\bar{w}_t} - A^{(t)}_{\bar{w}_t} \right\|_{P,2} \right] = o_p(n^{-1/4}),$$

where $\widehat{A}^{(t)}_{\pi,\bar{w}_t}$ denotes the auxiliary correction constructed from the estimated nuisance functions with the true prefix normalizers. Thus, the recursive integration and innovation-correction steps inherit the primitive nuisance rate; they do not require a separate, faster rate for the constructed quantities.

This also clarifies how the simulation learners fit into the assumptions. Random forest regression satisfies the boundedness requirement in (C2) when outcomes are bounded, because its predictions are convex averages of observed outcomes. With unbounded outcomes, clipping the fitted values enforces the same requirement. This boundedness check is different from the rate verification in Appendix D.2, where random forests are invoked as adaptive local averaging estimators with formal large-sample theory under regularity conditions (Athey et al., 2019).

For the transition component, the Gaussian MLP sampler used in the simulation studies satisfies the stability requirement when its output law is kept in a compact nondegenerate location-scale family and its network maps have controlled layer operator norms. Lipschitz activations together with bounded layer operator norms control the Lipschitz constant of the whole transition network (Gouk et al., 2021). On a compact nondegenerate Gaussian location-scale family, the map from the location and scale parameters to the induced probability law is locally Lipschitz in total variation. Hence perturbations in the fitted MLP parameters induce controlled perturbations in $p_{\hat{w}_\tau}^{(\tau)}$ under the TV-based $L_2(P)$ norm. The same interpretation applies to the zero-inflated log-normal MLP used in the empirical analysis, provided the zero-inflation probabilities and scale parameters are bounded away from degeneracy.

## E. Experimental Setup

### E.1. Computing Infrastructure

All experiments were conducted on a MacBook Pro equipped with an Apple M4 Max processor and 64 GB of unified memory, running macOS Sequoia (version 15.6.1).

### E.2. Data Generating Process

To evaluate the effectiveness of our method in capturing complex, dynamic causal relationships, we generate synthetic datasets based on a coupled dynamical system. The data generating process is designed to emulate a scenario where the covariates evolve according to a continuous-time chaotic system, which is influenced by both the outcome history and the treatment assignments. The full data generating process is defined by the following system of equations:

$$\begin{aligned} Y_{i,t} &= g(X_{i,t}, Y_{i,t-1}) + \tau(t)W_{i,t} + \varepsilon_{i,t}, \\ dX_{i,t} &= Q \cdot f(Q^\top X_{i,t}, Y_{i,t-1}, W_{i,t})dt + \sigma_X dB_t, \end{aligned} \tag{14}$$

where $i$ indexes the unit and $t$ indexes time. The outcome $Y_{i,t}$ is determined by a nonlinear function $g$, a time-varying treatment effect $\tau(t)$, and a Gaussian noise term $\varepsilon_{i,t} \sim \mathcal{N}(0, \sigma_Y^2)$. The $d$-dimensional covariates $X_{i,t}$ evolve according to a stochastic differential equation (SDE), driven by a drift function $f$ and a Brownian motion term $dB_t$ with diffusion scale $\sigma_X$. The initial states are sampled as $X_{i,1} \sim \mathcal{N}(F, \sigma_0^2 I_d)$ with $Y_{i,0} = 0$. The covariate dynamics $f$ follow the rotated Lorenz-96 model (Karimi & Paul, 2010), which is widely recognized as a standard benchmark for modeling complex chaotic dynamics:

$$f(S_{i,t}, Y_{i,t}, W_{i,t}) = \left[ (S_{i,j+1,t} - S_{i,j-2,t})S_{i,j-1,t} - S_{i,j,t} + F_t \right]_{j \in [d]},$$

Crucially, to introduce feedback loops between the outcome, treatment, and covariates, we modify the external forcing term to be time-dependent and state-dependent:

$$F_t = F + 0.5Y_{i,t-1} + 0.2W_{i,t},$$

where $F$ is the baseline constant forcing and $F_t$ represents the effective forcing coupled with the outcome and treatment. This coupling creates a challenging causal structure where $Y$ and $W$ dynamically influence the future evolution of $X$. Next, the nonlinear outcome function $g$ and the time-varying treatment effect $\tau(t)$ are specified as follows:

$$\begin{aligned} g(X_{i,t}, Y_{i,t-1}) &= \exp\left(-(X_{i,1,t} - F)^2/5\right) + \sin(X_{i,2,t}X_{i,3,t}/3) + 0.8\log(1 + \exp(Y_{i,t-1})), \\ \tau(t) &= 0.2 - 0.6\exp(-t/4) + 0.4\exp(-3t/5), \end{aligned}$$

This specification includes exponential and sinusoidal interactions among covariate dimensions, alongside an autoregressive dependency on $Y_{i,t-1}$. Although the observational data is discrete, the underlying covariate dynamics are continuous. To ensure high-fidelity simulation of the chaotic system, we solve the SDE in Eq. (14) using the explicit fourth-order Runge-Kutta (RK4) method for the deterministic drift component. The stochastic component is incorporated via the Euler-Maruyama approximation (Maruyama, 1955) consistent with the integration steps. Lastly, Table 2 summarizes the parameter values of the data generating process.

*Table 2.* Parameters for data generating process

| Symbol | Description | Value |
|--------|-------------|-------|
| $\Delta t$ | time step | 0.01 |
| $F$ | external forcing | 4.0 |
| $\sigma_0$ | initial standard deviation of $X$ | 0.8 |
| $\sigma_Y$ | noise standard deviation of $Y$ | 0.2 |
| $\sigma_X$ | noise standard deviation of $X$ | 0.2 |

## F. Additional Results

### F.1. Simulation Study

Figure 4 illustrates the statistical properties (RMSE, average 95% CI length, and coverage probability) of different estimators using synthetic data with a sample size of $n = 500$ and $n = 2000$. The results are consistent with the findings for $n = 1000$ presented in the main text. Also, we can observe a clear convergence in performance and a further reduction in the variance of the estimators as the sample size increases from $n = 500$ to $n = 2000$. This scaling behavior highlights the practical utility of our framework in both data-constrained settings and large-scale online experiments, ensuring that the estimator remains semiparametrically efficient as established in Theorem 5.

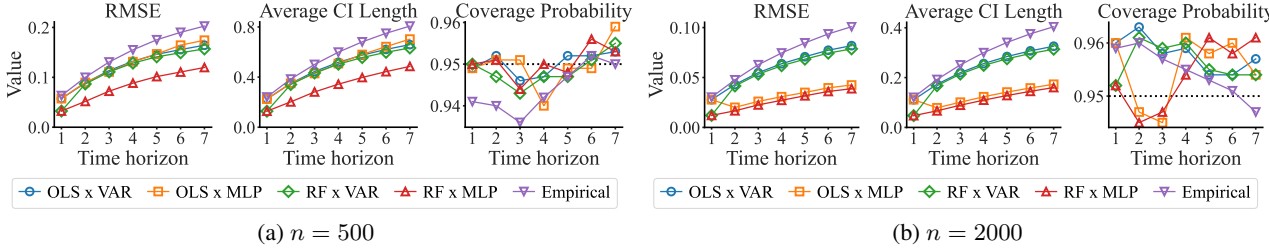

*Figure 4.* Additional statistical properties of different estimators on synthetic data with a sample size of (a) $n = 500$ and (b) $n = 2,000$.

### F.2. Performance Comparison

We compare our method to the two standard methods, g-formula and AIPW approaches, and report the RMSE at each time point below.

*Table 3.* RMSE comparison with g-formula and AIPW baselines on synthetic data with $n = 1000$.

| Time | 1 | 2 | 3 | 4 | 5 | 6 | 7 |
|------|------|------|------|------|------|------|------|
| Ours | 0.0188 | 0.0280 | 0.0385 | 0.0467 | 0.0550 | 0.0612 | 0.0669 |
| G-formula | 0.0195 | 0.0312 | 0.0438 | 0.0568 | 0.0687 | 0.0800 | 0.0932 |
| AIPW | 0.0188 | 0.0682 | 0.0895 | 0.107 | 0.123 | 0.135 | 0.146 |

### F.3. Monte Carlo Approximation

To evaluate the impact of Monte Carlo approximation error on the variance reduction gains, we conducted a sensitivity study with varying $S = \{16, 64, 256\}$. We can see that, for too small $S = 16$, the approximation error has a somewhat negative effect on the variance reduction gains, but for $S$ above a certain level, this effect is negligible.

### F.4. Computational Time

We measured the computational time of our algorithm using synthetic datasets of different lengths and sample sizes. As shown in the following results, the computational time scales approximately linearly with the sample size $n$ and quadratically with the time horizon $T$. This quadratic scaling with respect to $T$ is theoretically expected due to the recursive nature of the forward integration step in Algorithm 1.

*Table 4.* Effect of Monte Carlo sample size on RMSE for synthetic data with $n = 500$.

| Time | 1 | 2 | 3 | 4 | 5 | 6 | 7 |
|---|---|---|---|---|---|---|---|
| 16 | 0.0302 | 0.0486 | 0.0672 | 0.0843 | 0.100 | 0.114 | 0.130 |
| 64 | 0.0302 | 0.0473 | 0.0637 | 0.0799 | 0.0900 | 0.104 | 0.114 |
| 256 | 0.0302 | 0.0475 | 0.0644 | 0.0781 | 0.0900 | 0.105 | 0.112 |

*Table 5.* Computational times for different time horizons and sample sizes

| Time horizon $T$ | Sample size $n$ | Computational time (s) |
|---|---|---|
| 5 | 1000 | 55.361 |
| | 2000 | 91.630 |
| | 3000 | 150.895 |
| 10 | 1000 | 219.690 |
| | 2000 | 435.110 |
| | 3000 | 714.088 |
| 15 | 1000 | 600.408 |
| | 2000 | 1253.388 |
| | 3000 | 2216.349 |

