# OpenReview forum: "Modeling Covariate Transition for Efficient Estimation of Longitudinal Treatment Effects in Randomized Experiments"
_ICML.cc/2026/Conference — ICML 2026 regular_

### Official Review · Reviewer_1s3e · 2026-03-05

**Soundness:** 3
**Presentation:** 3
**Significance:** 2
**Originality:** 2
**Overall Recommendation:** 4
**Confidence:** 3

**Summary:**

The authors propose a method for estimating treatment effects in the longitudinal context. Specifically, they do so using a regression-adjustment method that is able to make use of post-randomization, time-varying covariates through the use of transition kernels. This allows for variance reduction, relative to naive techniques that are not regression-based, and versus traditional regression-based techniques which are only able to make use of baseline information. The method is flexible, allowing for a wide range of underlying machine learning models to be leveraged for the nuisance components. The authors present the method, demonstrate it's theoretical properties (semiparametric efficiency and asymptotic normality, under regularity conditions) and apply the technique to both simulated data and a real data example.

**Compliance With Llm Reviewing Policy:**

Affirmed.

**Final Justification:**

With the authors' proposed revisions, the major concerns I had were resolved (upgrading to a 4). I standby the initial commentary I made, with the authors addressing several of those points directly.

**Key Questions For Authors:**

**Q1. Conditions for verifying the assumptions**
Do you have checkable, sufficient conditions under which the various assumptions can be verified to hold? Alternatively, do you have a demonstration that specific model classes meet these assumptions (in particular cases, etc.).

The ability to address this, in a way that is practically useful, would dramatically improve my assessment of the soundness.

**Q2. Further methods for comparison**
Have you considered a deeper comparison against other baseline methods? Whether these are from the standard longitudinal causal inference literature, or simply regression-based techniques that only use baseline variates, etc.?

If yes, this would help to validate both (i) whether your proposed method provides a meaningful improvement over existing techniques, and (ii) where this improvement actually seems to stem from. For instance, if the comparison shows dramatic improvements over standard g-formula (using the same models), then this suggests your augmentation is doing the heavy lifting. This helps the soundness as well as the significance.

**Q3. Contrasting to dynamic policy evaluation**
Do you disagree with my framing regarding dynamic policy evaluation? That is, do you disagree that your context is a special case of the analysis of dynamic policies, and that methods that apply there are in fact relevant here as well? Have I missed something in this assessment?

This primarily addresses my soundness concerns.

**Q4. Practitioner guidance**
Do you have any specific guidance for attempts to apply your methods in practice? Whether this is related to model selection, hyperparameter tuning, verifying the assumptions, etc.? How should someone, who is not a methods researcher, take your techniques and use them in practice?

Addressing this would help improve both the presentation, and the significance of the paper.

**Limitations:**

The authors do spend some time discussing limitations of the work, though, I think they could be more clearly presented. One point that is important is that, if the time histories grow to be substantially long, there is likely extremely sparse representation of the treatment paths; this type of scaling does not appear to be discussed at all by the authors.

**Strengths And Weaknesses:**

## Soundness

### Strengths
- The theoretical claims in the paper are all well supported by proofs that appear to be correct. I have no overarching concerns with the veracity of what is claimed by the authors.
- The asymptotic properties demonstrated provide a substantial, and in certain settings, compelling rationale for the application of the methods. In terms of an asymptotic analysis, it seems to be fairly complete.
- The simulations that are considered are more thorough, as compared to many related toy problems, and do genuinely contend with several features of real-world data that are typically a nuisance. This renders them more relevant to an honest assessment of the techniques than many simulations would which, when supported by the theoretical analysis, is a powerful combination.
### Weaknesses
* The set of assumptions used to justify the theory (specifically, Assumptions 6 through 8) are both (i) quite strong, and (ii) quite abstract. These assumptions have the expected flavour of the types of assumptions required to establish the results, and so I do not object to them on this basis, however, I think that as composed they are incredibly difficult to leverage in practice. They are non-trivial to verify in any given situation, and the manuscript does not spend time attempting to establish their veracity for common settings, or presenting easier-to-verify sufficient versions that would be practically useful. Because of this there is a danger of individuals appealing to the theory for use of the methods inappropriately.
	* For specifics, I am concerned with Assumption 6 in high-dimensional environments, or when the temporal structure is sufficiently complex. With Assumption 7, it is not immediately clear to me when this stability is met for ML-based estimators, nor that recursive MC integration satisfies it.
* There is some concern with the way that the method is framed versus dynamic policy evaluation. Specifically, in the manuscript it states: "in other words, their applicability is restricted to settings where a well-defined policy exists. In contrast, our work addresses the estimation of arbitrary static longitudinal treatments." The setting that is being explored, where fixed, static regimes are contrasted, is fully subsumed by the dynamic setting. That is to say, there is no contrast here: the presented scenario is perfectly applicable to the setting that is being explored. This matters for two reasons: (i) it seems to suggest to readers that the manuscript solves a more involved problem (vis-a-vis the dynamic case), when the opposite is true, and (ii) it narrows the set of "applicable comparators". In fact, I think that it would be very worthwhile to consider how the proposed method compares to dynamic policy evaluation, even if those other techniques are "overkill" for the studied scenario.
* Despite their strength, I find the simulations insufficient in two distinct ways. First, they should likely include some level of additional complexity, specifically catered to stress-testing the assumptions (e.g., making the transitions difficult to correctly capture, etc.). Second, and perhaps more importantly, they should include other estimators to compare to. At a minimum they should compare to (i) regression-corrections that only use baseline values (which are specifically mentioned in the article); and (ii) a plug-in g-formula approach (with the same model choices as what the authors used) to isolate the impact of the augmentation. Additionally, (iii) AIPW, and/or (iv) a DR approach from policy evaluation broadly would be useful comparators that present a different set of trade-offs. I think it is likely, from the theory, that the proposed techniques do in fact outperform all of these techniques in this setting, however, the degree of this is critical (based on computational complexity, flexibility, other theoretical properties, utility outside of random settings, etc.).
	* It is important to note that, since the efficiency is an asymptotic result that requires correct model specification, there remains area for competing techniques to outperform the proposed methods, either because they have better finite sample guarantees (even if they are not asymptotically efficient) or because they are easier to specify (e.g., through double robustness, or related properties).
	* The lack of these types of additional guarantees (such as double robustness) is perhaps a knock against the method; however, if it turns out that the flexible choices for the nuisance models overcomes this, and that empirically the proposed technique performs better, this is perhaps able to be overlooked.

## Presentation
### Strengths
- Overall the paper is written well. It makes effective use of equations, the algorithm environment, and a clear structure to sufficiently discuss the estimation procedure. There are no major language concerns.
### Weaknesses
* For me, Figure 3b needs to be changed, as it is misleading. Specifically, the double axis should be removed, and likely the figure should just report the estimated SEs for each estimator (rather than overlaying percent differences). It feels intentionally designed to give a distorted view of the actual results.
* Related to the point above regarding the assumptions, I think that the lack of any practical guidance is a presentation concern here. There is little by way of discussion of how to select the transitional models, what  parameters ought to be used for the MC, etc. Even if a practitioner believed that the method could be influential for their work, it seems difficult to translate for this purpose, presently.
## Significance

### Strengths
* The manuscript addresses a legitimate gap, where many experiments (particularly digital ones) produce time-varying covariates. Methods that can effectively leverage these covariates, while maintaining the correct causal estimands, are critical.
* The method provides the capacity for substantial variance reduction, and does so in a theoretically justified manner. This is effective, and provides a strong foundation for other researchers to build from.
### Weaknesses
- The validity of the method relies heavily on the ability to correctly model the dynamics of the process; if these are incorrectly specified, it seems possible that mis-specified transition models may actually lead to variance increases. Without knowing how resilient it is to model specification, or how capable flexible modeling schemes are empirically, it is difficult to understand the immediate practical utility.
- The practical significance hinges on the empirical question of the methods performance in "real-world" type settings, which are underexplored. The asymptotic argument provides the starting point of a justification, however, this does not explain how it will perform under mis-specification, longer time horizons, or with more sparse data. As such, whether the technique has practical significance is difficult to judge without further empirical investigation (or a broader theoretical analysis that addresses these types of concerns).

## Originality

### Strengths
- The authors effectively combine existing ideas into a novel framework. Specifically, the use of transition kernels to propagate forward expectations, avoiding post-randomization biases, appears new to me in the context of randomized longitudinal experiments.
- The authors also further advance existing semiparametric, in novel directions.

### Weaknesses
- While the methods do appear new in the context of randomized longitudinal experiments, they appear quite similar to established ideas from RL/policy evaluation/longitudinal causal inference. This makes the contribution feel closer to a "transfer of ideas" rather than a completely novel creation.
- The proposed method has a very clear connection to the g-formula (Robins, 1986), which is not explicitly acknowledged. Specifically, it appears to me that the authors proposed methods are a (cleverly modified) version of the g-formula, which make the orthogonality/augmentation argument to improve efficiency. However, just like in the g-formula, the proposed methods rely on integrating the expressions over the covariate histories; the key difference seems to be in the "how". This does not necessarily undermine the authors' contributions, however, drawing an explicit parallel to this would be useful  for two reasons: (i) it draws a theoretical connection to important literature for the readers, and notably, literature that applies in the dynamic case, and (ii) it suggests another point of comparison that is worthwhile to include.

---

> ### Author Rebuttal · Authors · 2026-03-30
>
> We greatly appreciate the valuable suggestions and feedback on our manuscript. We will provide detailed responses below.
>
> **(1) Conditions for verifying the assumptions.** We thank the reviewer for raising this point. We provide practical sufficient conditions for Assumptions 6-8 below and will add them to the final version.
>
> **Assumption 6:** A rich body of literature has shown that modern ML models achieve the required $o_p(n^{-1/4})$ rate even in high-dimensional settings under sufficient structural conditions. Specifically, random forests achieve Assumption 6 for $m_t$ when the true function depends predominantly on a sparse subset of active features, or when it satisfies suitable smoothness and structural properties [1]. For $p_t$, Gaussian MLPs satisfy analogous sufficient conditions when the conditional mean/scale maps of the transition law are smooth enough (e.g., Hölder/compositional), so that the resulting kernel estimator attains the same $o_p(n^{-1/4})$ rate [2]. Furthermore, they possess a low-dimensional compositional structure, in which case the MLP can effectively mitigate the curse of dimensionality.
>
> **Assumption 7:** The uniform stability of the forward integration operators $\Gamma_{\bar{w}\_t}^{(\tau)}$ and the auxiliary correction term $A_{\bar{w}\_t}^{(t)}$ requires that small perturbations in the full nuisance functions $\eta_t=\lbrace m_{t},p_{t}\rbrace$ do not cause large changes in these components. Therefore, simple sufficient conditions are compact support of $(Y,X)$, bounded regression outputs (i.e., $\sup\vert\widehat{m}\_t\vert\le C$), treatment probabilities bounded away from zero, and uniformly bounded/Lipschitz transition operators induced by $\hat{p}\_t$. In our experiments, RF regression with bounded outcomes yields bounded $\hat m_t$ while MLP-based transition models with Lipschitz activations and bounded layer operator norms/output parameters induce uniformly bounded/Lipschitz transition operators [3]; with fixed $T$, these properties are inherited by the recursive constructions of $\Gamma_{\bar{w}\_t}^{(\tau)}$ and $A_{\bar{w}\_t}^{(t)}$.
>
> **Assumption 8:** The nuisance functions $\eta_t$ enter the moment function $\psi_{t}$ exclusively through recursive integrations over the transition kernels, as defined by $\Gamma_{\bar{w}\_t}^{(\tau)}$ and $A_{\bar{w}\_t}^{(t)}$. Thus, for fixed $T$, the mapping $\eta_t\mapsto\psi_{t}$ is a finite composition of bounded linear integral operators and products of nuisance components. Under boundedness, such maps are Gateaux-differentiable, with locally Lipschitz derivatives. Therefore, a practically useful sufficient condition for Assumption 8 is that the outcomes/covariates are a.s. bounded and that $m_t$ and the operators induced by $p_t$ are uniformly bounded—a condition closely related to the bounded-moment condition in Assumption 5.
>
> **(2) Further methods for comparison.** Thank you for the suggestion. We compare our method to the two standard methods, g-formula and aipw approaches, and report the RMSE at each time below.
> |Time|1|2|3|4|5|6|7|
> |---|---|---|---|---|---|---|---|
> |Ours|0.0188|0.0280|0.0385|0.0467|0.0550|0.0612|0.0669|
> |G-formula|0.0195|0.0312|0.0438|0.0568|0.0687|0.0800|0.0932|
> |AIPW|0.0188|0.0682|0.0895|0.107|0.123|0.135|0.146|
>
> **(3) Contrasting to dynamic policy evaluation.** Thank you for the valuable feedback. We acknowledge the reviewer’s framing regarding dynamic policy evaluation. We apologize for the misleading phrasing in Section 2 and will revise it. While dynamic methods apply, they are often statistically inefficient for static setups. Our estimator leverages the static structure to attain the semiparametric efficiency bound. We will add a detailed comparison to the final version.
>
> **(4) Practitioner guidance.** For non-expert practitioners, we recommend the following step-by-step guidance. First, for model selection, it is best to start with simple linear baselines for the nuisance functions $m_t$ and $p_t$ to establish a stable foundation. If the system’s dynamics are complex or linear models underperform, practitioners should scale up to nonlinear models. Second, regarding verifying assumptions, while theoretical conditions cannot be directly evaluated in real-world data, our cross-fitting procedure provides a built-in empirical diagnostic. Practitioners should rigorously monitor out-of-fold predictive performance (e.g., out-of-fold RMSE). High predictive accuracy without severe overfitting on the validation folds serves as a highly reliable, practical proxy for ensuring that the necessary structural assumptions are reasonably met. We will add such guidance to the final version.
>
> ---
> [1] Athey et al., Generalized Random Forests, Ann. Statist., 2019.
>
> [2] Schmidt-Hieber, Nonparametric regression using deep neural networks with ReLU activation function, Ann. Statist., 2020.
>
> [3] Gouk et al., Regularisation of Neural Networks by Enforcing Lipschitz Continuity, Machine Learning, 2021.

---

> > ### Author Rebuttal · Reviewer_1s3e · 2026-03-31
> >
> > Thank you for this! These resolve my major concerns with the work.

---

### Official Review · Reviewer_STMB · 2026-03-10

**Soundness:** 4
**Presentation:** 3
**Significance:** 4
**Originality:** 3
**Overall Recommendation:** 5
**Confidence:** 3

**Summary:**

Tech companies perform A/B tests, as we know, but a key point is that member A/B experiments are inherently longitudinal - each unit (the member/subscriber) produces multiple observations through time post-treatment. Tech companies also have a wealth of pre-treatment data that can be used for regression adjustment to achieve an estimator of reduced variance. However, standard techniques are limited to using pre-treatment covariates. Prior to this work, outcomes like $Y_{i,1}$, $Y_{i,2}$, $Y_{i,3}$ etc... could only be regressed on the baseline covariates recorded at time $t=1$ (since the historical data at time $t=1$ only contains pre-treatment covariates, i.e., $H_1 = X_1$ and $Y_0 = 0$). Naively conditioning on any subsequent post-treatment outcomes and covariates that dynamically evolve over time introduces bias, even in randomized settings.Longitudinal A/B tests typically run for quite a long time, during which the temporal behavior of a unit can vary as they adopt new viewing patterns and discover new content. As one might expect, outcomes at $t=30$ may not be well-predicted solely by outcomes and covariates at $t=1$. Hence, there is considerable opportunity for variance reduction by exploiting post-treatment histories, provided it does not bias the marginal causal estimand.The authors address this issue here by proposing a novel method whose moment condition coincides with the efficient influence function. To understand their approach, consider that if the transition kernels dictating covariate evolution were known perfectly, one could simply form the following forward-simulated estimator:

$\hat{\mu}_{\overline{w}_{t}}(t) = \frac{1}{n}\sum_{i=1}^{n}\Gamma_{\overline{w}_{t}}^{(1)}(X_{i,1})$

While this estimator would be unbiased, it leaves money on the table: it fails to leverage the actual outcomes and covariates observed within the experiment. To extract this in-experiment information, the authors subtract the conditional expectation of the trajectory (given the history) from its realized value at each time step. These step-by-step discrepancies are absorbed into an auxiliary correction term, $A^{(t)}$. Because this term is constructed as a mean-zero martingale difference sequence, it captures the stochastic deviations from the expected trajectories. This framework uses transition kernels and a recursive forward integration scheme to extract new variance-reducing information that is strictly orthogonal to pre-treatment covariates and past history, effectively shrinking the estimator's variance without injecting post-treatment bias.

**Compliance With Llm Reviewing Policy:**

Affirmed.

**Final Justification:**

This is a very well presented paper on a topic that is highly relevant and has very large scope for *longitudinal* A/B testing (e.g. on units that can be tracked through time, such as member of subscribers), which has received little attention. While there seems to be some discussion among other reviewers regarding novelty and superiority relative to other methods, I think this is nevertheless a good contribution to the literature.

**Key Questions For Authors:**

Could the authors please comment on whether this is a deployed solution at the tech company or whether this is a POC? Applied researchers would likely be interested in what computational considerations are required in order to scale this method in practice.

**Limitations:**

The limitations are clearly communicated following the conclusions. The authors discuss that their method is not robust to network interference, but this is commonly assumed away by SUTVA in most cases anyway.

**Strengths And Weaknesses:**

Significance: This has a very large impact and scope for A/B testing at tech companies, who perform A/B tests on members which can be tracked over time, which is typically the case when dealing with subscription models. Moreover effect sizes in these settings are typically small, and so any variance reduction that could be obtained would certainly be considered hugely advantageous.

Soundness: The authors squarely found their work within a potential outcomes framework and clearly state all assumptions: Sutva, independent units, random assignment, positivity. The propose a new estimator and prove Neymann orthogonality, asymptotic distribution of the estimator, and semiparametric efficiency. The authors empirical results also compelling - they consider a very challenging (Lorenz-96) simulated data generating process.

Presentation: While this is notationally very heavy, it is warranted and probably cannot be simplified further.

Originality: The authors have identified a clear methodological gap and have come up with a creative solution leveraging approaches from semiparametric efficiency theory.

---

> ### Author Rebuttal · Authors · 2026-03-30
>
> We greatly appreciate the valuable suggestions and comments on our manuscript and are encouraged by the positive score. We will provide detailed responses below.
>
> > Could the authors please comment on whether this is a deployed solution at the tech company or whether this is a POC?
> >
>
> Thank you for the question. Currently, this work serves as a POC to demonstrate the effectiveness and feasibility of our framework in a realistic setting. Although it has not yet been fully deployed in the tech company's production environment, the positive results from our offline experiments strongly suggest its potential for practical application. We are currently assessing what components will be required for future deployment.
>
> > Applied researchers would likely be interested in what computational considerations are required in order to scale this method in practice.
> >
>
> Thank you for the comment. We conducted additional experiments to assess computational time as the sample size $n$ and the time horizon $T$ varied. We used a random forest and a Gaussian MLP. The results are summarized in the table below.
>
> | Time horizon $T$ | Sample size $n$ | Computational time (s) |
> | --- | --- | --- |
> | 5 | 1000 | 55.361 |
> |  | 2000 | 91.630 |
> |  | 3000 | 150.895 |
> | 10 | 1000 | 219.690 |
> |  | 2000 | 435.110 |
> |  | 3000 | 714.088 |
> | 15 | 1000 | 600.408 |
> |  | 2000 | 1253.388 |
> |  | 3000 | 2216.349 |
>
> As we can see from the results, the computational time scales approximately linearly with the sample size $n$ and quadratically with the time horizon $T$. This quadratic scaling with respect to $T$ is theoretically expected due to the recursive nature of the forward integration step in Algorithm 1.
>
> To scale this method in practice for large-scale applications, several computational considerations can be applied:
>
> 1. **Parallelization:** Both nuisance function estimation across folds and Monte Carlo sampling for individual units in the recursive integration step are completely independent. Therefore, they can be highly parallelized across multiple CPUs or GPUs to significantly reduce wall-clock time.
> 2. **Markovian approximation:** Currently, our framework utilizes the entire historical trajectory $\bar{H}_t$ to capture long-term dependencies. However, to scale the method to large $T$, practitioners can introduce a $k$-th-order Markov assumption. By truncating the history and conditioning only on the most recent $k$ time steps rather than the full history, the input dimensions of models are strictly bounded. This prevents the state space from growing linearly with time and substantially reduces the computational overhead in both nuisance estimation and recursive integration.
> 3. **Choice of nuisance models:** The overall runtime is heavily influenced by the choice of supervised learning models and transition models. Unlike this experiment, using simpler, computationally efficient models (e.g., OLS or VAR) is expected to drastically reduce computation time compared to complex nonlinear models like deep neural networks, offering a practical trade-off.

---

> > ### Author Rebuttal · Reviewer_STMB · 2026-04-01
> >
> > Thank you for the response, I enjoyed the paper. I will keep my rating as accept.

---

### Official Review · Reviewer_wYMf · 2026-03-12

**Soundness:** 3
**Presentation:** 3
**Significance:** 3
**Originality:** 3
**Overall Recommendation:** 4
**Confidence:** 5

**Summary:**

This paper tackles the important and challenging problem of improving the precision of treatment effect estimates in longitudinal randomized controlled trials (RCTs). While regression adjustment using pre-treatment covariates is a well-established variance reduction technique, its extension to longitudinal settings is complicated by the presence of post-treatment covariates, which are themselves affected by the treatment and cannot be naively conditioned on without inducing bias.

The paper makes several significant contributions. First, it proposes a novel framework that models the dynamic evolution of covariates using transition kernels. By recursively integrating these kernels forward from baseline, the method constructs an expectation of the covariate trajectory under any treatment sequence, effectively creating a counterfactual model of how covariates would have evolved. Second, it introduces a dynamic regression-adjusted estimator (Equation 9) that combines (i) a residual term comparing observed outcomes to the model-implied expectation; (ii) a fully marginalized expectation term over all units; and (iii) an auxiliary correction term that captures the orthogonal innovations in the realized covariates, thereby extracting additional information for variance reduction without introducing bias. Third, the paper reformulates the estimation problem as a moment condition and proves Neyman orthogonality (Lemma 2), ensuring first-order robustness to errors in the nuisance functions (transition kernels and conditional mean functions). This crucial property allows the method to leverage flexible machine learning models for nuisance estimation while still delivering valid inference. Fourth, the paper establishes the asymptotic normality of the proposed estimator (Theorem 4) and, remarkably, proves that it attains the semiparametric efficiency bound (Theorem 5), demonstrating its theoretical optimality. Finally, the method is validated through a challenging simulation study based on a coupled Lorenz-96 chaotic system, showing substantial variance reduction compared to standard estimators.

**Compliance With Llm Reviewing Policy:**

Affirmed.

**Key Questions For Authors:**

\begin{enumerate}
    \item \textbf{Choice of transition kernel estimator:} The paper proposes using "models that support conditional sampling, such as vector autoregressions, Gaussian processes, or deep neural networks." The choice of estimator will likely impact finite-sample performance. In your simulations, what specific models did you use for $\hat{p}_{\bar{w}_\tau}^{(\tau)}$ and $\hat{m}_{\bar{w}_t}^{(t)}$? Were they correctly specified, and how sensitive are the results to the choice of learner?

    \item \textbf{Monte Carlo approximation error:} Algorithm 1 uses Monte Carlo sampling with $S$ draws to approximate the integral in Equation (8). How was $S$ chosen in the simulations? Could you provide guidance on the required size of $S$ relative to the time horizon $T$ and the dimensionality of the covariates? Is there a risk that approximation error could dominate the variance reduction gains?

    \item \textbf{Positivity and support:} Assumption 4 requires positivity for every treatment history $\bar{w}_t$. In practice, as $T$ grows, the number of possible histories grows exponentially, and some may have very few or zero observations. How does your method perform when the support for certain histories is sparse? Does the reliance on transition kernels help mitigate this issue compared to simply stratifying on the full history?

    \item \textbf{Interpretation of the auxiliary term:} The auxiliary correction term $A_{\bar{w}_t}^{(t)}(\bar{H}_t)$ in Equation (10) is crucial for variance reduction. Could you provide more intuition on why this term captures the "innovations" in the covariates and how its inclusion ensures orthogonality and eliminates bias? A simple illustrative example might be helpful for readers.

    \item \textbf{Extensions to non-binary treatments and continuous time:} The current framework assumes a discrete-time setting with a finite number of treatment arms. How might this approach be extended to settings with continuous treatments or to continuous-time processes (e.g., using neural ODEs to model the transition dynamics)? This could significantly broaden the applicability of the method.
\end{enumerate}

**Limitations:**

The authors' Impact Statement is insufficient for a paper with significant potential societal implications. The following limitations and potential negative consequences must be acknowledged:

\begin{enumerate}
    \item \textbf{Reliance on correct specification of transition kernels:} The method's validity hinges on the ability to consistently estimate the transition kernels $p_{\bar{w}_\tau}^{(\tau)}$. If these are severely misspecified, the auxiliary correction term may not fully capture the orthogonal innovations, potentially leading to biased estimates. While Neyman orthogonality provides robustness to small errors, large systematic misspecification could still compromise the results. Practitioners should be cautioned that model checking and validation of the transition dynamics are essential.

    \item \textbf{Complexity and accessibility:} The method is technically sophisticated, combining concepts from semiparametric theory, causal inference, and machine learning. This complexity may create a barrier to adoption for applied researchers who are not specialists in these areas. The paper would benefit from providing user-friendly, open-source software with clear documentation and default settings that work well across a range of applications, lowering the barrier to entry.

    \item \textbf{Computational cost:} While the method scales reasonably, the recursive integration and Monte Carlo sampling steps can be computationally intensive for long time horizons ($T$ large), large sample sizes ($n$ large), or high-dimensional covariates. This may limit its applicability in some big data settings with tight computational budgets. The authors should be transparent about these costs and discuss potential trade-offs (e.g., using fewer Monte Carlo samples $S$ in exchange for some increase in approximation error).

    \item \textbf{Potential for misuse in adaptive experiments:} The paper focuses on static, pre-specified treatment sequences. However, there is a risk that practitioners might attempt to apply this method to data from adaptive experiments (e.g., contextual bandits) where the treatment assignment mechanism depends on past outcomes. The assumptions (particularly Assumption 3, randomization) would be violated in such settings, and applying the method naively could lead to severely biased estimates. The paper should explicitly warn against this misuse.

    \item \textbf{Over-reliance on asymptotic approximations:} The theoretical results are asymptotic. In finite samples, especially with complex ML estimators for the nuisance functions, the coverage of confidence intervals may deviate from nominal levels. While the multiplier bootstrap is suggested as a finite-sample improvement, its performance in challenging settings (e.g., with highly nonlinear transition dynamics) should be further validated. Practitioners should be advised to use these inferential tools with caution, perhaps supplemented with sensitivity analyses.

\end{enumerate}

The authors should substantially revise the Impact Statement to address these concerns and provide responsible guidance to practitioners on the appropriate use and interpretation of the method.

**Strengths And Weaknesses:**

Soundness:
The paper is technically rigorous and the theoretical development is exceptionally sound. The core ideas are built on a solid foundation of semiparametric theory and causal inference. The key lemmas and theorems are correctly derived and clearly stated.

\begin{itemize}
    \item \textbf{Identification and Neyman Orthogonality:} The derivation of the moment condition and the proof of Neyman orthogonality (Lemma 2) are carefully executed. The decomposition of the moment function into martingale difference sequences in the proof of Theorem 5 elegantly shows its membership in the tangent space, which is a high-level technical contribution. This property is the cornerstone that justifies the use of complex, black-box ML models for nuisance estimation.
    \item \textbf{Asymptotic Theory:} The proof of asymptotic normality (Theorem 4) is thorough and well-structured. The four-step proof strategy (linear representation, bounding the empirical process remainder via cross-fitting, bounding the bias remainder via Neyman orthogonality and Lipschitz continuity, and finally the CLT) is clear and logically sound. The use of conditional variance arguments (Lemma 7) to control the cross-fitted empirical process term is elegant and correctly applied. The proof that the estimator achieves the semiparametric efficiency bound (Theorem 5) is a definitive theoretical achievement.
    \item \textbf{Simulation Design:} The choice of a Lorenz-96 chaotic system for the data generating process is inspired. It provides a realistic and highly challenging testbed for the proposed method, far beyond simple linear or additive models. The results clearly demonstrate the estimator's superior performance and its ability to reduce variance even in complex, nonlinear dynamical settings.
\end{itemize}

Presentation:
The paper is exceptionally well-written and organized. The exposition is clear, logical, and accessible, despite the complexity of the material.

\begin{itemize}
    \item \textbf{Clear Motivation:} The introduction and Section 2 effectively motivate the problem, explaining why simple regression adjustment fails in longitudinal settings and positioning the work within the existing literature.
    \item \textbf{Well-Structured Derivation:} The paper progresses logically from problem definition (Section 3) to the proposed estimator (Section 4.1), the dynamic extension with transition kernels (Section 4.2), and the crucial moment condition and Neyman orthogonality (Section 4.3, Lemmas 1-3). This step-by-step build-up helps the reader understand the rationale behind each component of the final estimator.
    \item \textbf{Excellent Notation:} Despite the complexity of longitudinal data, the notation (Table 1) is consistent, well-defined, and carefully maintained throughout the paper and the extensive appendix. This greatly aids readability.
    \item \textbf{Comprehensive Appendix:} The 15+ pages of appendices provide a wealth of supporting material, including detailed proofs, a primer on empirical process and semiparametric theory, and complete simulation specifications. This level of detail is commendable and enhances reproducibility.
\end{itemize}

Significance:
This paper makes a significant and lasting contribution to the field of causal inference and experimental design. Its importance can be assessed along several dimensions:

\begin{itemize}
    \item \textbf{Solves a Long-Standing Problem:} The challenge of using post-treatment covariates for variance reduction in longitudinal experiments without introducing bias has been a recognized issue for decades. This paper provides a principled and general solution by modeling the covariate transition process. The dynamic regression-adjusted estimator elegantly resolves the bias-variance trade-off.
    \item \textbf{Unlocks the Power of Modern ML:} By establishing Neyman orthogonality, the paper makes the framework compatible with any black-box machine learning model for estimating the nuisance functions. This is crucial for real-world applications where the true data-generating process is unknown and highly complex. It bridges the gap between classical semiparametric theory and modern machine learning practice.
    \item \textbf{Theoretical Optimality:} Proving that the estimator attains the semiparametric efficiency bound is a definitive result. It assures practitioners that, under the stated assumptions, no other regular estimator can achieve a lower asymptotic variance. This theoretical guarantee elevates the method beyond a heuristic and establishes it as a benchmark.
    \item \textbf{Practical Impact:} The method has direct applicability in a wide range of fields, from online A/B testing (where user behavior evolves over time) to clinical trials (where patient health markers are measured repeatedly). The simulation results demonstrate that the variance reduction can be substantial, potentially leading to more precise conclusions and shorter, more cost-effective experiments.
    \item \textbf{Extensible Framework:} The core idea of modeling transitions is not limited to the specific estimator presented. It could potentially be extended to handle more complex settings, such as survival outcomes, time-varying treatments, or general dynamic treatment regimes.
\end{itemize}

Originality:
The paper demonstrates a high degree of originality across multiple dimensions:

\begin{itemize}
    \item \textbf{Conceptual Originality:} The idea of using transition kernels to model covariate evolution and then using forward integration to create a counterfactual expectation for variance reduction is novel. This moves beyond simply "adjusting" for covariates and instead models the entire stochastic process. The auxiliary correction term that captures orthogonal innovations is a particularly clever and original device.
    \item \textbf{Methodological Originality:} The combination of transition kernel modeling with a Neyman-orthogonal moment condition to create an ML-compatible, semiparametrically efficient estimator is a significant methodological advance. While building on the double/debiased ML literature, the specific construction of the moment function for this longitudinal setting, with its recursive definitions and auxiliary term, is a new contribution.
    \item \textbf{Technical Originality:} The proof of the semiparametric efficiency bound (Theorem 5) and the demonstration that the moment function lies in the tangent space by decomposing it into a sum of martingale difference sequences is a technically elegant and original proof strategy. The control of the cross-fitted empirical process term using conditional variance bounds (in Theorem 4's proof) is also a clever adaptation of existing techniques.
    \item \textbf{Empirical Originality:} The use of a Lorenz-96 chaotic system to generate synthetic data is an original and powerful way to stress-test the method. It moves far beyond the typical linear or mildly nonlinear simulations and provides compelling evidence of the method's robustness and effectiveness in complex, realistic scenarios.
\end{itemize}

---

> ### Author Rebuttal · Authors · 2026-03-30
>
> We greatly appreciate the valuable suggestions and feedback on our manuscript and are encouraged by the positive score. We will provide detailed responses below.
>
> **(1) Choice of transition kernel estimator.** In our simulations, we used VAR and a Gaussian MLP for transition kernels, and OLS and random forests for regression models. Regarding transition kernels, a linear model cannot handle high nonlinearity via a Lorenz-96 model; in contrast, a nonlinear model can handle it and achieve substantial variance reduction. Please see Sec 6.1 for a detailed discussion.
>
> **(2) Monte Carlo approximation error.** In our simulations, we used $S=128$. Also, to evaluate the impact of Monte Carlo approximation error on the variance reduction gains, we conducted a sensitivity study with varying $S\in\lbrace16,64,256\rbrace$. Below, we report the RMSE at each time. We can see that, for too small $S=16$, the approximation error has a somewhat negative effect on the variance reduction gains, but for $S$ above a certain level, this effect is negligible.
> |Time|1|2|3|4|5|6|7|
> |---|---|---|---|---|---|---|---|
> |16|0.0302|0.0486|0.0672|0.0843|0.100|0.114|0.130|
> |64|0.0302|0.0473|0.0637|0.0799|0.0900|0.104|0.114|
> |256|0.0302|0.0475|0.0644|0.0781|0.0900|0.105|0.112|
>
> **(3) Positivity and support.** Our work focuses on static regimes. In other words, we can determine the number of treatment histories $\bar{w}_t$ before randomized experiments. Therefore, we ensure that each history receives sufficient observations by appropriately designing the experiments. And if we jointly learn the functions in which the treatment paths are encoded as inputs, transition kernels may help our algorithm work well when the support for certain histories is sparse. However, it also introduces a smoothing bias due to the sharing of unnecessary information across distinct paths. Since our goal is to estimate the parameter of interest as accurately as possible, we learn the functions separately by path. Accordingly, it cannot be expected that transition kernels mitigate the issue due to sparsity.
>
> **(4) Interpretation of the auxiliary term.** We view $A_{\bar{w}\_t}^{(t)}$ as a sequential control variate. Each summand in Eq. (10) is the realized continuation value after observing $\bar H_{t+1}$ minus its conditional expectation given $\bar H_t$ under the transition kernel. Hence, it keeps only the unexpected movement of post-treatment covariates while removing the predictable part. This is why it represents an innovation. Under the true kernel, each increment has a conditional mean of zero given the past, so it is orthogonal to any function of earlier history. Therefore, adding $A_{\bar{w}_t}^{(t)}$ does not change the estimand, but it uses the extra information revealed during follow-up to reduce variance. This same centering is also what yields Neyman orthogonality.
>
> **(5) Complexity and accessibility.** We will make the user-friendly source code publicly available upon acceptance to lower the barrier for researchers who are not specialists in these areas.
>
> **(6) Computational cost.** We measured computational time under various $T$ and $n$. **Please refer to the response (2) to Reviewer STMB for the results.**
>
> **(7) Revision of Impact Statement.** Thank you for the comments. We agree that it is important to add a warning against some misuses and guidance for practitioners to the Impact Statement. We will revise it in the final version as follows:
>
> This paper presents work whose goal is to advance the field of machine learning. There are many potential societal consequences of our work. Here, we highlight key points to ensure our method works well. First, while our method builds on the Neyman-orthogonal moment conditions, which provide first-order robustness to small errors in the nuisance estimate, severe misspecification can yield unexpected results. Then, it is essential to check and validate the nuisance functions. We reiterate that our work focuses on static regimes (i.e., pre-specified treatment sequences). Therefore, adaptive experiments (e.g., contextual bandits) are outside the scope of our method. Applying our method to such settings could lead to substantial bias. Also, the recursive integration and Monte Carlo sampling used in our algorithm are often computationally intensive, in particular for long time horizons, large sample sizes, and high-dimensional covariates. If the data is too large, our method may be impractical due to limited computational budgets. Lastly, the proposed estimator is asymptotically normal. However, the coverage of confidence intervals can deviate from the nominal level due to finite-sample bias. To improve finite-sample accuracy, the multiplier bootstrap is often an effective approach, but it still relies on empirical evaluations of the influence functions, meaning it cannot fully compensate for inference errors when nuisance models are extremely poorly estimated due to insufficient sample sizes.

---

> > ### Author Rebuttal · Reviewer_wYMf · 2026-04-03
> >
> > I thank the authors for addressing my questions, however, I will keep my score.

---

### Official Review · Reviewer_wHXo · 2026-03-13

**Soundness:** 3
**Presentation:** 4
**Significance:** 3
**Originality:** 2
**Overall Recommendation:** 3
**Confidence:** 3

**Summary:**

This paper proposes a DML-style methodology, together with theoretical guarantees, for estimating longitudinal treatment effects in randomized experiments by learning covariate transitions and using them for regression adjustment. The main idea is to model post-treatment covariate transitions rather than conditioning on post-treatment variables directly. The method uses orthogonalization and transition-based adjustment to incorporate evolving histories while aiming to preserve the marginal estimand. The paper establishes asymptotic normality and semiparametric efficiency under sufficiently fast learning rates for the nuisance components. The empirical section includes both synthetic data and a real application.

**Compliance With Llm Reviewing Policy:**

Affirmed.

**Final Justification:**

My main concern is the following. In a standard RCT setting, the propensity score is known. Because of this, even if the outcome regression is inconsistent, one would still expect $\sqrt{n}$-consistency, and as long as the outcome regression is merely consistent, semiparametric efficiency can be achieved. In particular, one does not need conditions such as an $n^{-1/4}$ rate for the outcome regression. This is one of the key advantages of the RCT setting, since it places only a very limited burden on estimating the outcome regression.

This work is also set in an RCT setting. Therefore, my understanding is that if one uses a DML procedure that fully exploits this fact, then semiparametric efficiency should still be achievable while placing only a minimal burden on the nuisance estimation. **However, the authors instead introduce two new nuisance functions and perform DML** based on them, and the condition they require for efficiency is that the product of the two nuisance errors must be $o(n^{-1/2})$. My question is: if we are already in an RCT setting, where a DML approach that uses the known propensity score would impose a much lighter nuisance burden, why is it necessary to introduce new nuisance functions and perform what appears to be a more difficult DML procedure?

The main point of their response is that their approach reduces the number of nuisance estimators that need to be learned. **However, I see a strong trade-off here. If one still follows the DML strategy in the style of [1], then I would expect $\sqrt{n}$-consistency even without strong assumptions on the outcome regression, and efficiency as long as the outcome regression is consistent.** In fact, I believe the authors themselves seem to acknowledge part of this point.

Therefore, my current conclusion is that, in the RCT setting considered in this paper, there exists a stronger competing approach, namely a DML procedure in the style of [1], which seems to dominate the authors' method in terms of nuisance burden. Of course, that alternative has the disadvantage that it requires estimating more functions $O(T^2)$. But in my view, its advantages are substantially greater.

That said, I do think the problem itself is very important, and I also find the authors' idea of performing DML through estimating covariate transitions to be novel and meaningful. However, I believe that this approach would become much more compelling in an observational setting rather than in an RCT setting. In that case, it could potentially be a stronger and more impactful contribution, although I also understand that many aspects of the current paper would then need to change significantly.

**Key Questions For Authors:**

1. Can the authors clarify more explicitly which part of the methodology fundamentally relies on randomization in the RCT setting, as opposed to being an orthogonal or DML-style construction that could in principle be extended to observational data with additional propensity-score nuisances?
2. How difficult is transition-kernel estimation in the intended applications when the post-treatment state is moderately or highly dimensional? If this still depends on data from specific treatment paths, the effective sample size may become very small and the estimation may become unstable. Is the treatment path encoded as an input and learned jointly, or is the estimation done separately by path?
3. Since the propensity score is known in an RCT, I also wonder whether one could define the outcome-side nuisance differently and obtain an effective method without explicitly modeling covariate transitions.

**Limitations:**

Most of the limitations I see are already described in the weakness section. My biggest concern is that nuisance learning, especially the transition-related part, seems potentially very difficult in practice. I am also somewhat unconvinced that a setup focusing on the outcome at a particular time $t$, rather than a cumulative outcome, is the most natural target in many applications.

**Strengths And Weaknesses:**

**Strengths**

- The paper studies an important problem: estimating longitudinal treatment effects in randomized controlled trials using a DML-style methodology that learns covariate transitions as nuisance components.
- If the two nuisance components, namely the covariate transition model and the outcome regression, satisfy sufficiently fast learning rates, the resulting estimator is asymptotically normal and semiparametrically efficient. I think this is a fairly strong theoretical result.
- The experimental results are also fairly strong. The paper validates the theory on synthetic data and then applies the method to real data.

**Weaknesses**

- First, the technical novelty seems somewhat limited. DML for dynamic treatment regimes has already been studied extensively, so my impression is that the proof may largely follow standard DML ideas with additional bookkeeping.
- Second, learning the nuisance functions may be very difficult in practice. In particular, both $m$ and $p$ depend on the treatment path $\bar{w}_t$, which suggests that the effective sample size available for nuisance learning may shrink very quickly with the length of the horizon. It is therefore not obvious to me that the nuisance-product error can be made sufficiently small in realistic settings. I would like to understand better how the method avoids this difficulty.
- Third, in many dynamic-treatment settings one is more interested in cumulative outcomes, whereas this paper focuses on the outcome at a specific time $t$. That target feels somewhat unusual to me, and I would like to better understand whether there are many important real-world applications in which this is the primary estimand of interest.

---

> ### Author Rebuttal · Authors · 2026-03-31
>
> We greatly appreciate the valuable suggestions and comments on our manuscript. We will provide detailed responses to address your main concerns below.
>
> **(1) Extension to observational data analysis.** Thanks for this insightful question. As you pointed out, the core idea of our methodology is not confined to the RCT setting. Indeed, the proposed method can be extended to observational data by estimating the propensity score, provided standard assumptions hold. We chose to focus on the RCT setup mainly because the empirical data at hand satisfy the RCT setup and also because we found that more researchers are facing the issue we tackle in this context. It is worth noting that even in the observational case, incorporating covariates beyond those used for treatment assignment estimation still yields variance reduction, although some extra theoretical work is needed. We will clarify this point more explicitly in the revised version.
>
> **(2) Difficulty in transition-kernel estimation.** Thank you for raising this important point. We agree that estimating transition kernels becomes challenging when the post-treatment state is moderately or high-dimensional. However, fully nonparametric estimation in arbitrarily high-dimensional, highly branching sequential problems is known to be intractable due to the curse of dimensionality, necessitating structural or semiparametric assumptions [1]. In our work, rather than making such assumptions, we focus on a randomized experiment over a small, prespecified set of static regimes. In such settings, we ensure that each history receives sufficient observations by appropriately designing the experiments. In addition, we also see moderately or even highly dimensional post-treatment states as a natural extension of our framework, provided there is credible structure to support it. For example, estimating the transition kernel as in [2] is one of promising approaches for high-dimensional settings when credible graphical structure is available. We will revise our manuscript to clarify this scope more explicitly.
>
> **(3) Separate estimation or joint learning.** Our goal is to estimate the parameter of interest as accurately as possible. Therefore, we learn the functions separately by path. This is because jointly learning a single function, where the treatment paths are encoded as inputs, can introduce a smoothing bias from the sharing of unnecessary information across distinct paths [3]. In contrast, estimating separately by path maximizes the flexibility of estimation for each path.
>
> **(4) Outcome-side nuisance.** Thank you for the suggestion. We agree that defining the nuisance on the outcome side recursively is a mathematically valid alternative. However, we deliberately chose the forward-integration approach with explicit transition kernels $p_{\bar{w}_\tau}^{(\tau)}$ for the following critical design reasons:
> - (i) *Incorporation of system dynamics*: Explicit modeling allows us to directly incorporate domain knowledge regarding system dynamics. This is highly beneficial in domains with structural constraints.
>
> - (ii) *Generative simulation and flexibility*: By retaining a forward generative model, analytical integration can be effectively approximated using Monte Carlo integration. This generative nature allows researchers to easily simulate and visualize arbitrary counterfactual trajectories.
>
> - (iii) *Foundation for continuous-time extensions*: Explicitly capturing how post-treatment covariates evolve provides a natural pathway for addressing irregular sampling intervals and missing data through continuous-time modeling.
>
> **(5) Outcome types.** We recognize that prior studies, especially in reinforcement learning, standardly formulate the target parameter as a cumulative outcome. However, we focus on outcomes at specific time steps, as these yield essential insights into the temporal dynamics of treatment effects. A cumulative metric aggregates the entire trajectory into a single value, potentially obscuring essential temporal dynamics, such as whether the treatment yields an immediate impact, a delayed effect, or persistence over time. In comparison, step-wise evaluation enables us to characterize the detailed temporal profile of how systems respond to interventions. This level of granularity is valuable in practice for interpretability and for precisely identifying the timing of effects. Although our methodology can be extended to cumulative outcomes by aggregating stepwise estimates, we intentionally emphasize time-specific outcomes to highlight their temporal interpretability.
>
> ---
> [1] Susan A. Murphy, Optimal dynamic treatment regimes, Journal of the Royal Statistical Society Series B, 2003.
>
> [2] Vandermeulen et al., Breaking the curse of dimensionality in structured density estimation, NeurIPS 2024.
>
> [3] Künzel et al., Metalearners for estimating heterogeneous treatment effects using machine learning, Proceedings of the National Academy of Sciences, 2019.

---

> > ### Author Rebuttal · Reviewer_wHXo · 2026-04-01
> >
> > Thank you for the detailed rebuttal. I still have one important concern regarding related work and the paper’s positioning.
> >
> > I am quite familiar with Bradic, Ji, and Zhang, “High-dimensional Inference for Dynamic Treatment Effects” (Annals of Statistics, 2024), and I was surprised not to see a comparison to that line of work. Although that paper studies cumulative outcomes, it seems to share important conceptual similarities with the present work, so I would appreciate clarification on the relationship.
> >
> > At a high level, my question is the following. In randomized settings, where the propensity score is known, one might expect semiparametric efficiency to be attainable under relatively mild requirements on the outcome-side nuisance functions (Bradic, Ji and Zhang). By contrast, the efficiency result in the present paper appears to rely on learning two potentially difficult nuisance objects, namely the outcome regression and the covariate transition kernel, at sufficiently fast rates. This makes me wonder why the proposed approach introduces an additional transition-kernel nuisance, rather than relying on an existing dynamic DR/DML framework (Bradic-style) under which the nuisance burden may simplify substantially in the randomized case.
> >
> > More specifically, I would appreciate the authors’ response to the following questions:
> > 1. Why is the Bradic-Ji-Zhang line of work not discussed in the related-work section, given how conceptually close it seems to the present paper?
> >
> > 2. In the RCT special case considered here, why is it necessary to learn the additional transition-kernel nuisance in order to obtain the claimed efficiency gains, compared with Bardic-style DML?
> >
> > 3. More concretely, if one applies an existing dynamic DR/DML estimator in the randomized setting, does one already obtain an estimator with lighter nuisance requirements and comparable efficiency guarantees? If not, could the authors explain precisely where the proposed transition-kernel formulation yields a genuinely different or stronger result?

---

> > > ### Author Response · Authors · 2026-04-07
> > >
> > > We are grateful to the reviewer for her/his insightful comments. We agree with the reviewer that a dynamic doubly robust estimator can utilize the known propensity score in an RCT setting, thereby not necessarily requiring the estimation of transition kernels when targeting a single expected final outcome $\mathbb{E}[Y(\bar{w}\_T)]$. However, when the objective is to estimate the full collection of expected potential outcomes $\lbrace\mathbb{E}[Y\_t(\bar{w}\_t)]\rbrace_{t=1}^T$, explicitly modeling covariate transitions offers distinct advantages. Specifically, while our approach requires estimating transition kernels in addition to mean regression functions, the total number of nuisance functions reduces from $O(T^2)$  to $O(T)$. Please see our point-by-point response to the reviewer's questions below.
> > >
> > > > 1. Why is the Bradic-Ji-Zhang line of work not discussed in the related-work section, given how conceptually close it seems to the present paper?
> > >
> > > We thank the reviewer for pointing this out. Our literature review focused on methods based on the Neyman-orthogonal moment conditions for longitudinal treatment effects, with [2] serving as the key reference, as our dynamic regression-adjusted estimator likewise builds upon this framework. Although [1] is also based on this framework, this work specifically focuses on how to use a sequential doubly robust representation to solve complex problems arising when estimating expected potential outcomes in the presence of high-dimensional confounders. While both works address longitudinal treatment effects, differences in their focus and objectives led us to center our discussion on [2]. We will incorporate a discussion of [1] in the revised version and further situate our approach within the broader landscape of these frameworks.
> > >
> > > > 2. In the RCT special case considered here, why is it necessary to learn the additional transition-kernel nuisance in order to obtain the claimed efficiency gains, compared with Bardic-style DML?
> > >
> > > > 3. More concretely, if one applies an existing dynamic DR/DML estimator in the randomized setting, does one already obtain an estimator with lighter nuisance requirements and comparable efficiency guarantees? If not, could the authors explain precisely where the proposed transition-kernel formulation yields a genuinely different or stronger result?
> > >
> > > Thank you for the question. The approach using transition kernels yields to lighter nuisance requirements when we require to estimate all expected potential outcomes $\lbrace\mathbb{E}[Y\_t(\bar{w}\_t)]\rbrace_{t=1}^T$, rather than a single cumulative outcome $\mathbb{E}[Y(\bar{w}_T)]$. Our main goal is to capture the temporal evolution of treatment effects over time, such as immediate impacts, delayed effects, or persistence, which requires estimating the expected potential outcome $\mathbb{E}[Y\_t(\bar{w}\_t)]$ for every time step $t=1,\ldots,T$. We compare the nuisance requirements of the two approaches below.
> > >
> > > If one applies [1] to estimate all expected potential outcomes $\lbrace\mathbb{E}[Y_t(\bar{w}\_t)]\rbrace\_{t=1}^T$, the procedure must be run separately at each target time $t$, because intermediate conditional means for one target time generally cannot be reused for another. Consequently, estimating the full trajectory over horizon $T$ requires $T(T+1)/2$ nuisance functions.
> > >
> > > In contrast, our formulation using transition kernels amortizes post-treatment dynamics across target times. To estimate $\lbrace\mathbb{E}[Y_t(\bar{w}\_t)]\rbrace\_{t=1}^T$, we only need to estimate $T$ mean regression functions $\lbrace m_{\bar{w}\_t}^{(t)}(\cdot)\rbrace\_{t=1}^T$ and $T-1$ transition kernels $\lbrace p_{\bar{w}\_\tau}^{(\tau)}(\cdot\mid\bar{h}\_\tau)\rbrace\_{\tau=1}^{T-1}$, totaling $2T-1$ nuisance models.
> > >
> > > The shared dynamic nuisance parameterization via transition kernels reduces the burden from $O(T^2)$ to $O(T)$, which in turn reduces the risk that estimation noise or instability in individual nuisance components will propagate across the full set of estimated potential outcomes.
> > >
> > > We will cite [1] and incorporate this discussion into the revised manuscript, which will significantly strengthen the work. We are grateful once again for the opportunity to engage in such a constructive and insightful exchange.
> > >
> > > ---
> > > [1] Bradic et al., High-dimensional Inference for Dynamic Treatment Effects, Ann. Statist., 2024.
> > >
> > > [2] Lewis & Syrgkanis, Double/debiased machine learning for dynamic treatment effect, NeurIPS 2021.

---

### Decision · Program_Chairs · 2026-04-30

**Decision:**

Accept (regular)

**Comment:**

This work looks at variance reduction in longitudinal RCTs using post-treatment covariates. The authors approach this problem by modeling covariate evolution via transition kernels and constructing a Neyman-orthogonal moment condition around them. Reviewers wYMf and STMB, found the theoretical contributions (semiparametric efficiency, asymptotic normality, Neyman orthogonality) to be sound and significant. Reviewer STMB specifically highlighted the large practical scope for A/B testing at tech companies with longitudinal user tracking. The simulation design using a Lorenz-96 chaotic system is notably rigorous. The rebuttal addressed the majority of concerns. Practical sufficient conditions for the key assumptions were provided, additional comparisons to g-formula and AIPW were included, and practitioner guidance was offered. Reviewers 1s3e and STMB both marked their concerns fully resolved.

Reviewer wHXo's critque is that in an RCT, the propensity score is known, so a standard DML approach can achieve semiparametric efficiency under very weak conditions on the outcome regression. Rather than using this, the proposed method instead introduces two nuisance functions (outcome regression and transition kernel) and requires their product error to decay at a sufficient rate. The authors justify this by noting their approach reduces the total number of nuisance models needed to estimate the full treatment effect trajectory. I am inclined to agree, but the authors should include a discussion and comparison to Bradic-Ji-Zhang.

The authors should be sure to incorporate the promised revisions. Namely, a discussion of and comparison to the Bradic-Ji-Zhang line of work, practical sufficient conditions for Assumptions 6 through 8, additional baseline comparisons (g-formula, AIPW), revised Impact Statement, and clarified framing relative to dynamic policy evaluation.